# THEMIS: TOWARDS HOLISTIC EVALUATION OF MLLMS FOR SCIENTIFIC PAPER FRAUD FORENSICS

**Tzu-Yen Ma**[*], **Bo Zhang**[*], **Zichen Tang, Junpeng Ding, Haolin Tian, Yuanze Li,
Zhuodi Hao, Zixin Ding, Zirui Wang, Xinyu Yu, Shiyao Peng, Yizhuo Zhao,
Ruomeng Jiang, Yiling Huang, Peizhi Zhao, Jiayuan Chen, Weisheng Tan,
Haocheng Gao, Yang Liu, Jiacheng Liu, Zhongjun Yang, Jiayu Huang, Haihong E**[†]
Beijing University of Posts and Telecommunications

🐙 https://bupt-reasoning-lab.github.io/THEMIS
⬛ https://github.com/BUPT-Reasoning-Lab/THEMIS
🤗 https://huggingface.co/datasets/BUPT-Reasoning-Lab/THEMIS

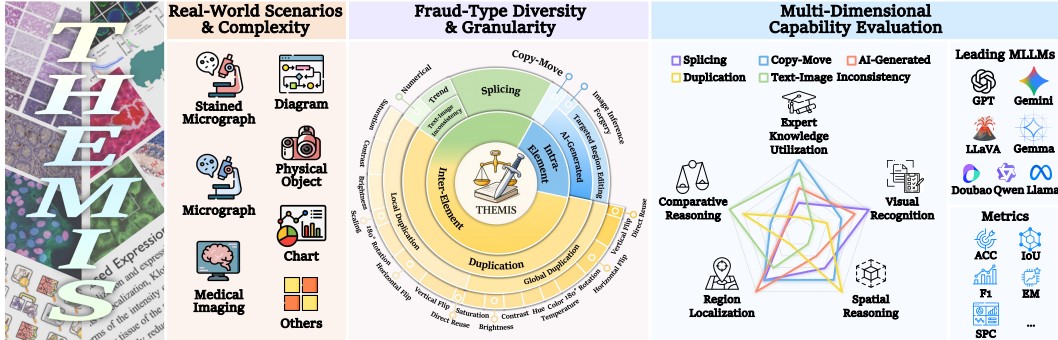

Figure 1: **Overview of THEMIS.** (1) **Real-World Scenarios and Complexity**, comprising over 4,000 questions across 7 representative scenarios; (2) **Fraud-Type Diversity and Granularity**, covering 5 challenging fraud methods with 16 fine-grained manipulation operations; (3) **Multi-Dimensional Capability Evaluation**, mapping fraud types to 5 core reasoning capabilities.

## ABSTRACT

We present **THEMIS**, a novel multi-task benchmark designed to comprehensively evaluate multimodal large language models (MLLMs) on visual fraud reasoning within real-world academic scenarios. Compared to existing benchmarks, THEMIS introduces three major advances. (1) **Real-World Scenarios and Complexity**: Our benchmark comprises over 4,000 questions spanning seven scenarios, derived from authentic retracted-paper cases and carefully curated multimodal synthetic data. With 60.47% complex-texture images, THEMIS bridges the critical gap between existing benchmarks and the complexity of real-world academic fraud. (2) **Fraud-Type Diversity and Granularity**: THEMIS systematically covers five challenging fraud types and introduces 16 fine-grained manipulation operations. On average, each sample undergoes multiple stacked manipulation operations, with the diversity and difficulty of these manipulations demanding a high level of visual fraud reasoning from the models. (3) **Multi-Dimensional Capability Evaluation**: We establish a mapping from fraud types to five core visual fraud reasoning capabilities, thereby enabling an evaluation that reveals the distinct strengths and specific weaknesses of different models across these core capabilities. Experiments on 16 leading MLLMs show that even the best-performing model, GPT-5, achieves an overall performance of only 56.15%, demonstrating that our benchmark presents a stringent test. We expect THEMIS to advance the development of MLLMs for complex, real-world fraud reasoning tasks.

---

[*]Equal contribution. [†]Corresponding author.

# 1 INTRODUCTION

Multimodal large language models (MLLMs) (Singh et al., 2025; OpenAI, 2025; Bai et al., 2023; 2025; Wang et al., 2025c; Meta, 2025) have emerged as a powerful paradigm for unifying vision and language, driving rapid progress in multimodal understanding and reasoning. Early studies have primarily validated MLLMs on simple visual capabilities, including fundamental perceptual skills such as object recognition, grounding, and caption generation (Plummer et al., 2015; Chen et al., 2015). Recent research has further demonstrated that MLLMs also exhibit advanced abilities, extending beyond basic perception to more complex benchmarks such as comprehensive multimodal evaluation and fine-grained capability assessment (Liu et al., 2025b; Zhang et al., 2025).

Despite the demonstrated progress on **simple** and **advanced** visual capabilities, **expert-level** visual capabilities, the ability to perform robust and comprehensive reasoning in realistic, complex scenarios, remain insufficiently explored and rigorously validated. To probe whether MLLMs truly possess such competence in deep visual understanding and reasoning, we adopt scientific paper fraud as the boundary scenario for evaluation, as such forensics demand not only pixel-level anomaly perception but also an understanding of the underlying scientific context and logical consistency within the image. This setting introduces unprecedented challenges, raising the demands on intrinsic visual reasoning depth, precision, and robustness to a level unseen in existing benchmarks.

A major bottleneck in pursuing this agenda is the absence of appropriate benchmarks. Existing benchmarks (Li et al., 2025b; Zhou & Hong, 2024; Shi et al., 2025; Wang et al., 2025b; Liu et al., 2025a; Wang et al., 2025a) generally fall short of meeting the demands of expert-level visual reasoning in terms of **real-world scenario complexity**, **fine-grained fraud-type diversity**, and **multi-dimensional capability evaluation**.

To fill this gap, as shown in Figure 1, we present **THEMIS**, a holistic multi-task benchmark of 4,054 questions derived from authentic retracted-paper cases and synthetic data, to systematically evaluate the fine-grained visual fraud reasoning abilities of MLLMs. By grounding evaluation in realistic contexts, THEMIS provides the foundation for probing expert-level capability. Specifically, THEMIS achieves this through three core design principles:

- **Real-World Scenarios and Complexity.** THEMIS spans seven representative academic scenarios (e.g., Micrograph and Medical Imaging), derived from authentic retracted cases and carefully constructed synthetic data, ensuring realism and controllability. Importantly, 60.47% of the images—primarily micrographs, physical objects, and medical imaging samples—contain complex textures, which substantially increases the difficulty of manipulation detection.

- **Fraud-Type Diversity and Granularity.** THEMIS systematically covers five challenging fraud methods (e.g., AI-Generated and Duplication) and introduces 16 fine-grained manipulation operations (e.g., scaling and color temperature). On average, each sample involves 2.08 stacked operations, producing highly diverse manipulations that require robust reasoning over composite alterations.

- **Multi-Dimensional Capability Evaluation.** To dissect model performance on these expert-level tasks, THEMIS establishes a principled mapping from fraud methods to five core reasoning capabilities that characterize expert-level visual forensics. **Expert Knowledge Utilization** evaluates whether models can incorporate prior domain knowledge to contextualize manipulations. **Visual Recognition** tests their ability to accurately perceive and distinguish complex visual elements. **Spatial Reasoning** requires understanding positional and structural relationships among manipulated components. **Region Localization** focuses on precisely identifying tampered areas at the sub-figure level. Finally, **Comparative Reasoning** assesses the ability to contrast multimodal evidence, such as cross-image or text–image consistency.

We evaluated 16 leading MLLMs on THEMIS, revealing three key findings: (1) a universal limitation in expert-level reasoning, with the state-of-the-art (SOTA) GPT-5 achieving only 56.15% overall performance; (2) a pronounced vulnerability to compound transformations, where GPT-5's F1 score on the **Duplication Identification** task plummets from 38% to 17% as the number of stacked manipulations increases from one to three; (3) imbalanced capability profiles across all models, exemplified by GPT-5's disparate performance on **Visual Recognition** (53.50%) versus **Region Localization** (24.14%). These findings collectively underscore the challenge of our benchmark and the critical reasoning gap in current MLLMs.

Table 1: **Comparison of THEMIS and other related benchmarks. Mod.**: Modality; **Manip.**: Manipulation Operations; **Real**: Contains real fraud cases; **I**: Image; **T-I**: Cross Text–Image; **Task**: **BC**: Binary Classification; **IP-L**: Image Partial Localization. **SMF**, **CMO**, and **CMI** denote Single-Mode Forgery, Composite Manipulation Operations, and Cross-Modal Inconsistency, respectively, with suffixes **-I** and **-L** indicating Identification and Localization (**-IL** for both). **Fraud Type**: ① Splicing, ② Copy-Move, ③ AI-Generated, ④ Duplication, and ⑤ Text–Image Inconsistency.

| Benchmark | Mod. | Fraud Type | # Manip. | Real | # QA Pairs | Task |
|---|---|---|---|---|---|---|
| FakeBench | I | ③ | 1 | ✘ | 54k | BC |
| DiffuSyn Bench | I | ③ | 7 | ✘ | 848 | BC |
| SHIELD | I | ①③ | 12 | ✔ | 1.5k | BC/CMO-I |
| MFC-Bench | T-I | ①③⑤ | 7 | ✔ | 35k | BC/CMI-I |
| MMFakeBench | T-I | ①③⑤ | 12 | ✔ | 11k | BC/CMI-I |
| Forensics-Bench | I/T-I | ①②③④⑤ | 15 | ✘ | 63k | BC/IP-L/CMI-IL |
| **THEMIS (ours)** | **I/T-I** | **①②③④⑤** | **16** | **✔** | **4k** | **SMF-IL/CMO-I/CMI-IL** |

## 2 RELATED WORK

### 2.1 VISUAL REASONING OF MLLMS

The advent of MLLMs has catalyzed a paradigm shift from basic visual perception to complex visual reasoning. Pioneering models like BLIP-2 (Li et al., 2023) bridged modalities via efficient alignment. Building on this, the LLaVA series (Liu et al., 2023; 2024) significantly advanced visual instruction tuning, while GPT-4o (OpenAI et al., 2024) enabled unified end-to-end multimodal reasoning. Recent developments such as the o1 family (OpenAI, 2025) demonstrate that integrating inference-time Chain-of-Thought (CoT) (Wei et al., 2022) further enhances reasoning depth.

**However, a critical blind spot persists in current evaluations.** While MLLMs excel on benchmarks requiring domain-specific knowledge (e.g., mathematics or chart analysis) (Yue et al., 2024; Lu et al., 2024; Wang et al., 2024b), their performance often hinges on the powerful textual reasoning of the underlying large language model (LLM) rather than on genuine visual acuity. In these tasks, visual content can frequently be abstracted into text descriptions, which the LLM then solves. Consequently, existing evaluations fail to decouple visual reasoning from this textual dependence. This limitation effectively obscures the true state of MLLMs' intrinsic visual capabilities, particularly in tasks like forensics where visual evidence cannot be losslessly converted into text.

### 2.2 VISUAL FRAUD REASONING BENCHMARK

Although benchmarks designed for traditional image forensics, such as CIMD (Zhang et al., 2024) and GIM (Chen et al., 2025), have established rigorous standards for specific detection tasks, they are ill-suited for assessing the holistic reasoning capabilities of MLLMs. However, as shown in Table 1, current benchmarks targeting MLLMs still fall short of expert-level standards. **The limitations of existing works are threefold**: (1) **Restricted Scope**: FakeBench (Li et al., 2025b), DiffuSyn Bench (Zhou & Hong, 2024), and SHIELD (Shi et al., 2025) cover only a narrow range of fraud types; (2) **Coarse Granularity**: MFC-Bench (Wang et al., 2025b) and MMFakeBench (Liu et al., 2025a) primarily focus on binary classification and identification tasks, lacking fine-grained, multi-dimensional evaluation metrics; (3) **Lack of Realism**: Forensics-Bench (Wang et al., 2025a) relies heavily on synthetic data while ignoring authentic cases, thereby failing to capture the complexity of real-world scenarios.

To bridge these gaps, we propose THEMIS. Addressing the lack of realism, THEMIS integrates 152 real-world forensic cases distilled from a rigorous screening of 1,432 retracted papers, alongside high-fidelity synthetic data to ensure scenario authenticity. Unlike prior works limited to coarse metrics, THEMIS establishes a principled mapping from diverse fraud types to five core reasoning capabilities. This enables a precise, fine-grained diagnosis of MLLMs' intrinsic visual forensic competencies, aligning task difficulty with the rigorous demands of expert forensics.

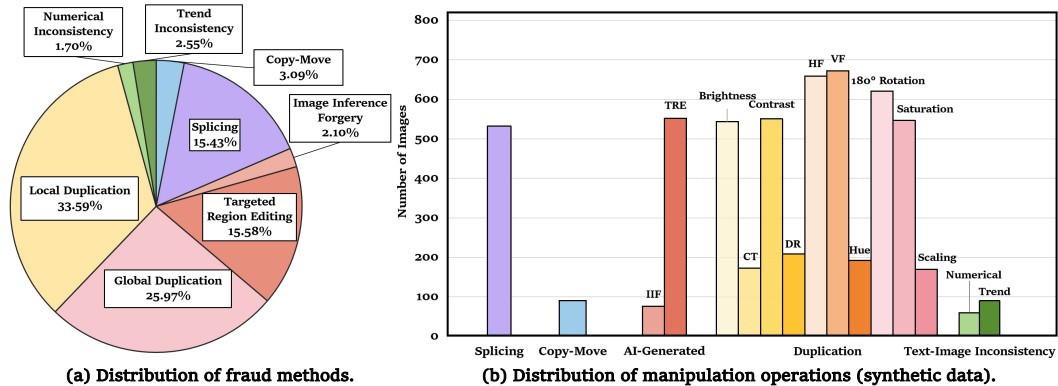

Figure 2: **Statistics of THEMIS. (a)** Distribution of fraud methods. **(b)** Distribution of manipulation operations (synthetic data). See Table 5 for real cases. **IIF**: Image Inference Forgery; **TRE**: Targeted Region Editing; **CT**: Color Temperature; **DR**: Direct Reuse; **HF**: Horizontal Flip; **VF**: Vertical Flip.

## 3 THEMIS BENCHMARK

### 3.1 OVERVIEW OF THEMIS

As illustrated in Figure 2, THEMIS constructs a comprehensive benchmark with a hierarchical forensic structure. Sourced from academic publications and retracted papers, the dataset undergoes detailed manual annotation and human–AI collaborative augmentation, covering five major fraud methods. Certain categories are further subdivided into secondary sub-types to support fine-grained visual fraud reasoning evaluation.

Building upon this taxonomy, we design three distinct tasks: **Single-Mode Forgery Identification and Localization**, **Composite Manipulation Operations Identification**, and **Cross-Modal Inconsistency Identification and Localization**. These questions fully encompass the aforementioned fraud methods. In total, 4,054 high-quality QA pairs are constructed, including 152 manually annotated samples derived from real-world academic retraction cases, which significantly enhances the reliability and annotation accuracy of the benchmark.

### 3.2 DATASET CONSTRUCTION

As shown in Figure 3, the dataset construction is divided into two stages: the first stage involves the extraction and parsing of source data, while the second stage focuses on fraud data generation and quality control for the five major fraud types.

**Stage 1: Extraction and Parsing.** We utilized two primary data sources for our benchmark, with summary statistics in Appendix A.1. (1) **Real Data**: We curated a gold-standard set of real fraud cases by collecting retracted papers from Retraction Watch[1] and PubPeer[2]. From this, experts extracted 194 authentic panels (defined as subgraph units with independent semantics within a larger figure). The collection and annotation process is described in Appendix A.2. (2) **Synthetic Data**: We gathered 5,253 research papers from PubMed Central[3]. After a rigorous selection, 879 high-quality papers were identified as source material. From this, we extracted 41,422 high-resolution panels. For cross-modal tasks, we curated 150 (`figure`, `caption`, `related sentence`) triplets, selecting one from each of 150 chosen papers. The six-step extraction and parsing pipeline, including YOLOv7-based (Wang et al., 2023) panel segmentation, is detailed in Appendix A.3.1.

**Stage 2: Fraud Data Generation.** To ensure a comprehensive evaluation of forensic capabilities, we simulate realistic academic paper fraud through a generation pipeline covering five major fraud data types. The detailed methodology and quality control for each type are described in Appendix A.3. The key strategies are summarized as follows:

---

[1] https://retractionwatch.com

[2] https://www.pubpeer.com

[3] https://pmc.ncbi.nlm.nih.gov

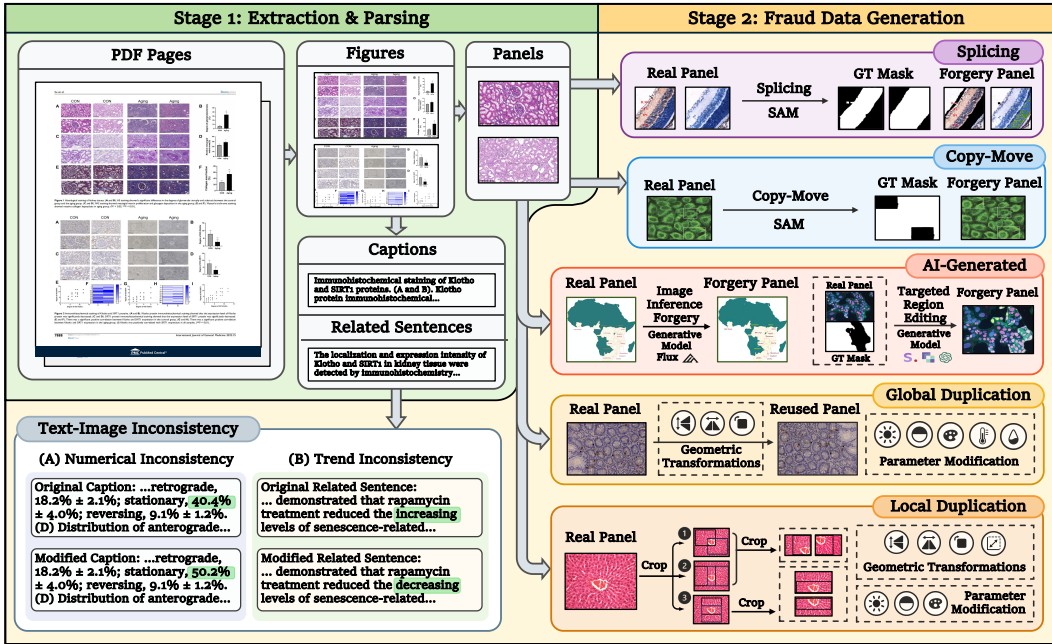

Figure 3: **Dataset construction pipeline of THEMIS.** The dataset is built through 2 stages: **Stage 1: Extraction and Parsing**, where figures, captions, and related sentences are parsed from scientific PDFs and segmented into panels; **Stage 2: Fraud Data Generation**, where 5 major fraud types (**Splicing**, **Copy-Move**, **AI-Generated**, **Duplication**, and **Text–Image Inconsistency**) are applied to construct challenging tasks.

- **Splicing.** We collected high-similarity panel pairs as the basis and created them by recombining the foreground and background of the panels, while employing bidirectional splicing and boundary fusion to ensure visual consistency and coherence.

- **Copy-Move.** We generated Copy-Move samples by replicating and translating specific objects within the same panel. During this process, we utilized the Segment Anything Model (SAM) (Kirillov et al., 2023) to perform automatic object extraction and mask optimization, and employed an adaptive grid positioning strategy to determine the placement of copied regions.

- **AI-Generated.** We constructed two types of AI-Generated panels: (1) **Image Inference Forgery**, which used the Flux (Labs et al., 2025) model to generate full-image forgeries; (2) **Targeted Region Editing**, which was based on text prompts or mask information and employs models such as Stable Diffusion (Esser et al., 2024), SenseNova V6 Miaohua (SenseTime, 2025) and GPT-Image-1 (OpenAI, 2025) for local editing.

- **Duplication.** We implemented this operation to simulate the reuse of panel content under various manipulations. We designed two distinct strategies to capture different scales of duplication: (1) **Global Duplication**, which targeted reuse of the entire image by performing direct reuse or applying geometric transformations and parameter modifications to the original panel to generate a manipulated duplicate; (2) **Local Duplication**, which simulated the subtler reuse of specific regions. This process began by extracting two panels with overlapping regions from the original panel. Then, employing a **crop-then-transform** strategy, we applied geometric transformations or parameter modifications to a specific area of one panel. Finally, both panels were reassembled onto a standardized canvas to form a visually coherent duplication pair with consistent dimensions.

- **Text–Image Inconsistency.** We constructed 150 inconsistent text–image pairs by altering captions or related sentences so that they no longer matched the image content. We divided the modifications into two categories: (1) **Numerical Inconsistency**, where we modified numerical values such as data and proportions; (2) **Trend Inconsistency**, where we replaced trend-related keywords with their antonyms.

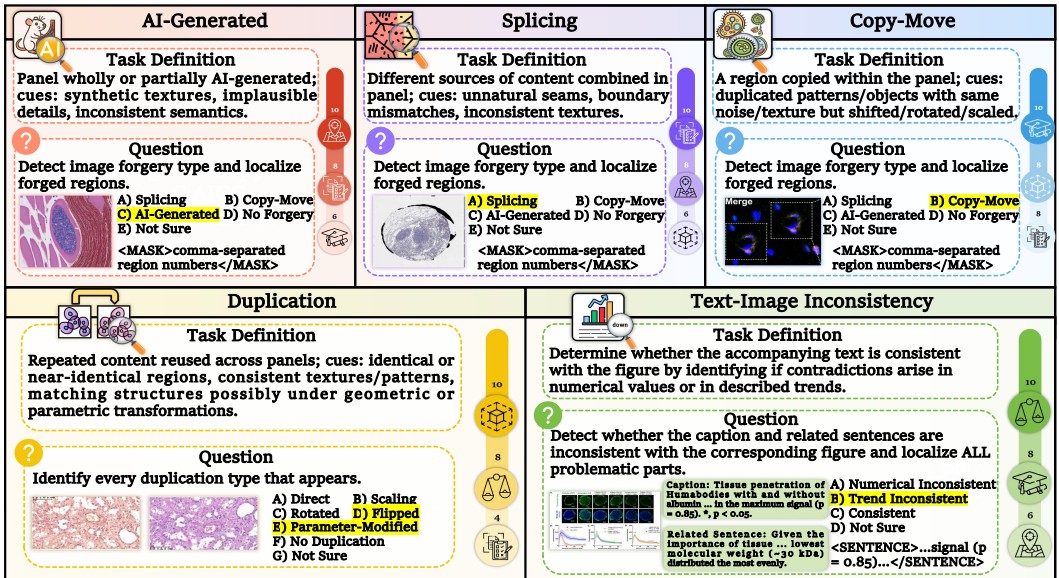

Figure 4: **Evaluation task design of THEMIS.** A principled mapping from 5 fraud types to 5 core reasoning capabilities (**Expert Knowledge Utilization**, **Visual Recognition**, **Spatial Reasoning**, **Region Localization**, and **Comparative Reasoning**). The capability distribution bars on the right of each box illustrate the reasoning skills involved and their relative emphasis, with the darkest color highlighting the primary capability being evaluated.

**Quality Control.** The operation involves over 16 academic review experts, supported by GPT-4o mini. The entire quality control process took approximately 200 hours, and about 20% of low-quality or unreasonable samples were removed from the original synthetic data. After this rigorous filtering, 4,635 positive samples were retained, alongside 1,540 negative samples without fraud.

## 3.3 Evaluation Task Design

As shown in Figure 4, to evaluate the comprehensive perception and reasoning capabilities of MLLMs across five fraud methods, we designed three core evaluation tasks. For each, the detailed definition, strict input/output format, and prompting instructions are provided in Appendix A.4.

**Task 1: Single-Mode Forgery Identification and Localization (SMF-IL).** This task corresponds to three fraud methods: **Splicing**, **Copy-Move**, and **AI-Generated**. The model must determine whether the input panel is *No Forgery*, *Not Sure*, or one of the three specific forgery types. If forged, the model must further provide block-based localization of the forged region. We evaluate identification with **Accuracy (ACC)** and localization with **Intersection over Union (IoU)**.

**Task 2: Composite Manipulation Operations Identification (CMO-I).** This task corresponds to **Duplication**. The model is required to analyze a pair of panels and discriminate the types of manipulation operations involved, selecting from seven potential categories: *Direct Reuse*, *Scaling*, *Rotation*, *Flip*, *Parameter Modification*, *No Duplication*, and *Not Sure*. Since multiple selections are permitted for a single pair, we employ the **Set-based F1 Score** for evaluation.

**Task 3: Cross-Modal Inconsistency Identification and Localization (CMI-IL).** This task corresponds to **Text–Image Inconsistency**. It requires the model to judge whether **Numerical** or **Trend** inconsistency exists based on the input image and its corresponding text. If detected, the model must further localize the minimum sentence unit with inconsistency in the text and provide corrected content. We evaluate identification with **ACC** and localization with **Text-based F1 Score**.

**Real Cases.** THEMIS includes 152 real-world forensic issues derived from a rigorous screening of 1,432 retracted papers sourced from Retraction Watch and PubPeer. Following strict inclusion criteria, 41 high-quality manuscripts were retained as the source corpus. These issues predominantly involve three major fraud types: **Splicing**, **Copy-Move**, and **Duplication**. To ensure data reliability,

we employed a multi-annotator collaborative protocol where each paper was independently reviewed by three experts in academic image forensics. Consequently, this subset represents a curated gold standard of real academic fraud cases to support fine-grained reasoning.

# 4 EXPERIMENTS

## 4.1 EXPERIMENTAL SETUP

**Benchmarked MLLMs.** We evaluate nine proprietary models (Singh et al., 2025; OpenAI, 2025; Comanici et al., 2025; Bai et al., 2023; Team, 2025; Anthropic, 2025; Team et al., 2026) and seven open-source models (Meta, 2025; Bai et al., 2025; Team et al., 2025; Li et al., 2025a; Liu et al., 2024; Wang et al., 2025c) covering mainstream industrial progress and diverse open-weight architectures.

**Input Processing.** We use high-resolution PNG images whenever possible. If an input exceeds a model's limit, a two-stage adaptive compression is applied: (1) scale the long edge to each model's maximum input size while keeping aspect ratio; (2) further reduce total pixels to satisfy model constraints, with iterative 10% reduction if necessary.

**Evaluation Metrics.** We adopt two primary metric categories: **Identification Score (Id Score)** and **Localization Score (Loc Score)**, applied according to the diverse visual fraud reasoning tasks.

- **Single-Mode Forgery Identification and Localization.** We use **ACC** as the Id Score and **IoU** as the Loc Score.
- **Composite Manipulation Operations Identification.** We employ the **Set-based F1 Score** as the Id Score, since multiple selections are permitted.
- **Cross-Modal Inconsistency Identification and Localization.** We use **ACC** as the Id Score and **Text-based F1 Score** as the Loc Score, reflecting precise localization of inconsistent textual units.

We further introduce a **Balanced Robustness Index (BRI)**, which adjusts mean performance by penalizing large variance across tasks. The formal mathematical formulations for all metrics, alongside the detailed derivation and sensitivity analysis of the BRI, are provided in Appendix B.3.

Table 2: **Model performance across tasks on THEMIS synthetic data.** We report the **Id Score** (Identification Score) and **Loc Score** (Localization Score) for 5 fraud methods: **SPL** (Splicing), **CM** (Copy-Move), **AIG** (AI-Generated), **DUP** (Duplication), and **TII** (Text–Image Inconsistency), with **BRI** (Balanced Robustness Index). The **best** and second-best scores are highlighted.

| Model | Single-Mode Forgery Identification (Id Score) | | | | Single-Mode Forgery Localization (Loc Score) | | | | Composite Manipulation Operations Identification (Id Score) | Cross-Modal Inconsistency Identification & Localization | | BRI |
|---|---|---|---|---|---|---|---|---|---|---|---|---|
| | SPL (807) | CM (242) | AIG (897) | Avg. | SPL (807) | CM (242) | AIG (897) | Avg. | DUP (2,079) | TII (300) Id Score | Loc Score | |
| *Proprietary MLLMs* | | | | | | | | | | | | |
| GPT-5 | 43.51 | 72.73 | 44.26 | 53.50 | 16.67 | 36.41 | 19.33 | 24.14 | 33.32 | 60.67 | 27.44 | **56.15** |
| OpenAI o4-mini-high | 41.49 | 77.89 | 35.67 | 51.68 | 10.44 | 29.78 | 19.79 | 20.00 | 30.34 | **66.33** | 32.22 | 52.34 |
| Qwen-VL-Max | 30.37 | **87.40** | 35.43 | 51.07 | 40.07 | 48.34 | 35.85 | 41.42 | 23.33 | 56.00 | 15.36 | 49.83 |
| Gemini 2.5 Flash | **63.39** | 67.56 | 35.45 | **55.47** | **56.72** | 46.98 | 38.87 | **47.52** | 24.96 | 36.33 | 28.24 | 44.70 |
| Doubao-Seed-1.6-thinking | 35.67 | 74.17 | 36.57 | 48.80 | 12.09 | 13.58 | 15.19 | 13.62 | 20.22 | 60.00 | 31.71 | 37.14 |
| Doubao-Seed-1.6-vision | 20.84 | 61.78 | 27.54 | 36.72 | 31.42 | 26.97 | 29.90 | 29.43 | **45.13** | 37.00 | 30.43 | 33.47 |
| Gemini 2.5 Pro | 30.60 | 47.46 | 14.90 | 30.99 | 46.65 | 46.61 | **47.20** | 46.82 | 21.49 | 44.67 | **37.31** | 31.97 |
| GLM-4.5V | 29.23 | 58.43 | 54.51 | 47.39 | 8.54 | 22.23 | 10.63 | 13.80 | 21.84 | 53.67 | 22.69 | 31.57 |
| Claude Sonnet 4.5 | 29.71 | 75.66 | 30.43 | 45.27 | 14.95 | 44.12 | 20.98 | 26.68 | 21.86 | 35.67 | 27.42 | 29.96 |
| *Open-Source MLLMs* | | | | | | | | | | | | |
| Qwen2.5-VL-72B | 36.40 | 77.27 | 51.06 | 54.91 | 51.66 | 55.16 | 35.57 | 47.46 | 16.75 | 61.33 | 12.32 | 47.16 |
| InternVL3.5-8B | 30.97 | 58.42 | 67.00 | 52.13 | 43.38 | 40.24 | 33.19 | 38.94 | 33.28 | 55.00 | 4.78 | 38.73 |
| Llama 4 Maverick | 23.37 | 51.24 | 58.19 | 44.27 | 17.97 | 28.50 | 19.71 | 22.06 | 19.01 | 54.00 | 16.08 | 34.78 |
| LLaVA-Interleave-7B-DPO | 41.80 | 47.55 | 46.93 | 45.43 | 35.97 | 25.22 | 32.06 | 31.08 | 8.01 | 50.00 | 12.57 | 23.59 |
| LLaVA-NeXT-34B | 32.00 | 84.00 | 34.00 | 50.00 | 50.70 | 45.25 | 34.01 | 43.32 | 10.73 | 41.33 | 4.28 | 18.40 |
| Qwen2.5-VL-32B | 30.31 | 57.48 | 39.24 | 42.34 | 5.91 | 18.93 | 7.45 | 10.76 | 15.26 | 58.33 | 17.94 | 18.22 |
| Gemma 3 27B | 25.39 | 28.87 | 34.23 | 29.50 | 27.78 | 32.80 | 26.44 | 29.01 | 12.20 | 31.67 | 21.41 | 9.59 |

## 4.2 MAIN RESULTS

Table 2 presents the evaluation results across all models on synthetic data. Our main findings are:

**Overall performance remains limited.** Across all evaluated tasks, MLLMs exhibit significant room for improvement, with even the SOTA GPT-5 reaching a peak BRI of only 56.15%. Notably, the open-source Qwen2.5-VL-72B achieved 47.16%, performing on par with several proprietary models and highlighting the competitive potential of open-weight architectures.

**Models exhibit pronounced specialization.** Most MLLMs excel in one or two subtasks but fall short in others, revealing substantial imbalance. This specialization highlights the lack of integrated visual fraud reasoning capabilities in current models.

**Localization is markedly harder than identification.** Performance consistently declined across all MLLMs when shifting from **Single-Mode Forgery Identification** task to **Localization** task, with GPT-5 dropping by 55% and OpenAI o4-mini-high by 61%. By contrast, Gemini 2.5 Flash exhibited only minor declines (14%), indicating relatively stronger spatial perceptual capacity.

**Limitations in fine-grained cross-modal alignment.** Although they demonstrate relatively high judgment accuracy on **Text–Image Inconsistency** subtask, they struggle to ground these judgments in specific textual spans. This limitation may stem from an overreliance on global semantic associations while lacking sufficient modeling of local cross-modal mappings.

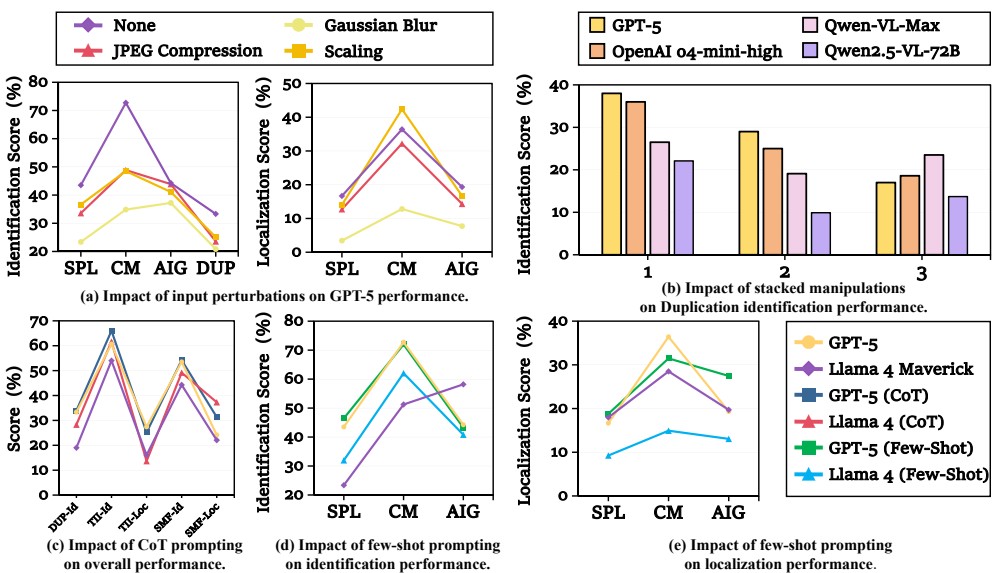

Figure 5: **Impact of different factors on model performance using synthetic data. SPL**: Splicing; **CM**: Copy-Move; **AIG**: AI-Generated; **DUP**: Duplication; **DUP-Id**: Id Score for Duplication; **TII-Id**: Id Score for Text–Image Inconsistency; **TII-Loc**: Loc Score for Text–Image Inconsistency; **SMF-Id**: Average Id Score of Single-Mode Forgery Identification; **SMF-Loc**: Average Loc Score of Single-Mode Forgery Localization.

## 4.3 FINE-GRAINED ANALYSIS

Tables 2–3 and Figure 5 present the fine-grained results from our further analysis. For a detailed quantitative analysis, we provide two sets of supplementary results in the appendix: a breakdown of performance on real versus synthetic data and under input perturbations in Appendix C, and a per-fraud-type performance analysis for all models in Appendix D. The key findings are as follows:

**Lack of transformation sensitivity.** Models perform reasonably on direct reuse but degrade markedly under geometric transformations or appearance adjustments. They often fail to detect duplication and cannot determine the transformation type, underscoring the limited spatial reasoning ability of current MLLMs and their insufficient robustness to transformations.

This reveals that current models have basic similarity-matching ability but remain weak in edge sensitivity.

**Current models lack robustness to input perturbations.** To test MLLM sensitivity on synthetic data, we applied Gaussian blur, JPEG compression, and scaling. Gaussian blur caused the steepest drops across tasks, while JPEG compression and scaling also degraded performance.

**Insufficient edge perception.** Models perform better on **Copy-Move** than **Splicing**, as the former can exploit both edge anomalies and region similarity, while the latter depends mainly on boundary cues.

Table 3: **Performance comparison on real cases.** We report the results for Single-Mode Forgery Identification and Localization (**SMF-IL**) and Composite Manipulation Operations Identification (**CMO-I**) tasks.

| Model | SMF-IL | | CMO-I |
|---|---|---|---|
| | Id Score | Loc Score | Id Score |
| Qwen2.5-VL-72B | 50.00 | 45.41 | 22.39 |
| Gemini 2.5 Flash | 44.55 | 43.07 | 49.58 |
| GPT-5 | 43.64 | 28.68 | 23.22 |
| OpenAI o4-mini-high | 34.55 | 21.67 | 36.04 |
| Gemma 3 27B | 20.00 | 16.15 | 42.18 |
| Qwen-VL-Max | 6.67 | 9.43 | 20.95 |
| Llama 4 Maverick | 3.34 | 5.44 | 39.75 |

**Synthetic fraud data exhibit a level of deceptiveness comparable to real fraud data.** In the **Composite Manipulation Operations Identification** task, the synthetic data impose even greater identification pressure than real data. In the **Single-Mode Forgery Identification and Localization** task, their difficulty is similarly on par with real data. The main exception lies in **Splicing** subtask, where real data remain more challenging due to the sophistication and subtlety of splicing patterns in real-world manipulations.

## 4.4 ERROR ANALYSIS

We conducted an error analysis on the best-performing model, GPT-5. Specifically, we sampled 20 failure cases for each of the five core fraud types, resulting in a total of 100 erroneous instances. These errors are then mapped to the five core reasoning capabilities. Notably, a single case may involve multiple deficiencies, leading to overlapping classifications. For a detailed qualitative analysis, Appendix E visualizes 22 representative cases covering the full spectrum of seven academic scenarios and five fraud types.

- **Expert Knowledge Utilization (43/100)**: The model often fails to leverage prior domain knowledge or recognize specific manipulation patterns, resulting in a lack of contextualized reasoning.

- **Visual Recognition (37/100)**: When confronted with high texture complexity or fine-grained manipulations, the model exhibits insufficient perceptual capacity, leading to perception-level errors.

- **Spatial Reasoning (19/100)**: The model struggles in cases requiring spatial reasoning, failing to correctly infer positional and geometric relationships among components.

- **Region Localization (25/100)**: Predicted fraud regions frequently appear blurred or spatially misaligned, reflecting unreliable fraud localization performance.

- **Comparative Reasoning (21/100)**: The model often overlooks subtle differences or fails to establish effective cross-modal comparisons, thereby producing erroneous conclusions.

## 4.5 PROMPTING STRATEGIES

In our benchmark, we evaluate two prompting strategies on synthetic data. Few-shot (Brown et al., 2020) provides only limited and inconsistent benefits, whereas CoT (Wei et al., 2022) consistently enhances reasoning performance. Detailed performance breakdowns across specific fraud types, along with the full CoT and few-shot prompt templates, are provided in Appendix C.2.

**Few-shot prompting yields mixed effects on forensic reasoning.** While it brings moderate gains in certain subtasks, such as **Copy-Move** and **AI-Generated**, performance often declines in other subtasks, and overall improvements are inconsistent. We hypothesize that few-shot examples may constrain models to imitate local patterns in the demonstrations rather than generalize, limiting their ability to adapt to diverse fraud types.

**CoT prompting proves effective for deeper forensic reasoning.** In our benchmark, GPT-5 (the strongest model) and Llama 4 Maverick (a weaker baseline) both benefit from CoT, showing consistent improvements. By enforcing stepwise analysis of visual cues and decision options, CoT enables models to move beyond surface-level correlations in scientific figures. Notably, Llama 4 Maverick benefits more from CoT than GPT-5, suggesting that models with weaker native reasoning abilities benefit disproportionately from stepwise prompting.

## 5 CONCLUSION

In this paper, we introduce THEMIS, a novel benchmark designed to evaluate the expert-level visual reasoning capabilities of MLLMs. By integrating **real-world forgery scenarios**, **diverse, fine-grained fraud methods**, and a **multi-dimensional capability evaluation** within a unified framework, THEMIS systematically probes the performance of MLLMs on complex academic multimodal fraud detection. Our extensive evaluation of 16 leading MLLMs reveals significant limitations in their current abilities, with even SOTA models performing poorly on compound manipulations and exhibiting highly imbalanced capability profiles across different reasoning dimensions. We hope this work will inspire future research into more robust and diagnostic evaluation paradigms for MLLMs.

### BROADER IMPACT

The rapid evolution of MLLMs presents a dual-edged sword to the scientific community. While these technologies accelerate research, they simultaneously lower the barrier for fabricating sophisticated scientific data, enabling new forms of misconduct—such as AI-generated imagery and text—that are increasingly difficult to distinguish from authentic work.

Current defense mechanisms, which rely heavily on manual inspection by editorial boards or specialized models with limited scope, are becoming inadequate against these emerging, scalable AI-generated content (AIGC) threats. The relatively low number of publicly exposed retraction cases likely reflects a detection gap rather than a rarity of misconduct, as the community currently lacks the capacity for large-scale, automated, and precise verification.

Our work addresses this critical asymmetry by advocating for an **AI for AI Governance** approach. By introducing THEMIS, we provide the first comprehensive benchmark that simulates the complexity of real-world academic fraud, ranging from subtle pixel-level tampering to logical inconsistencies in generated content. This benchmark serves as a foundational testbed for developing trustworthy multimodal systems capable of automated forensic reasoning. We hope THEMIS will catalyze the development of high-throughput forensic agents that can assist human experts, thereby safeguarding scientific integrity and restoring trust in the era of generative AI.

### ETHICS STATEMENT

This work adheres to the ICLR Code of Ethics. In this study, no human subjects or animal experiments were involved. All datasets used were sourced in compliance with relevant usage guidelines, ensuring no violation of privacy. We have taken care to avoid any biases or discriminatory outcomes in our research process. No personally identifiable information was used and no experiments were conducted that could raise privacy or security concerns. We are committed to maintaining transparency and integrity throughout the research process.

### REPRODUCIBILITY STATEMENT

We are committed to making our research fully reproducible. All code, datasets, and evaluation scripts have been publicly released on our Project Page, GitHub repository, and Hugging Face. The experimental setup, including detailed model configurations and hardware specifications, is provided in Section 4.1 and Appendix B. We also provide a comprehensive description of THEMIS to facilitate future research and benchmarking in academic fraud forensics. We believe these resources will enable other researchers to replicate our findings and further advance the field of multimodal reasoning.

ACKNOWLEDGMENTS

This work is supported by the National Natural Science Foundation of China (Grant Nos. 62473271, 62176026), the Beijing Natural Science Foundation (Grant No. QY25338), the Fundamental Research Funds for the Beijing University of Posts and Telecommunications (Grant No. 2025AI4S03), and the BUPT Innovation and Entrepreneurship Support Program (Grant No. 2025-YC-A042). This work is also supported by the Engineering Research Center of Information Networks, Ministry of Education, China. We would also like to thank the anonymous reviewers and area chairs for constructive discussions and feedback.

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

# Appendix Contents

## A   BENCHMARK CONSTRUCTION AND TASK DESIGN

### A.1   DATASET STATISTICS

#### A.1.1   BASIC STATISTICS

Table 4 summarizes the fundamental statistics of THEMIS, illustrating the scale of the dataset and the complexity of the manipulation operations (e.g., average number of operations per synthetic image). Note that the paper categories are not mutually exclusive, as an individual source paper may provide both original (authentic) samples and their manipulated (synthetic) counterparts. Complementing this, Table 5 provides a detailed breakdown of the authentic retracted papers, highlighting the distribution of specific real-world forgery techniques involved in our benchmark.

Table 4: Basic statistics of THEMIS.

| Statistics | Value |
|---|---|
| # Total Papers | 920 |
| # Total Images | 6,025 |
| # Papers Providing Authentic Samples | 483 |
| # Papers Providing Synthetic Samples | 744 |
| # Papers with Real Issues | 41 |
| # Inter-Element Images | 5,145 |
| # Manip. per Synthetic Image (Avg.) | 2.08 |
| # Images per Scenario (Avg.) | 860.71 |
| # Segments per Mask (Avg.) | 4.84 |

Table 5: Basic statistics of retracted papers.

| Statistics | Value |
|---|---|
| # Retracted Papers | 41 |
| # Total Panels in Retracted Papers | 194 |
| # Splicing Samples | 11 |
| # Copy-Move Samples | 17 |
| # Direct Reuse Samples | 74 |
| # Rotation Samples | 28 |
| # Flip Samples | 9 |
| # Scaling Samples | 8 |
| # Parameter Modification Samples | 11 |

#### A.1.2   PAPER SOURCING AND CURATION

We first collected 1,432 retracted papers from the open-source repositories Retraction Watch and PubPeer. To ensure the scientific validity of our fraud-related dataset, multiple annotators conducted a rigorous manual auditing pipeline. Specifically: (1) we examined retraction notices and excluded cases unrelated to scientific paper fraud (e.g., voluntary withdrawals, journal mergers, and publisher-side issues); (2) we carefully inspected image-related evidence and discarded cases that only involved minor typographical or formatting errors; and (3) we adopted a cross-review mechanism, where each retained paper was independently verified by at least two annotators. Following this process, only 41 papers were ultimately confirmed to strictly match our definition of fraud types.

Beyond the retracted papers, we also adopted a rigorous multi-stage screening process to curate the broader source corpus, which was initially sourced from the PubMed Central open-access repository, for synthetic data construction. Concretely, we (1) excluded all non-open-access or incomplete articles, (2) filtered out papers with missing figures, low-resolution images, or ambiguous captions, and (3) performed manual inspection by multiple annotators to retain only those satisfying the requirements of clarity, integrity, and domain relevance. After this multi-stage filtering, we selected 879 high-quality papers from an initial pool of 5,253, forming the foundation of our synthetic dataset.

#### A.1.3   DISTRIBUTION AND VISUALIZATION OF SEVEN SCENARIOS

In order to ensure comprehensive coverage of visual contexts in scientific papers, we categorized all images into seven representative scenarios: **Chart**, **Diagram**, **Micrograph**, **Stained Micrograph**, **Physical Object**, **Medical Imaging**, and **Others**. The Others scenario further included equations, data tables, covers, UI screenshots, legends, Western blots, and multi-panel compositions, thereby complementing the main scenarios to span the full spectrum of scientific visual elements. To obtain reliable annotations, we adopted a semi-automatic pipeline in which Doubao-Seed-1.6 (Team, 2025) was employed to provide preliminary classification, followed by careful refinement and verification by more than three human experts. As shown in Figure 6, the final distribution of annotated samples is presented as follows: Chart (1,403), Diagram (286), Micrograph (854), Stained Micrograph (2,490), Physical Object (191), Medical Imaging (220), and Others (581).

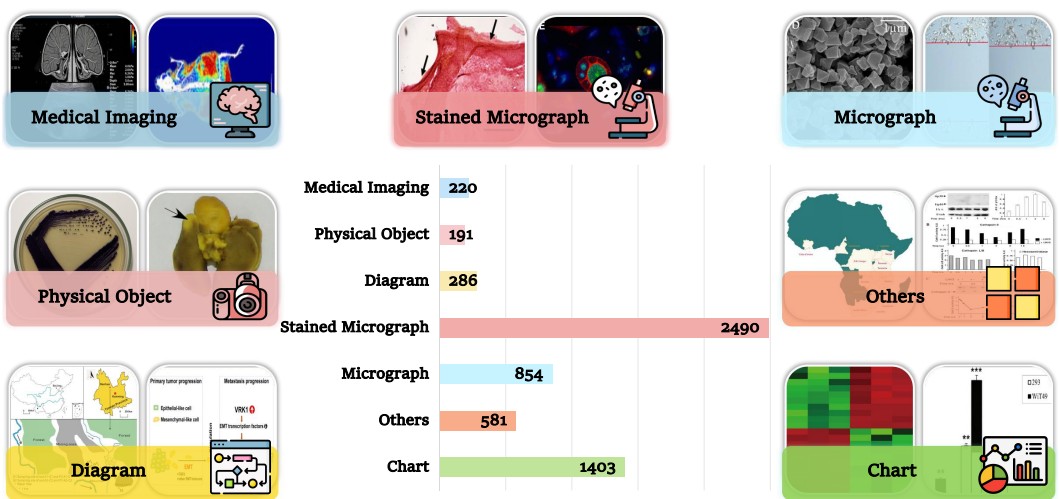

Figure 6: Distribution and visualization of 7 scenarios.

### A.1.4 DISTRIBUTION OF FRAUD TYPES

Table 6 details the fine-grained distribution of fraud types and manipulation operations specifically for the synthetic portion of our dataset. Because real-world retraction notices rarely provide fine-grained documentation of the exact manipulation operations (e.g., scaling or color temperature), the 41 authentic retracted papers are tallied separately based on their macro-level fraud methods, as previously detailed in Table 5.

Table 6: Detailed distribution of fraud types and manipulation operations across the synthetic data.

| Scope | Fraud Type | Manipulation Operation | # Samples |
|---|---|---|---|
| Intra-Element | Copy-Move | Copy-Move | 92 |
| | AI-Generated | Image Inference Forgery | 74 |
| | | Targeted Region Editing | 550 |
| Inter-Element | Splicing | Splicing | 534 |
| | Duplication | Global Direct Reuse | 104 |
| | | Global 180° Rotation | 283 |
| | | Global Vertical Flip | 315 |
| | | Global Horizontal Flip | 289 |
| | | Global Brightness | 184 |
| | | Global Contrast | 185 |
| | | Global Saturation | 203 |
| | | Global Hue | 190 |
| | | Global Color Temperature | 171 |
| | | Local Direct Reuse | 103 |
| | | Local 180° Rotation | 336 |
| | | Local Scaling | 168 |
| | | Local Vertical Flip | 356 |
| | | Local Horizontal Flip | 368 |
| | | Local Brightness | 358 |
| | | Local Contrast | 364 |
| | | Local Saturation | 342 |
| | Text–Image Inconsistency | Numerical Inconsistency | 60 |
| | | Trend Inconsistency | 90 |

## A.2 REAL DATA CONSTRUCTION

**Data Collection.** We constructed our real-data subset by systematically collecting 1,432 retracted papers from the open-source repositories Retraction Watch and PubPeer. To ensure coverage and diversity, we first aggregated metadata (`title`, `authors`, `journal`, `retraction reason`) and then obtained the full-text PDFs where available. Since not all retracted cases involve visual fraud, we applied preliminary filtering to remove non-open-access articles, incomplete manuscripts, and papers without figures. Following this process, a total of 41 high-quality papers were retained from an initial pool of candidates, providing the source corpus for subsequent annotation and validation.

**Dataset Annotation.** To precisely capture fraudulent manipulations in retracted papers, we adopted a multi-annotator collaborative annotation protocol. Each paper was reviewed independently by at least three annotators with expertise in biomedical image forensics and scholarly integrity. The annotation primarily focused on panel-level issues, such as Copy-Move and Splicing. Annotators marked tampered regions using bounding boxes and masks, and additionally recorded the issue type, subtype, and supporting evidence in a structured JSON schema. To facilitate fine-grained reasoning tasks, each annotation was further linked to metadata such as caption text, page number, and the figure–panel hierarchy.

**Quality Control.** Ensuring the reliability of annotations is critical. We implemented a three-stage validation pipeline: (1) cross-validation among annotators, requiring consensus between at least two annotators for an issue to be retained; (2) expert arbitration, where disputed cases were resolved by senior domain experts; and (3) consistency checks, including overlap verification between bounding boxes and masks, and completeness of metadata. In addition, a random sample of 10% of the annotated papers underwent blind re-annotation, yielding an inter-annotator agreement score above 0.87, demonstrating the robustness and reproducibility of the dataset. Ultimately, from the 1,432 retracted papers, only 41 papers were confirmed to strictly match our definition of fraud types, forming a curated gold-standard subset of real fraudulent cases.

## A.3 SYNTHETIC DATA CONSTRUCTION

### A.3.1 EXTRACTION AND PARSING

We collected 5,253 papers from the PubMed Central open-access repository and performed a preliminary expert screening to remove samples with irregular layouts, poor image quality, or lacking experimental visual content, ultimately retaining 879 high-quality papers as the core data source. Based on the original PDF files, we designed a systematic extraction and parsing pipeline to construct a standardized and traceable annotation framework. As shown in Figure 7, the entire process consists of six steps:

**Step 1: Figure and Textual Content Extraction.** Each PDF was parsed page by page using the Fitz[4] library to extract high-resolution figure composites. In parallel, GPT-4o mini was employed to assist human experts in parsing and extracting figure-related captions and associated sentences, which were then cross-checked and corrected by at least three human annotators.

**Step 2: Panel Segmentation.** We customized YOLOv7 (Wang et al., 2023) with task-specific modifications, including anchor adjustments, a confidence threshold of 0.5 (balancing quality and recall), and an overlap threshold of 0.35 (suitable for multi-panel figures). The detector was trained on our in-house annotations to generate dedicated weights for panel-level segmentation, thereby enabling fine-grained subdivision of figure composites.

**Step 3: Coordinate Mapping.** A transformation function was applied to convert figure coordinates back into page coordinates, and all results were consistently recorded as pixel-level bounding boxes (`bbox_page_pixel`) to preserve semantic layout consistency.

**Step 4: Ordering and Renaming.** All panels were systematically ordered and renamed following a top-to-bottom and left-to-right reading logic to ensure interpretability.

---

[4]`https://pymupdftest.readthedocs.io/en/stable/module.html`

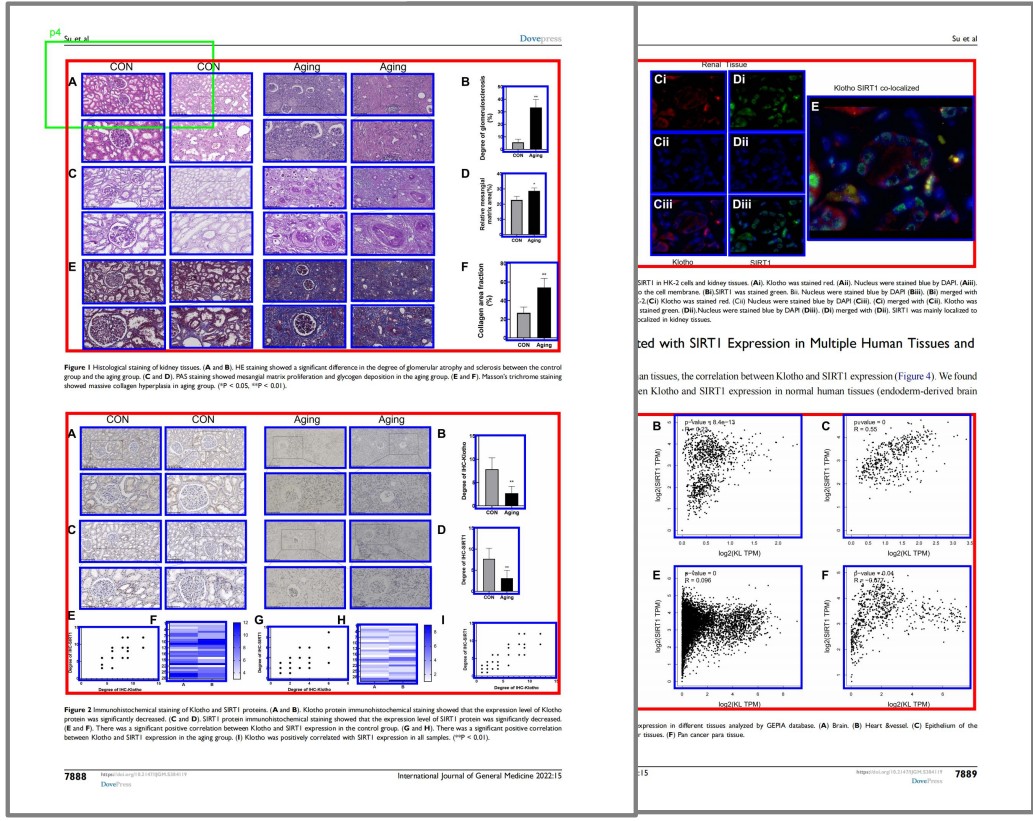

Figure 7: **Stage 1: Extraction and Parsing.** The regions highlighted with red boxes correspond to **figures**, while those highlighted with blue boxes correspond to **panels**. Green boxes are used to verify whether the visualization is effective.

**Step 5: Visualization and Verification.** For each paper, visualization files with bounding box overlays were generated and manually inspected to validate the accuracy of segmentation and coordinate mapping.

**Step 6: Structured Archiving.** All parsing outputs were consolidated into a hierarchical structure spanning `paper-page-figure-panel`, which encapsulates file paths, coordinates, identifiers, and potential issue flags to facilitate downstream analysis.

### A.3.2 SPLICING FORGERIES

**Generation Pipeline.** The Splicing pipeline is designed to generate panel-level synthetic forgeries that remain visually plausible and covert. To ensure that the manipulation strategy is both reasonable and effective, we systematically identified panel pairs exhibiting the highest degree of visual similarity across multiple dimensions. All candidate panels were first resized to a standard resolution of $224 \times 224$ pixels and converted to grayscale, which reduces the influence of color bias and ensures consistent feature extraction across the dataset.

To evaluate similarity, we adopted a multi-dimensional metric that integrates three complementary features:

(1) **Histogram Correlation (40%).** We computed 256-bin grayscale histograms with `cv2.calcHist()` and measured correlation coefficients between histograms with `cv2.compareHist()`. This captures distributional similarities in pixel intensities, reflecting color and brightness patterns across images. A weight of $0.4$ was assigned to emphasize the importance of distribution-level consistency.

(2) **Structural Similarity Index (SSIM, 40%).** Following the simplified SSIM formulation, we computed the mean values $(\mu_1, \mu_2)$, variances $(\sigma_1^2, \sigma_2^2)$, and covariance $(\sigma_{12})$ of paired images, incorporating stability constants $c_1 = 0.01^2$ and $c_2 = 0.03^2$. SSIM quantifies structural fidelity by comparing luminance, contrast, and structural information, thereby ensuring that the internal organization of panels is aligned. With a weight of $0.4$, this component plays a dominant role in guiding similarity assessment.

(3) **Size Similarity (20%).** We measured geometric compatibility by comparing the relative differences in height and width between two panels. Specifically, similarity is defined as

$$S_{size} = \left(1 - \frac{|h_1 - h_2|}{\max(h_1, h_2)}\right) \times \left(1 - \frac{|w_1 - w_2|}{\max(w_1, w_2)}\right), \tag{1}$$

where $h_1, h_2$ denote heights and $w_1, w_2$ denote widths. This component, weighted at $0.2$, prevents severe scale mismatches while retaining moderate variability.

The overall similarity score is computed as a weighted sum of the three components:

$$S_{total} = \omega_1 \times S_{hist,norm} + \omega_2 \times \max(S_{ssim}, 0) + \omega_3 \times S_{size}, \tag{2}$$

where $S_{hist,norm} = \frac{S_{hist}+1}{2}$ normalizes histogram correlation into $[0, 1]$, $\omega_1 = 0.4$, $\omega_2 = 0.4$, and $\omega_3 = 0.2$. The final score $S_{total}$ ranges between $0$ and $1$, with values closer to $1$ indicating higher similarity between panels.

Through exhaustive search over all possible panel pairs, the method selects those with the highest similarity scores as candidates for splicing. This multi-dimensional assessment guarantees that paired panels share consistent distributional, structural, and geometric properties, thereby providing a strong basis for visually convincing splicing forgeries.

For each selected pair (panel A, panel B), foreground objects were extracted from the source panel using a combination of thresholding, morphological refinement, and connected-component analysis. To ensure accurate object segmentation, we adopted the following procedure:

(1) Otsu's thresholding algorithm was applied with the `cv2.threshold()` function using the `cv2.THRESH_BINARY + cv2.THRESH_OTSU` flag, which automatically determines the optimal threshold to convert the grayscale panel into a binary mask, effectively separating foreground objects from the background.

(2) Morphological operations were performed with a $5 \times 5$ kernel via the `cv2.morphologyEx()` function: a closing operation (`cv2.MORPH_CLOSE`) was applied to fill internal holes within objects, followed by an opening operation (`cv2.MORPH_OPEN`) to suppress small noise points, thereby improving segmentation quality and object regularity.

(3) Connected-component analysis used `cv2.connectedComponentsWithStats()` to label all contiguous regions in the binary mask and compute statistics such as area via `cv2.CC_STAT_AREA`. The region with the maximum area was selected as the primary object to ensure completeness of the extracted foreground.

The resulting object mask was post-processed (hole filling and removal of minor components), resized, and geometrically aligned to match the dimensions of the target background panel, providing a consistent and precise object mask for subsequent splicing synthesis.

Pixel-level composition was performed via alpha blending to ensure that the extracted foreground object could be seamlessly integrated into the background of the target panel. Specifically, the binary object mask was first normalized to the range $[0, 1]$ and then used as the alpha channel for weighted pixel-wise fusion. The blending operation follows the equation:

$$I_{syn}(x,y) = I_{obj}(x,y) \times M_{norm}(x,y) + I_{bg}(x,y) \times \big(1 - M_{norm}(x,y)\big), \tag{3}$$

where $I_{syn}(x,y)$ denotes the synthetic image pixel at position $(x,y)$, $I_{obj}(x,y)$ is the foreground object pixel, $I_{bg}(x,y)$ is the background pixel, and $M_{norm}(x,y)$ is the normalized mask value at $(x,y)$, computed as $M_{norm}(x,y) = M(x,y)/\max(M)$. The complement $(1 - M_{norm}(x,y))$ represents the background mask, ensuring that the blending weights always sum to one, thereby preventing unnatural brightness artifacts at the splice boundary.

To guarantee dimensional consistency, all images and masks were resized to the same spatial resolution using `cv2.resize()` with the `cv2.INTER_NEAREST` interpolation method to preserve

the discreteness of the binary mask. The single-channel mask was expanded into a three-channel format via `np.stack()` to match the RGB image representation. This process ensured that both the object and background were aligned at the pixel level, which facilitated precise and artifact-free composition.

To increase data diversity, two variants were synthesized for each panel pair: (1) the object from panel A composited onto the background of panel B, and (2) the object from panel B composited onto the background of panel A. Each synthesis produced a structured triplet (`original source panel, synthetic panel, object mask`), along with metadata including panel identifiers, similarity scores, object-to-background area ratio, and transformation parameters.

**Quality Control.** To guarantee both the quality of the generated Splicing data and its practical utility for the Single-Mode Forgery Identification and Localization task, we adopted a rigorous quality control and filtering protocol. Specifically, we identified and removed visually meaningless manipulations, such as cases where the inserted object was entirely incompatible with the target background or where the object-to-background size ratio was severely distorted. In addition, we excluded samples exhibiting overly obvious artifacts, including unnatural seams at the splice boundary, abrupt color transitions, or visible traces of post-processing. The filtering process integrated multiple criteria, including similarity score thresholds, object area ratio constraints, and post-composition quality evaluation, thereby ensuring that the retained Splicing samples were both visually plausible and sufficiently challenging. This guarantees that the final dataset provides high-quality and well-structured samples to support scientific paper fraud detection research. After these steps, a total of 534 high-quality Splicing samples were retained for downstream analysis.

### A.3.3 COPY-MOVE FORGERIES

**Generation Pipeline.** This pipeline was developed to construct panel-level Copy-Move forgeries, simulating the common pattern of duplicating regions within scientific images. We employed the SAM (Kirillov et al., 2023) to perform automatic object segmentation and mask generation. The raw masks were further refined through denoising, morphological opening and closing operations using `cv2.morphologyEx()`, and contour smoothing, resulting in clean and well-defined binary masks. From these masks, valid objects were selected based on an area ratio criterion, retaining only those whose area occupied between 10% and 60% of the panel region.

For each valid object, we computed its centroid and adaptively defined a grid based on the object's width and height, with the grid step size determined as $step = \max(20, \min(w_{obj}, h_{obj})/2)$. The farthest grid position from the source centroid was chosen as the target location for object placement. To avoid overly regular results, a random noise bias was added to the computed coordinates, while enforcing strict boundary constraints to ensure that the duplicated object remained within the valid image region. During this process, a corresponding binary mask of the tampered region was generated and saved. Each operation ultimately yielded a structured triplet (`original image, forged image, tampering mask`), providing standardized data for the Single-Mode Forgery Identification and Localization task.

**Quality Control.** To ensure the usability and realism of the synthetic data, a multi-stage screening strategy was applied: (1) Objects that fell outside the designated area ratio threshold (smaller than 10% or larger than 60%) were discarded, as these produced unrealistic manipulations either too trivial or too dominant; (2) Synthetic images with duplicated regions that extend beyond valid panel boundaries or overlap with semantically meaningless background (e.g., blank margins, figure legends, or annotation text) were excluded; (3) Samples were filtered if the duplicated region introduced strong visual artifacts, such as unnatural seams, sharp edges, or visible copy boundaries that would make detection trivial; and (4) Human annotators performed a manual validation stage to review a subset of the generated data, removing images that failed to preserve visual plausibility. After these steps, a total of 92 high-quality Copy-Move samples were retained for downstream analysis.

### A.3.4 AI-GENERATED FORGERIES

**Generation Pipeline.** To construct an AI-Generated forgery dataset at the panel level, we designed controlled strategies to ensure both realism and diversity while retaining detailed fidelity to authentic academic images. Two representative forgery types were considered:

(1) **Image Inference Forgery**: The entire panel was globally reconstructed to simulate complete replacement, primarily using the high-performing Flux (Labs et al., 2025) model.

(2) **Targeted Region Editing**: Partial manipulations were performed while preserving the majority of image content, where localized regions were replaced or edited based on textual prompts or binary masks. This class of forgeries was implemented using mainstream generative models including Stable Diffusion (Esser et al., 2024), SenseNova V6 Miaohua (SenseTime, 2025), and GPT-Image-1 (OpenAI, 2025). During synthesis, we systematically recorded the original panel, the forged panel, and the corresponding region masks (the latter only for targeted editing), thereby forming structured triplets `(original image, forged image, mask/metadata)`. In total, more than 5,000 AI-Generated panels were produced as negative samples.

**Quality Control.** To guarantee the authenticity and research value of the synthetic dataset, a rigorous multi-stage quality control pipeline was adopted. At the automated stage, rules were applied to filter out invalid or failed generations, including samples with excessive blur, severe distortions, corrupted text, or local edits that conflicted with global semantics. At the manual stage, human annotators further inspected and excluded low-quality cases that evaded automated filtering but still exhibited implausible details. Through iterative refinement combining automated screening and human verification, a total of 74 high-quality entirely AI-Generated images and 550 partially AI-Generated images were retained, forming a reliable and challenging negative sample set for the subsequent Identification and Localization task.

### A.3.5 DUPLICATION FORGERIES

**Generation Pipeline.** This pipeline focuses on constructing Duplication forgeries at the panel level, which are divided into two categories: **Global Duplication** and **Local Duplication**.

(1) **Global Duplication**: This process is performed by directly reusing the entire panel after applying a randomly selected geometric transformation, including horizontal flip (`cv2.flip(image,1)`), vertical flip (`cv2.flip(image,0)`), or 180° rotation (`cv2.rotate(image,ROTATE_180)`). In addition, up to two random parameter modifications are optionally applied, such as brightness modification, contrast adjustment, saturation control, hue shifting, and color temperature variation. These operations are implemented via OpenCV routines (e.g., `cv2.cvtColor()`, `cv2.convertScaleAbs()`, and `cv2.LUT()`), ensuring realistic duplication while preserving overall image integrity.

(2) **Local Duplication**: This category specifically targets stained micrographs, which typically exhibit highly self-similar textures and repetitive structural units. Such properties make localized duplications visually subtle and challenging to detect. To ensure sufficient detail, only panels with resolutions larger than $350 \times 350$ pixels are considered. For each eligible image, an initial overlapping crop is performed by randomly selecting one of three geometric modes: diagonal, vertical window, or horizontal window, producing two sub-panels (denoted as panel A and panel B) with a clear overlapping region.

Forgery generation then follows an innovative **crop-then-transform** strategy. Based on the initial crop, we adaptively perform a secondary cropping step to precisely extract blocks containing the overlapping area. Panel A is preserved as reference, while panel B undergoes random manipulation operations, including geometric transformations (i.e., rotation, flip, or scaling) and parameter modifications (i.e., brightness, contrast, saturation, hue, or color temperature). To maintain dimensional consistency under scaling, we design a **reverse cropping** mechanism: given a random target scaling factor, the required crop size is inversely computed so that, after scaling, panel B's block perfectly matches panel A in size.

All manipulation operations are recorded in metadata, together with a key indicator, the **duplication ratio**, defined as the proportion of the reused area within panel A relative to its total block area. Each forgery sample is stored as a structured triplet `(original panel, duplicated panel, metadata)`, supporting downstream model evaluation.

**Quality Control.** To guarantee usability and prevent trivial cases, multiple filtering mechanisms are applied. Panels that fail the minimum resolution requirement ($350 \times 350$ pixels) are excluded, as they lack sufficient texture for meaningful duplication. For **Global Duplication**, samples producing strong distortions or obvious artifacts after geometric or parameter operations are discarded.

For **Local Duplication**, pairs with excessively small overlapping areas, severe misalignment after transformations, or visible boundaries at splice seams are removed. A manual inspection stage further eliminates unrealistic duplications, such as cases where duplicated regions overlap with non-semantic areas (e.g., blank margins, legends, or annotation text). This multi-stage process ensures that the retained Duplication dataset exhibits both high visual plausibility and adequate difficulty for forgery detection research. After these steps, a total of 917 high-quality Global Duplication samples and 2,124 high-quality Local Duplication samples are retained.

### A.3.6 TEXT–IMAGE INCONSISTENCY FORGERIES

**Generation Pipeline.** We adopt two controlled types of caption/related-sentence manipulations at the panel level: **Numerical Inconsistency** and **Trend Inconsistency**.

(1) **Numerical Inconsistency**: We randomly select one to three numeric expressions appearing in the figure caption or the set of related sentences, and replace each selected numeric token with a different number (ensuring the replacement is not identical to the original).

(2) **Trend Inconsistency**: We randomly select one to three trend-describing keywords (e.g., increase, rise, and upward) and replace each with an appropriate antonym (e.g., decrease, fall, and downward).

The generation process is subject to the following constraints:

(1) A candidate token for modification must not be located inside the same punctuation-delimited substring (comma, period or semicolon). Concretely, we split captions and related sentences by the delimiters `[.,;]` and only allow substitutions on tokens belonging to different segments so as to avoid altering multiple values/claims within a single tight clause.

(2) A candidate keyword/number must **not** be directly present as OCR text within the image (i.e., it must require reasoning beyond raw visual OCR). We verify this by running OCR on the panel and rejecting any candidate token that appears verbatim in the OCR output.

(3) For numerical replacements, we enforce syntactic and unit consistency (preserve units when present, avoid inserting non-numeric characters), and ensure the replaced value differs meaningfully (e.g., absolute or relative difference above a small threshold) to avoid near-identical noise edits.

(4) For trend replacements, substitutions are chosen from a curated antonym mapping and are applied with part-of-speech checks to preserve grammaticality.

The pipeline draws candidate modifications automatically (via regex for numeric tokens and a curated trend-keyword list for trend tokens), applies candidate substitutions, and records for each generated sample the tuple `(original text, modified text, modification type (numerical/trend), modified token indices, original token values, replacement values)`.

**Quality Control.** All generated Text–Image Inconsistency samples undergo manual proofreading by at least two human experts. During the quality control phase, we pruned the candidate pool by filtering out nonsensical substitutions and manually adjusting borderline cases to ensure high fidelity. Through this manual validation process, we guarantee that the final dataset retains 150 high-quality, reliable samples suitable for evaluating models' ability to identify and localize cross-modal inconsistency.

### A.4 EVALUATION TASK DESIGN

To evaluate the comprehensive perception and reasoning capabilities of MLLMs across the five fraud methods, we designed three core evaluation tasks. We provide the standardized prompt templates for each task below, including the specific system prompts and user instructions used in our benchmark.

#### A.4.1 SINGLE-MODE FORGERY IDENTIFICATION AND LOCALIZATION (SMF-IL)

This task corresponds to three forgery methods: **Splicing**, **Copy-Move**, and **AI-Generated**. It requires the model to judge whether the input academic panel is forged and select the fraud type from five options (*Splicing*, *Copy-Move*, *AI-Generated*, *No Forgery*, and *Not Sure*). If forged, the model must further provide block-based localization of the forged region.

---

**Single-Mode Forgery Identification and Localization (SMF-IL)**

**System Prompt**:
**Role**:
You are an image forensics expert. Detect the image forgery type and localize if needed.

**Task**:
Single-choice task (pick **EXACTLY** one):
A) **Splicing** — different sources of content combined in one image; cues: unnatural seams, boundary mismatches, inconsistent lighting/textures. Note: Splicing **NEVER** involves duplicated regions.
B) **Copy-Move** — a region copied and pasted within the **SAME** image; cues: duplicated patterns/objects with same noise/texture but shifted/rotated/scaled. Reminder: if any part of the image is duplicated, it **MUST** be classified as Copy-Move, not Splicing.
C) **AI-Generated** — image wholly or partially AI-generated/modified; cues: synthetic textures, implausible details, inconsistent semantics.
D) **No Forgery** — the image appears genuine; no strong evidence of manipulation.
E) **Not Sure** — insufficient evidence to decide.

**Input**:
- ONE image (a panel/subfigure).
- A textual description of how the image is partitioned into numbered regions (`mask_caption`).

**Mask Rules**:
- If mask **IS REQUIRED** and your choice is A/B/C → output indices of forged regions (comma-separated).
- If mask is **NOT** required (e.g., whole-image forgery) → `<MASK>` must be **EMPTY**, even if you chose A/B/C.
- If choice is D/E → always leave `<MASK>` **EMPTY**.

**STRICT OUTPUT (exact format, no extra text)**:
`<CHOICE>A/B/C/D/E</CHOICE>`
`<MASK>`comma-separated region numbers; **EMPTY** when not required or when choice is D/E`</MASK>`
`<EXPLANATION>`Brief reasoning: state what visual cues led to your choice; if A/B/C with mask, mention where forgery cues were observed.`</EXPLANATION>`

**User Instruction**:
Single-panel image for forensics.
**(Non-global AI-generation for the input image)**
Mask **IS REQUIRED** if you choose A/B/C; output indices from `mask_caption`.
**(Global AI-generation for the input image)**
Mask is **NOT** required for this question; LEAVE `<MASK>` **EMPTY**.
mask_caption: {`mask_caption` or `[no caption]`}

---

### A.4.2 COMPOSITE MANIPULATION OPERATIONS IDENTIFICATION (CMO-I)

This task corresponds to **Duplication**. Given a pair of panels, the model is required to identify whether a duplication relationship exists and the specific types of manipulation operations (*Direct Reuse*, *Scaling*, *Rotation*, *Flip*, and *Parameter Modification*) involved. Multiple selections are permitted, covering positive, negative, and uncertain cases.

---

**Composite Manipulation Operations Identification (CMO-I)**

**System Prompt**:
**Role**:
You are an image forensics expert specializing in cross-panel duplication.

**Task**:
Multi-select task (choose **ALL** that apply):
A) **Direct** — near-identical content across panels without geometric transform; many keypoints align at similar scale/position.
B) **Scaling** — content in one panel appears in another after zoom-in/zoom-out.
C) **Rotated** — one panel is a rotated version of another (e.g., 90°/180°).
D) **Flipped** — one panel is a mirrored (horizontal/vertical) version of another.
E) **Parameter-Modified** — duplication exists but with image-level adjustments (e.g., brightness, contrast, and color).
F) **No Duplication** — there is no convincing duplication across panels.
G) **Not Sure** — evidence is insufficient or ambiguous.

**Rules**:
- Pick A/B/C/D/E for every duplication type that appears. Multiple types may co-exist (e.g., scaling + rotated ⇒ choose B,C).
- If none of A–E apply, choose F (no duplication).
- If uncertain, choose G (not sure).
- Do **NOT** mix F or G with A–E.

**Input**:
- You will receive **MULTIPLE** panels (subfigures) from the same figure.
- A short text may describe panel numbering, but you do not need to output indices.

**STRICT OUTPUT (exact format)**:
<CHOICES>comma-separated letters from {A,B,C,D,E,F,G};
uppercase only</CHOICES>
<EXPLANATION>Brief reasoning: which panels appear duplicated and why; mention cues like keypoint matches, geometric transforms, or parameter changes.</EXPLANATION>

**User Instruction**:
Multiple panels for cross-panel duplication forensics.
panel note: {mask_caption or [no caption]}

---

### A.4.3 CROSS-MODAL INCONSISTENCY IDENTIFICATION AND LOCALIZATION (CMI-IL)

This task corresponds to **Text–Image Inconsistency**. It requires the model to judge whether **Numerical** or **Trend** inconsistency exists based on the input image and its corresponding text. If detected, the model must further localize the minimum sentence unit with inconsistency in the text and provide corrected content.

---

**Cross-Modal Inconsistency Identification and Localization (CMI-IL)**

**System Prompt**:
**Role**:
You are an expert in text–image consistency for scientific figures.

**Task**:
Single-choice (pick **EXACTLY** one):
A) **Numerical Inconsistent** — the numeric values/assertions in text contradict the figure's numbers/labels/scales.
B) **Trend Inconsistent** — the trend described in text (e.g., increase, decrease, peak, or monotonicity ordering) contradicts the figure's visual trends.
C) **Consistent** — text matches the figure.
D) **Not Sure** — image/text too ambiguous or unreadable to decide.

**Rules**:
- Decide A/B/C/D.
- If you choose A or B, localize **ALL** problematic parts.
- Output `<PARTS>` with one or more entries: `caption` if the issue is in the caption, `related` if the issue is in the related text.
- Output `<SENTENCES>` with **ALL** minimal inconsistent sentences.
- A *minimal sentence* is the smallest segment separated by '.', '?', '!', ',', ';', ':'.
- If you choose C or D, set `<PARTS>none</PARTS>` and leave `<SENTENCES>` **EMPTY**.

**Input**:
You will receive:
- ONE figure image.
- figure_caption: the figure's caption.
- figure_related: a short context related to the figure.

**STRICT OUTPUT (exact format)**:
`<CHOICE>A/B/C/D</CHOICE>`
`<PARTS>comma-separated list of caption/related/none</PARTS>`
`<SENTENCES>list ALL minimal inconsistent sentences;`
**EMPTY** for C/D`</SENTENCES>`
`<EXPLANATION>`Brief reasoning: what in the text contradicts which element in the figure; or why consistent/not sure.`</EXPLANATION>`

**User Instruction**:
Figure for text–image consistency checking.
figure caption: {`figure_caption` or `[empty]`}
figure related: {`figure_related` or `[empty]`}
Remember: choose A/B/C/D, and if A/B, return PART=caption/related and a single minimal SENTENCE.

---

# B EXPERIMENTAL DETAILS AND SETUP

## B.1 EVALUATION ENVIRONMENT

For evaluation experiments, most model inferences are conducted via the OpenRouter API[5]. Exceptions include InternVL3.5-8B and the LLaVA series, which are downloaded from Hugging Face[6] and executed locally, as well as the Doubao series, which is accessed via the Volcano Engine API[7].

The system configuration is summarized below:

- **CPU**: Dual-socket Intel Xeon Gold 6148 (2.40 GHz), 20 cores per socket, 80 threads total
- **GPU**: 8 × NVIDIA A40 (48 GB VRAM each)
- **GPU Driver**: 575.57.08
- **CUDA**: 11.8
- **cuDNN**: 8.9.6 (compiled with CUDA 11.8)
- **Operating System**: Ubuntu 22.10

## B.2 BENCHMARKED MODELS

We evaluate 16 MLLMs in total, including nine proprietary and seven open-source models. The proprietary models come from OpenAI (Singh et al., 2025; OpenAI, 2025), Google (Comanici et al., 2025), Alibaba (Bai et al., 2023), ByteDance (Team, 2025), Anthropic (Anthropic, 2025) and Zhipu AI (Team et al., 2026), representing mainstream industrial progress. The open-source models are released by Google (Team et al., 2025), Meta (Meta, 2025), Alibaba (Bai et al., 2025), the LLaVA community (Li et al., 2025a; Liu et al., 2024) and OpenGVLab (Wang et al., 2025c), covering different scales and architectures of open-source MLLMs. The following list details these models.

Table 7: Detailed information on benchmarked MLLMs.

| Provider | Version | Parameters | Access Links |
|---|---|---|---|
| *Proprietary MLLMs* | | | |
| OpenAI | GPT-5 | N/A | `https://openrouter.ai/openai/gpt-5` |
| | OpenAI o4-mini-high | N/A | `https://openrouter.ai/openai/o4-mini-high` |
| Google | Gemini 2.5 Flash | N/A | `https://openrouter.ai/google/gemini-2.5-flash` |
| | Gemini 2.5 Pro | N/A | `https://openrouter.ai/google/gemini-2.5-pro` |
| Alibaba | Qwen-VL-Max | N/A | `https://openrouter.ai/qwen/qwen-vl-max` |
| ByteDance | Doubao-Seed-1.6-thinking | N/A | `https://console.volcengine.com/ark/region:ark+cn-beijing/model/detail?Id=doubao-seed-1-6-thinking` |
| | Doubao-Seed-1.6-vision | N/A | `https://console.volcengine.com/ark/region:ark+cn-beijing/model/detail?Id=doubao-seed-1-6-vision` |
| Anthropic | Claude Sonnet 4.5 | N/A | `https://openrouter.ai/anthropic/claude-sonnet-4.5` |
| Z.ai | GLM-4.5V | N/A | `https://openrouter.ai/z-ai/glm-4.5v` |
| *Open-Source MLLMs* | | | |
| Google | Gemma 3 27B | 27B | `https://openrouter.ai/google/gemma-3-27b-it` |
| Meta | Llama 4 Maverick | 400B-A17B | `https://openrouter.ai/meta-llama/llama-4-maverick` |
| Alibaba | Qwen2.5-VL-32B | 32B | `https://openrouter.ai/qwen/qwen2.5-vl-32b-instruct` |
| | Qwen2.5-VL-72B | 72B | `https://openrouter.ai/qwen/qwen2.5-vl-72b-instruct` |
| LLaVA Community | LLaVA-Interleave-7B-DPO | 7B | `https://huggingface.co/llava-hf/llava-interleave-qwen-7b-dpo-hf` |
| | LLaVA-NeXT-34B | 34B | `https://huggingface.co/llava-hf/llava-v1.6-34b-hf` |
| OpenGVLab | InternVL3.5-8B | 8B | `https://huggingface.co/OpenGVLab/InternVL3_5-8B` |

---

[5] `https://openrouter.ai`
[6] `https://huggingface.co`
[7] `https://www.volcengine.com`

## B.3 EVALUATION METRICS AND PROTOCOLS

### B.3.1 TASK-SPECIFIC METRICS

To comprehensively assess model performance, we employ **five core metrics** tailored to **three tasks**. For all metrics described below, the final performance reported in our results represents the mean score across all samples in the respective task.

**Task 1: Single-Mode Forgery Identification and Localization (SMF-IL).** This task involves **two complementary metrics**:

(1) **Forgery Type Identification**. The model is required to categorize the input image into specific forgery types or mark it as authentic (a single-choice classification). We use **Accuracy (ACC)** as the primary metric:

$$\text{ACC}_{\text{sm}} = \frac{1}{N} \sum_{i=1}^{N} \mathbf{1}(\hat{y}_i = y_i), \tag{4}$$

where $N$ is the number of samples, $\hat{y}_i$ is the predicted class, and $y_i$ is the ground-truth label.

(2) **Forged Regions Localization**. For images identified as forged, the model must output a binary mask indicating the tampered region. We quantify the overlap between the predicted mask $\hat{M}$ and the ground-truth mask $M$ using **Intersection-over-Union (IoU)**:

$$\text{IoU} = \frac{|M \cap \hat{M}|}{|M \cup \hat{M}|}. \tag{5}$$

**Task 2: Composite Manipulation Operations Identification (CMO-I).** Since a single image may undergo multiple manipulation operations (e.g., scaling and rotation), this is formulated as a multi-label classification problem. We adopt the **Set-based F1 Score** to measure the match between the predicted set of operations $\hat{S}$ and the ground-truth set $S$:

$$\text{Precision} = \frac{|\hat{S} \cap S|}{|\hat{S}|}, \quad \text{Recall} = \frac{|\hat{S} \cap S|}{|S|}, \tag{6}$$

$$\text{F1}_{\text{set}} = \frac{2 \cdot \text{Precision} \cdot \text{Recall}}{\text{Precision} + \text{Recall}}. \tag{7}$$

**Task 3: Cross-Modal Inconsistency Identification and Localization (CMI-IL).** This task involves **two complementary metrics**:

(1) **Inconsistency Identification**. The model determines the existence and type of inconsistency (Numerical or Trend). We use **ACC** as the primary metric:

$$\text{ACC}_{\text{cm}} = \frac{1}{N} \sum_{i=1}^{N} \mathbf{1}(\hat{y}_i = y_i). \tag{8}$$

(2) **Inconsistency Localization**. The model must extract the specific text span containing the error. We compute the **Text-based F1 Score** by comparing the predicted text span $\hat{T}$ with the ground-truth span $T$:

$$\text{F1}_{\text{text}} = \frac{2 \cdot |\hat{T} \cap T|}{|\hat{T}| + |T|}. \tag{9}$$

Here, $|\cdot|$ denotes the length of the span.

### B.3.2 BALANCED ROBUSTNESS INDEX (BRI)

Beyond task-specific metrics, we define a composite score to holistically measure both **overall accuracy** and **robustness across tasks**.

**Score Normalization.** Since different tasks use different metric scales (e.g., IoU vs. ACC), we first normalize the raw scores to ensure comparability. Let $R_{i,j}$ denote the raw score of model $i$ on task dimension $j$. The normalized score $s_{i,j}$ is computed via Min-Max normalization across all $M$ benchmarked models:

$$s_{i,j} = \frac{R_{i,j} - \min_{k \in \{1...M\}} R_{k,j}}{\max_{k \in \{1...M\}} R_{k,j} - \min_{k \in \{1...M\}} R_{k,j}}. \tag{10}$$

For each model $i$, we collect its normalized performance vector across the five metric dimensions:

$$\mathbf{s}_i = [s_{i,1}, s_{i,2}, s_{i,3}, s_{i,4}, s_{i,5}].$$

**Index Calculation.** We then compute the mean performance $\mu_i$ and the stability penalty $\Delta_i$ (representing the performance gap across tasks):

$$\mu_i = \frac{1}{5} \sum_{j=1}^{5} s_{i,j}, \tag{11}$$

$$\Delta_i = \max_j(s_{i,j}) - \min_j(s_{i,j}). \tag{12}$$

Finally, the **Balanced Robustness Index (BRI)** is defined as:

$$\text{BRI}_i = \mu_i - \lambda \cdot \Delta_i, \tag{13}$$

where $\lambda$ is a tunable penalty weight (set to $0.25$ in our main experiments to balance average performance and cross-task variance). A higher BRI indicates that a model not only achieves strong average performance but also avoids over-specialization (i.e., maintaining a small $\Delta_i$).

**Sensitivity Analysis of $\lambda$.** To examine the stability of model rankings with respect to the penalty weight $\lambda$, we report BRI scores under different $\lambda$ values in Table 8.

Table 8: Sensitivity of the BRI under different values of $\lambda$.

| Model | $\lambda$=0.10 | $\lambda$=0.25 | $\lambda$=0.30 | $\lambda$=0.40 |
|---|---|---|---|---|
| GPT-5 | 64.56 | 56.15 | 53.35 | 47.75 |
| OpenAI o4-mini-high | 63.57 | 52.34 | 48.60 | 41.11 |
| Qwen-VL-Max | 57.31 | 49.83 | 47.34 | 42.35 |
| Gemini 2.5 Flash | 57.67 | 44.70 | 40.36 | 31.71 |
| Doubao-Seed-1.6-thinking | 48.43 | 37.14 | 33.37 | 25.85 |
| Doubao-Seed-1.6-vision | 46.17 | 33.47 | 29.24 | 20.78 |
| Gemini 2.5 Pro | 46.10 | 31.97 | 27.25 | 17.83 |
| GLM-4.5V | 40.66 | 31.57 | 28.54 | 22.48 |
| Claude Sonnet 4.5 | 38.74 | 29.96 | 27.03 | 21.18 |
| Qwen2.5-VL-72B | 58.60 | 47.16 | 43.34 | 35.71 |
| InternVL3.5-8B | 51.58 | 38.73 | 34.45 | 25.89 |
| Llama 4 Maverick | 40.00 | 34.78 | 33.04 | 29.56 |
| LLaVA-Interleave-7B-DPO | 32.79 | 23.59 | 20.52 | 14.38 |
| LLaVA-NeXT-34B | 31.68 | 18.40 | 13.97 | 5.11 |
| Qwen2.5-VL-32B | 29.76 | 18.22 | 14.37 | 6.68 |
| Gemma 3 27B | 17.37 | 9.59 | 7.00 | 1.81 |

Overall, the relative ranking of top-performing models (e.g., GPT-5, OpenAI o4-mini-high, and Qwen-VL-Max) remains stable across different penalty weights. As $\lambda$ increases, absolute BRI values decrease, reflecting a stronger emphasis on stability over peak task performance.

## C  SUPPLEMENTARY RESULTS AND ANALYSIS

### C.1  EVALUATION ON REAL-WORLD DATA

Table 9: **Performance comparison on synthetic versus real data.** We report the **Id Score** (Identification Score) and **Loc Score** (Localization Score) across 3 subtasks corresponding to different fraud types.

| Model | Data Type | Splicing | | Copy-Move | | Duplication |
|---|---|---|---|---|---|---|
| | | Id Score | Loc Score | Id Score | Loc Score | Id Score |
| *Proprietary MLLMs* | | | | | | |
| GPT-5 | Synthetic | 43.51 | 16.67 | 72.73 | 36.41 | 33.32 |
| | Real | 27.27 | 23.96 | 60.00 | 33.39 | 23.22 |
| OpenAI o4-mini-high | Synthetic | 41.49 | 10.44 | 77.89 | 29.78 | 30.34 |
| | Real | 9.09 | 11.99 | 60.00 | 31.35 | 36.04 |
| Qwen-VL-Max | Synthetic | 30.37 | 40.07 | 87.40 | 48.34 | 23.33 |
| | Real | 0.00 | 15.15 | 13.33 | 3.70 | 20.95 |
| Gemini 2.5 Flash | Synthetic | 63.39 | 56.72 | 67.56 | 46.98 | 24.96 |
| | Real | 9.09 | 38.55 | 80.00 | 47.58 | 49.58 |
| *Open-Source MLLMs* | | | | | | |
| Qwen2.5-VL-72B | Synthetic | 36.40 | 51.66 | 77.27 | 55.16 | 16.75 |
| | Real | 0.00 | 54.54 | 100.00 | 36.27 | 22.39 |
| Llama 4 Maverick | Synthetic | 23.37 | 17.97 | 51.24 | 28.50 | 19.01 |
| | Real | 0.00 | 8.66 | 6.67 | 2.22 | 39.75 |
| Gemma 3 27B | Synthetic | 25.39 | 27.78 | 28.87 | 32.80 | 12.20 |
| | Real | 0.00 | 15.25 | 40.00 | 17.04 | 42.18 |
| Avg. | Synthetic | 37.70 | 31.62 | 66.14 | 39.71 | 22.84 |
| | Real | 6.49 | 24.01 | 51.43 | 24.51 | 33.44 |

As shown in Table 9, our synthetic data exert comparable visual fraud reasoning pressure to real data, as they fully reflect diverse real-world fraud behaviors and cover the known fraudulent operations reported in retracted papers. A detailed comparison across specific subtasks reveals several key observations:

- Synthetic data surpass real data in difficulty for the Duplication subtask because they incorporate multiple compounded manipulations. These include geometric transformations (i.e., horizontal flip, vertical flip, and 180° rotation) and parameter modifications (i.e., brightness, contrast, saturation, hue, and color temperature), which are generally more complex than the direct reuse or scaling duplication commonly observed in real cases. For example, Gemini 2.5 Flash reached a 49.58% Id Score on real cases, yet dropped to only 24.96% on synthetic data.

- Model performance on synthetic and real data is closely aligned for the Copy-Move subtask. For instance, GPT-5 achieved a 36.41% Loc Score on synthetic data, compared to 33.39% on real data.

- The primary exception lies in the Splicing subtask, where real samples remain more challenging due to their inherently more sophisticated merging patterns. Real-world forgery operations often employ more complex and subtle splicing strategies, such as localized contrast, brightness adjustment, and texture smoothing at the merged boundaries.

Moreover, model performance on synthetic data serves as a reliable indicator of real-world capability. For example, GPT-5, the strongest model on synthetic visual fraud reasoning tasks, also exhibits relatively stable performance on real samples. Thus, synthetic data not only provide comprehensive

testing of reasoning capabilities but also function as a predictive signal for real scientific review scenarios.

The convergence of performance between synthetic and real data demonstrates that our fraud data generation pipeline can continuously produce high-quality and scalable fraud samples despite the scarcity and prohibitively high annotation cost of real-world cases. This enables robust evaluation of five core capabilities (i.e., Expert Knowledge Utilization, Visual Recognition, Spatial Reasoning, Region Localization, and Comparative Reasoning) within an extensible testbed for multimodal scientific fraud assessment.

An improved splicing synthesis method will be incorporated in future work to better mimic these nuanced operations and more effectively expose current models' limitations in boundary perception.

## C.2 IMPACT OF PROMPTING STRATEGIES

Table 10: **Performance comparison of different MLLMs under various prompting strategies on synthetic data.** We report the **Id Score** (Identification Score) and **Loc Score** (Localization Score) for 5 fraud methods: **SPL** (Splicing), **CM** (Copy-Move), **AIG** (AI-Generated), **DUP** (Duplication), and **TII** (Text–Image Inconsistency). **Llama**: Llama 4 Maverick.

| Model | Prompting Strategy | SPL | | CM | | AIG | | DUP | TII | |
|-------|-------------------|-----|-----|-----|-----|-----|-----|-----|-----|-----|
| | | Id Score | Loc Score | Id Score | Loc Score | Id Score | Loc Score | Id Score | Id Score | Loc Score |
| GPT-5 | None | 43.51 | 16.67 | 72.73 | 36.41 | 44.26 | 19.33 | 33.32 | 60.67 | 27.44 |
| | CoT | 44.85 (+1.34) | 25.16 (+8.49) | 74.46 (+1.73) | 38.74 (+2.33) | 43.36 (-0.90) | 30.10 (+10.77) | 33.95 (+0.63) | 66.00 (+5.33) | 25.16 (-2.28) |
| | Few-Shot | 46.47 (+2.96) | 18.80 (+2.13) | 71.97 (-0.76) | 31.47 (-4.94) | 43.26 (-1.00) | 27.46 (+8.13) | - | - | - |
| Llama | None | 23.37 | 17.97 | 51.24 | 28.50 | 58.19 | 19.71 | 19.01 | 54.00 | 16.08 |
| | CoT | 28.24 (+4.87) | 49.34 (+31.37) | 60.21 (+8.97) | 28.47 (-0.03) | 59.08 (+0.89) | 34.06 (+14.35) | 28.29 (+9.28) | 61.66 (+7.66) | 13.64 (-2.44) |
| | Few-Shot | 31.97 (+8.60) | 9.25 (-8.72) | 62.03 (+10.79) | 14.93 (-13.57) | 40.80 (-17.39) | 13.06 (-6.65) | - | - | - |

We further compare two prompting methods, CoT (Wei et al., 2022) and few-shot (Brown et al., 2020), across different fraud types. The results in Table 10 indicate that:

- CoT generally provides consistent gains, but the magnitude varies with model strength. For GPT-5, the improvements were modest (e.g., +1.34% on SPL-Id and +1.73% on CM-Id), suggesting that a model with strong intrinsic reasoning ability benefits only marginally. By contrast, Llama 4 Maverick showed large boosts with CoT (e.g., +4.87% on SPL-Id, +8.97% on CM-Id, and +9.28% on DUP-Id), demonstrating that stepwise prompting disproportionately enhances weaker models by guiding them toward structured reasoning steps.

- Few-shot prompting, however, yields mixed and unstable effects. While it improved certain scores such as SPL-Id (+8.60%) and CM-Id (+10.79%) for Llama 4 Maverick, it simultaneously caused severe drops in other subtasks, including AIG-Id (-17.39%) and AIG-Loc (-6.65%). GPT-5 exhibited a similar trend: slight improvements in some subtasks (e.g., +2.96% on SPL-Id and +8.13% on AIG-Loc) but declined in others (e.g., -4.94% on CM-Loc). These inconsistencies suggest that the few-shot strategy may bias models toward local demonstration patterns rather than enabling generalizable reasoning, leading to degradation when facing diverse fraud types.

Overall, the evidence indicates that CoT is a more reliable prompting strategy for scientific image forensics, providing stable and interpretable improvements, whereas few-shot prompting lacks robustness and exhibits limited practical value in our setting.

We provide the enhanced prompting templates used in our benchmark, including CoT and few-shot strategies.

**Chain-of-Thought Single-Mode Forgery Identification and Localization**

**System Prompt**:
**Role**:
You are an image forensics expert. Detect the image forgery type and localize if needed.

**Task**:
Single-choice task (pick **EXACTLY** one):
A) **Splicing** — different sources of content combined in one image; cues: unnatural seams, boundary mismatches, inconsistent lighting/textures. Note: Splicing **NEVER** involves duplicated regions.
B) **Copy-Move** — a region copied and pasted within the **SAME** image; cues: duplicated patterns/objects with same noise/texture but shifted/rotated/scaled. Reminder: if any part of the image is duplicated, it **MUST** be classified as Copy-Move, not Splicing.
C) **AI-Generated** — image wholly or partially AI-generated/modified; cues: synthetic textures, implausible details, inconsistent semantics.
D) **No Forgery** — the image appears genuine; no strong evidence of manipulation.
E) **Not Sure** — insufficient evidence to decide.

**Input**:
- ONE image (a panel/subfigure).
- A textual description of how the image is partitioned into numbered regions (`mask_caption`).

**Mask Rules**:
- If mask **IS REQUIRED** and your choice is A/B/C → output indices of forged regions (comma-separated).
- If mask is **NOT** required (e.g., whole-image forgery) → `<MASK>` must be **EMPTY**, even if you chose A/B/C.
- If choice is D/E → always leave `<MASK>` **EMPTY**.

**Reasoning Requirement (Chain-of-Thought)**:
- First, **carefully observe the image** and list suspicious visual cues such as textures, boundaries, lighting, duplication, and semantic plausibility.
- Next, **compare against all five options (A–E)** and explicitly explain why each is ruled in or ruled out.
- Then, **narrow down to the most plausible choice**.
- If A/B/C is chosen, explain **where the forgery appears** and why the mask is required or not.
- Keep the reasoning concise but clearly step by step.

**STRICT OUTPUT (exact format, no extra text)**:
`<CHOICE>A/B/C/D/E</CHOICE>`
`<MASK>`comma-separated region numbers; **EMPTY** when not required or when choice is D/E`</MASK>`
`<EXPLANATION>`Provide the step-by-step reasoning process as described above, ending with justification for the final choice.`</EXPLANATION>`

## Chain-of-Thought Composite Manipulation Operations Identification

**System Prompt**:
**Role**:
You are an image forensics expert specializing in cross-panel duplication.

**Task**:
Multi-select task (choose **ALL** that apply):
A) **Direct** — near-identical content across panels without geometric transform; many keypoints align at similar scale/position.
B) **Scaling** — content in one panel appears in another after zoom-in/zoom-out.
C) **Rotated** — one panel is a rotated version of another (e.g., 90°/180°).
D) **Flipped** — one panel is a mirrored (horizontal/vertical) version of another.
E) **Parameter-Modified** — duplication exists but with image-level adjustments (e.g., brightness, contrast, and color).
F) **No Duplication** — there is no convincing duplication across panels.
G) **Not Sure** — evidence is insufficient or ambiguous.

**Rules**:
- Pick A/B/C/D/E for every duplication type that appears. Multiple types may co-exist (e.g., scaling + rotated ⇒ choose B,C).
- If none of A–E apply, choose F (no duplication).
- If uncertain, choose G (not sure).
- Do **NOT** mix F or G with A–E.

**Input**:
- You will receive **MULTIPLE** panels (subfigures) from the same figure.
- A short text may describe panel numbering, but you do not need to output indices.

**Reasoning Requirement (Chain-of-Thought)**:
- First, **systematically compare all panels** to check for visual similarities.
- Next, **analyze the nature of the similarity**: decide whether it is a direct reuse, scaling, rotation, flip, or parameter modification.
- Then, **evaluate each option A–E one by one**, ruling them in or out.
- If none of A–E apply, consider F (no duplication).
- If evidence is weak or ambiguous, consider G (not sure).
- In <EXPLANATION>, provide step-by-step reasoning: what similarities were found, which transformations are involved, and why the final choice(s) were made.

**STRICT OUTPUT (exact format)**:
<CHOICES>comma-separated letters from {A,B,C,D,E,F,G};
uppercase only</CHOICES>
<EXPLANATION>Step-by-step reasoning: explain which panels appear duplicated, what transformations or adjustments are observed, and why the final choices are selected.</EXPLANATION>

---

### Chain-of-Thought Cross-Modal Inconsistency Identification and Localization

**System Prompt**:
**Role**:
You are an expert in text–image consistency for scientific figures.

**Task**:
Single-choice (pick **EXACTLY** one):
A) **Numerical Inconsistent** — the numeric values/assertions in text contradict the figure's numbers/labels/scales.
B) **Trend Inconsistent** — the trend described in text (e.g., increase, decrease, peak, or monotonicity ordering) contradicts the figure's visual trends.
C) **Consistent** — text matches the figure.
D) **Not Sure** — image/text too ambiguous or unreadable to decide.

**Rules**:
- Decide A/B/C/D.
- If you choose A or B, localize **ALL** problematic parts.
- Output `<PARTS>` with one or more entries: `caption` if the issue is in the caption, `related` if the issue is in the related text.
- Output `<SENTENCES>` with **ALL** minimal inconsistent sentences.
- A *minimal sentence* is the smallest segment separated by '.', '?', '!', ',', ';', ':'.
- If you choose C or D, set `<PARTS>none</PARTS>` and leave `<SENTENCES>` **EMPTY**.

**Input**:
You will receive:
- ONE figure image.
- figure_caption: the figure's caption.
- figure_related: a short context related to the figure.

**Reasoning Requirement (Chain-of-Thought)**:
- First, **carefully observe the figure** and extract key information (eg., numeric values, axis labels, visual trends, and peaks).
- Next, **analyze the caption and related text step by step**, checking their numeric claims and described trends.
- Then, **compare systematically against all four options (A–D)**, explaining why each is ruled in or out.
- Finally, choose the most plausible option.
- If A or B is chosen, clearly indicate whether the inconsistency appears in the caption or the related text, and list all minimal inconsistent sentences.
- If C or D is chosen, set `<PARTS>none</PARTS>` and leave `<SENTENCES>` empty.
- In `<EXPLANATION>`, write out the reasoning step by step (observation → checking text → option comparison → final decision).

**STRICT OUTPUT (exact format)**:
`<CHOICE>A/B/C/D</CHOICE>`
`<PARTS>comma-separated list of caption/related/none</PARTS>`
`<SENTENCES>list ALL minimal inconsistent sentences;`
**EMPTY** for C/D`</SENTENCES>`
`<EXPLANATION>`Step-by-step reasoning process as described above, ending with justification for the final choice.`</EXPLANATION>`

---

### Few-Shot Single-Mode Forgery Identification and Localization

**System Prompt**:
**Role**:
You are an image forensics expert. Detect the image forgery type and localize if needed.

**Task**:
Single-choice task (pick **EXACTLY** one):
A) **Splicing** — different sources of content combined in one image; cues: unnatural seams, boundary mismatches, inconsistent lighting/textures. Note: Splicing **NEVER** involves duplicated regions.
B) **Copy-Move** — a region copied and pasted within the **SAME** image; cues: duplicated patterns/objects with same noise/texture but shifted/rotated/scaled. Reminder: if any part of the image is duplicated, it **MUST** be classified as Copy-Move, not Splicing.
C) **AI-Generated** — image wholly or partially AI-generated/modified; cues: synthetic textures, implausible details, inconsistent semantics.
D) **No Forgery** — the image appears genuine; no strong evidence of manipulation.
E) **Not Sure** — insufficient evidence to decide.

**Input**:
- ONE image (a panel/subfigure).
- A textual description of how the image is partitioned into numbered regions (`mask_caption`).

**Mask Rules**:
- If mask **IS REQUIRED** and your choice is A/B/C → output indices of forged regions (comma-separated).
- If mask is **NOT** required (e.g., whole-image forgery) → `<MASK>` must be **EMPTY**, even if you chose A/B/C.
- If choice is D/E → always leave `<MASK>` **EMPTY**.

**STRICT OUTPUT (exact format, no extra text)**:
`<CHOICE>`A/B/C/D/E`</CHOICE>`
`<MASK>`comma-separated region numbers; **EMPTY** when not required or when choice is D/E`</MASK>`
`<EXPLANATION>`Brief reasoning: state what visual cues led to your choice; if A/B/C with mask, mention where forgery cues were observed.`</EXPLANATION>`

**Few-Shot Examples**:
*Example 1 - AI-Generated (C)*
`<CHOICE>`**C**`</CHOICE>`
`<MASK></MASK>`
`<EXPLANATION>`Synthetic/global artifacts and implausible micro-textures suggest AIGC.`</EXPLANATION>`
*Example 2 - Copy-Move (B)*
`<CHOICE>`**B**`</CHOICE>`
`<MASK></MASK>`
`<EXPLANATION>`Duplicated patch with the same noise/texture implies Copy-Move.`</EXPLANATION>`
*Example 3 - Splicing (A)*
`<CHOICE>`**A**`</CHOICE>`
`<MASK></MASK>`
`<EXPLANATION>`Boundary inconsistencies and lighting mismatch without duplication indicate Splicing.`</EXPLANATION>`

## C.3 Analysis of Post-Processing Results

Table 11: **Performance comparison of different MLLMs under various input perturbations on synthetic data.** We report the **Id Score** (Identification Score) and **Loc Score** (Localization Score) for 4 fraud methods: **SPL** (Splicing), **CM** (Copy-Move), **AIG** (AI-Generated), and **DUP** (Duplication). **Gemini**: Gemini 2.5 Flash.

| Model | Post-Processing | SPL | | CM | | AIG | | DUP |
|---|---|---|---|---|---|---|---|---|
| | | Id Score | Loc Score | Id Score | Loc Score | Id Score | Loc Score | Id Score |
| GPT-5 | None | 43.51 | 16.67 | 72.73 | 36.41 | 44.26 | 19.33 | 33.32 |
| | Gauss | 23.37 (-20.14) | 3.43 (-13.24) | 34.85 (-37.88) | 12.81 (-23.60) | 37.22 (-7.04) | 7.73 (-11.60) | 20.84 (-12.48) |
| | JPEG | 33.59 (-9.92) | 12.74 (-3.93) | 48.86 (-23.87) | 32.19 (-4.22) | 43.89 (-0.37) | 14.32 (-5.01) | 23.56 (-9.76) |
| | Scaling | 36.63 (-6.88) | 14.05 (-2.62) | 48.48 (-24.25) | 42.37 (+5.96) | 41.11 (-3.15) | 16.67 (-2.66) | 25.08 (-8.24) |
| Gemini | None | 63.39 | 56.72 | 67.56 | 46.98 | 35.45 | 38.87 | 24.96 |
| | Gauss | 48.13 (-15.26) | 33.63 (-23.09) | 50.38 (-17.18) | 39.74 (-7.24) | 31.11 (-4.34) | 23.20 (-15.67) | 25.84 (+0.88) |
| | JPEG | 64.82 (+1.43) | 60.01 (+3.29) | 54.93 (-12.63) | 42.46 (-4.52) | 33.90 (-1.55) | 39.93 (+1.06) | 25.56 (+0.60) |
| | Scaling | 61.37 (-2.02) | 57.46 (+0.74) | 46.97 (-20.59) | 43.86 (-3.12) | 34.44 (-1.01) | 40.75 (+1.88) | 25.31 (+0.35) |

We further introduce three common image perturbations (Gaussian blur, JPEG compression, and scaling) to systematically evaluate model robustness under input distortions. Our analysis yields the following observations:

- Gaussian blur directly disrupts fine-grained textures and boundary information, and the results showed that it had the most severe impact, causing substantial performance drops across nearly all subtasks. By contrast, JPEG compression primarily introduces quantization noise and artifacts, while scaling alters image resolution and aspect ratios; both operations also led to degradation, with the effects varying across models and subtasks.

- Under Gaussian blur, GPT-5 experienced a drop of over 23% in localization performance for Copy-Move, while Gemini 2.5 Flash suffered substantial degradation in localization performance for both Splicing and AI-Generated.

These findings highlight that current MLLMs are highly vulnerable even to mild perturbations, as they rely excessively on delicate low-level features. The lack of robust representations makes them susceptible to common distortions and post-processing operations, thereby limiting their reliability in practical forensic applications.

# D    DETAILED RESULTS BY FRAUD TYPE

## D.1    SPLICING

Table 12: **Experimental results on THEMIS synthetic data for Splicing.** We report the **ACC**, **IoU**, and **Set-based F1**. The best and second-best scores are highlighted.

| Model | Splicing | | |
|---|---|---|---|
| | **ACC** | **IoU** | **Set-based F1** |
| *Proprietary MLLMs* | | | |
| GPT-5 | 43.51 | 16.67 | 20.27 |
| OpenAI o4-mini-high | 41.49 | 10.44 | 12.81 |
| Qwen-VL-Max | 30.37 | 40.07 | 52.84 |
| Gemini 2.5 Flash | **63.39** | **56.72** | **65.08** |
| Doubao-Seed-1.6-thinking | 35.67 | 12.09 | 16.24 |
| Doubao-Seed-1.6-vision | 20.84 | 31.42 | 42.49 |
| Gemini 2.5 Pro | 30.60 | 46.65 | 59.50 |
| GLM-4.5V | 29.23 | 8.54 | 9.76 |
| Claude Sonnet 4.5 | 29.71 | 14.95 | 17.20 |
| *Open-Source MLLMs* | | | |
| Qwen2.5-VL-72B | 36.40 | 51.66 | 57.60 |
| InternVL3.5-8B | 30.97 | 43.38 | 46.37 |
| Llama 4 Maverick | 23.37 | 17.97 | 20.51 |
| LLaVA-Interleave-7B-DPO | 41.80 | 35.97 | 47.50 |
| LLaVA-NeXT-34B | 32.00 | 50.70 | 52.82 |
| Qwen2.5-VL-32B | 30.31 | 5.91 | 7.23 |
| Gemma 3 27B | 25.39 | 27.78 | 34.62 |

As shown in Table 12, the performance of different MLLMs on Splicing reveals a clear divergence between proprietary and open-source models:

• Among proprietary models, Gemini 2.5 Flash achieved the highest overall performance, with notable gains across all three metrics (ACC 63.39%, IoU 56.72%, and Set-based F1 65.08%), indicating its strong capacity for both forgery identification and fine-grained localization. In contrast, GPT-5 reached the second-best ACC (43.51%) but lagged significantly in IoU (16.67%) and Set-based F1 (20.27%), suggesting that while it can often detect the existence of splicing, it struggles to precisely delineate tampered regions. Interestingly, Qwen-VL-Max exhibited the opposite trend, with relatively low ACC (30.37%) yet competitive localization performance (IoU 40.07% and Set-based F1 52.84%), reflecting an imbalance between visual recognition and region localization.

• On the open-source side, Qwen2.5-VL-72B delivered competitive localization scores (IoU 51.66% and Set-based F1 57.60%), in some cases rivaling or surpassing proprietary counterparts. However, its classification ACC remained modest (36.40%), highlighting a persistent gap in holistic decision-making.

These results collectively indicate that while proprietary models currently dominate in balanced performance, open-source models have made notable progress in localization. A key takeaway is that splicing forensics challenges models differently at the identification and localization levels, with many MLLMs excelling at one but not both. Bridging this discrepancy represents a critical direction for advancing visual fraud reasoning.

## D.2 COPY-MOVE

Table 13: **Experimental results on THEMIS synthetic data for Copy-Move.** We report the **ACC**, **IoU**, and **Set-based F1**. The best and second-best scores are highlighted.

| Model | Copy-Move | | |
|---|---|---|---|
| | ACC | IoU | Set-based F1 |
| *Proprietary MLLMs* | | | |
| GPT-5 | 72.73 | 36.41 | 46.11 |
| OpenAI o4-mini-high | 77.89 | 29.78 | 39.32 |
| Qwen-VL-Max | **87.40** | 48.34 | 60.94 |
| Gemini 2.5 Flash | 67.56 | 46.98 | 57.90 |
| Doubao-Seed-1.6-thinking | 74.17 | 13.58 | 18.47 |
| Doubao-Seed-1.6-vision | 61.78 | 26.97 | 36.65 |
| Gemini 2.5 Pro | 47.46 | 46.61 | 60.28 |
| GLM-4.5V | 58.43 | 22.23 | 26.75 |
| Claude Sonnet 4.5 | 75.66 | 44.12 | 50.31 |
| *Open-Source MLLMs* | | | |
| Qwen2.5-VL-72B | 77.27 | **55.16** | **64.58** |
| InternVL3.5-8B | 58.42 | 40.24 | 47.75 |
| Llama 4 Maverick | 51.24 | 28.50 | 35.47 |
| LLaVA-Interleave-7B-DPO | 47.55 | 25.22 | 36.99 |
| LLaVA-NeXT-34B | 84.00 | 45.25 | 53.35 |
| Qwen2.5-VL-32B | 57.48 | 18.93 | 20.79 |
| Gemma 3 27B | 28.87 | 32.80 | 41.58 |

Table 13 presents the results of THEMIS on Copy-Move, revealing distinct trends compared to Splicing:

- Among proprietary models, Qwen-VL-Max achieved the highest identification ACC (87.40%), demonstrating strong capability in determining whether Copy-Move forgeries exist. However, its localization performance was moderate (IoU 48.34% and Set-based F1 60.94%), indicating that while detection is reliable, precise delineation of duplicated regions remains challenging. Interestingly, Gemini 2.5 Flash attained a competitive balance with relatively strong localization (IoU 46.98% and Set-based F1 57.90%), albeit at lower ACC (67.56%), suggesting better sensitivity to duplicated textures even when overall identification is less stable.

- On the open-source side, Qwen2.5-VL-72B showed the strongest localization capability (IoU 55.16% and Set-based F1 64.58%), outperforming most proprietary counterparts. This highlights the potential of scaling open-source MLLMs to large capacities for improving spatial reasoning in forensic analysis. Additionally, LLaVA-NeXT-34B achieved competitive performance (ACC 84.00%, IoU 45.25%, and Set-based F1 53.35%), positioning it as one of the few open-source models capable of approaching the performance of proprietary models in overall effectiveness.

Overall, these results suggest that open-source MLLMs are progressively bridging the performance gap with their proprietary counterparts in the Copy-Move subtask, particularly in terms of localization precision at the regional level.

## D.3 AI-GENERATED

Table 14: **Experimental results on THEMIS synthetic data for AI-Generated.** We report the **ACC**, **IoU**, and **Set-based F1**. The best and second-best scores are highlighted.

| Model | AI-Generated | | |
|---|---|---|---|
| | ACC | IoU | Set-based F1 |
| *Proprietary MLLMs* | | | |
| GPT-5 | 44.26 | 19.33 | 22.21 |
| OpenAI o4-mini-high | 35.67 | 19.79 | 22.86 |
| Qwen-VL-Max | 35.43 | 35.85 | 43.87 |
| Gemini 2.5 Flash | 35.45 | 38.87 | 44.76 |
| Doubao-Seed-1.6-thinking | 36.57 | 15.19 | 18.09 |
| Doubao-Seed-1.6-vision | 27.54 | 29.90 | 35.11 |
| Gemini 2.5 Pro | 14.90 | **47.20** | **54.66** |
| GLM-4.5V | 54.51 | 10.63 | 12.41 |
| Claude Sonnet 4.5 | 30.43 | 20.98 | 25.00 |
| *Open-Source MLLMs* | | | |
| Qwen2.5-VL-72B | 51.06 | 35.57 | 41.58 |
| InternVL3.5-8B | **67.00** | 33.19 | 37.77 |
| Llama 4 Maverick | 58.19 | 19.71 | 23.46 |
| LLaVA-Interleave-7B-DPO | 46.93 | 32.06 | 41.56 |
| LLaVA-NeXT-34B | 34.00 | 34.01 | 40.13 |
| Qwen2.5-VL-32B | 39.24 | 7.45 | 9.17 |
| Gemma 3 27B | 34.23 | 26.44 | 32.92 |

The results in Table 14 indicate that AI-Generated forgery detection remains highly challenging for current MLLMs:

- Proprietary models struggled to provide consistent performance, with Gemini 2.5 Pro achieving the highest localization scores (IoU 47.20% and Set-based F1 54.66%) but exhibiting very low ACC (14.90%). Gemini 2.5 Flash showed a similar trend, reaching competitive localization (IoU 38.87% and Set-based F1 44.76%) while maintaining only moderate ACC (35.45%). These findings suggest that proprietary models can sometimes identify forged regions but lack robust discrimination between Authentic and AI-Generated panels.

- In contrast, open-source models demonstrate relatively stronger classification ability. InternVL3.5-8B attained the best ACC (67.00%), significantly outperforming all proprietary counterparts, though its localization quality remains weak (IoU 33.19% and Set-based F1 37.77%). Qwen2.5-VL-72B struck a more balanced profile, combining strong classification (ACC 51.06%) with reasonable localization (IoU 35.57% and Set-based F1 41.58%), highlighting the benefits of scaling open-source architectures for AI-Generated forgery detection.

Overall, the results reveal a notable discrepancy between identification and localization performance. Proprietary models excel marginally at spatial consistency checks but fail to reliably separate Authentic from AI-Generated panels, whereas large open-source models achieve stronger binary discrimination but weak spatial grounding.

## D.4 DUPLICATION

Table 15: **Experimental results on THEMIS synthetic data for Duplication.** We report the **Set-based F1**, **EM** (Exact Match), and **SPC** (Subset Partial Credit). **EM** requires strict equality with the ground truth; **SPC** awards credit for correct subsets only if no incorrect elements (false positives) are predicted. The  best  and  second-best  scores are highlighted.

| Model | Duplication | | |
|---|---|---|---|
| | **Set-based F1** | **EM** | **SPC** |
| *Proprietary MLLMs* | | | |
| GPT-5 | 33.32 | 19.05 | 22.72 |
| OpenAI o4-mini-high | 30.34 | 17.12 | 21.35 |
| Qwen-VL-Max | 23.33 | 12.13 | 15.29 |
| Gemini 2.5 Flash | 24.96 | 15.78 | 17.41 |
| Doubao-Seed-1.6-thinking | 20.22 | 14.24 | 17.08 |
| Doubao-Seed-1.6-vision | 45.13 | 17.99 | 19.95 |
| Gemini 2.5 Pro | 21.49 | 15.63 | 19.63 |
| GLM-4.5V | 21.84 | 15.73 | 20.05 |
| Claude Sonnet 4.5 | 21.86 | 15.30 | 17.98 |
| *Open-Source MLLMs* | | | |
| Qwen2.5-VL-72B | 16.75 | 11.54 | 14.35 |
| InternVL3.5-8B | 33.28 | 10.34 | 10.89 |
| Llama 4 Maverick | 19.01 | 13.90 | 16.07 |
| LLaVA-Interleave-7B-DPO | 8.01 | 0.33 | 0.33 |
| LLaVA-NeXT-34B | 10.73 | 0.11 | 0.11 |
| Qwen2.5-VL-32B | 15.26 | 14.28 | 14.95 |
| Gemma 3 27B | 12.20 | 9.57 | 10.35 |

Table 15 reports results on Duplication. Overall, performance across all models is markedly weaker compared to other fraud types, underscoring the intrinsic difficulty of duplication forensics:

- Among proprietary models, Doubao-Seed-1.6-vision achieved the highest Set-based F1 (45.13%), suggesting relatively stronger discriminative ability, but its EM (17.99%) and SPC (19.95%) remained low, indicating limited capability in producing strictly correct multi-label predictions. GPT-5 attained the second-best Set-based F1 (33.32%) and led in SPC (22.72%), while also yielding the best EM (19.05%), showing a modest advantage in producing fully correct predictions. Nevertheless, even these leading scores are far from satisfactory, reflecting the challenge of simultaneously identifying multiple manipulation operations with high precision.

- In contrast, open-source models exhibit consistently poor performance. Qwen2.5-VL-32B and Llama 4 Maverick achieved slightly higher EM (14.28%/13.90%) and SPC (14.95%/16.07%) compared to other open-source MLLMs, but remained substantially behind proprietary models. LLaVA-Interleave-7B-DPO and LLaVA-NeXT-34B failed almost entirely, with EM and SPC close to zero, suggesting limited capability in reasoning about multiple manipulation operations.

Taken together, these results highlight Duplication as the hardest subtask within THEMIS, where neither proprietary nor open-source models show satisfactory performance. The stark contrast with other fraud types reveals that detecting subtle geometric transformations and parameter modifications across panels requires fine-grained relational reasoning and robust representation of geometric and parameter variations. Addressing Duplication thus remains a critical bottleneck for advancing visual fraud reasoning with MLLMs.

## D.5 Text–Image Inconsistency

Table 16: **Experimental results on THEMIS synthetic data for Text–Image Inconsistency.** We report the **ACC**, **Macro-F1**, and **Text-based F1**. The best and second-best scores are highlighted.

| Model | Text–Image Inconsistency | | |
| --- | --- | --- | --- |
| | ACC | Macro-F1 | Text-based F1 |
| *Proprietary MLLMs* | | | |
| GPT-5 | 60.67 | 44.95 | 27.44 |
| OpenAI o4-mini-high | 66.33 | 48.58 | 32.22 |
| Qwen-VL-Max | 56.00 | 33.74 | 15.36 |
| Gemini 2.5 Flash | 36.33 | 25.30 | 28.24 |
| Doubao-Seed-1.6-thinking | 60.00 | 44.56 | 31.71 |
| Doubao-Seed-1.6-vision | 37.00 | 27.98 | 30.43 |
| Gemini 2.5 Pro | 44.67 | 31.42 | 37.31 |
| GLM-4.5V | 53.67 | 39.83 | 22.69 |
| Claude Sonnet 4.5 | 35.67 | 27.33 | 27.42 |
| *Open-Source MLLMs* | | | |
| Qwen2.5-VL-72B | 61.33 | 36.51 | 12.32 |
| InternVL3.5-8B | 55.00 | 25.61 | 4.78 |
| Llama 4 Maverick | 54.00 | 33.95 | 16.08 |
| LLaVA-Interleave-7B-DPO | 50.00 | 16.78 | 12.57 |
| LLaVA-NeXT-34B | 41.33 | 24.74 | 4.28 |
| Qwen2.5-VL-32B | 58.33 | 38.71 | 17.94 |
| Gemma 3 27B | 31.67 | 16.91 | 21.41 |

Table 16 summarizes the results on Text–Image Inconsistency, evaluated with ACC, Macro-F1, and Text-based F1:

- Compared to other fraud types, this subtask poses unique challenges as it requires models to integrate textual semantics with fine-grained visual reasoning. Proprietary models showed stronger overall performance, with OpenAI o4-mini-high achieving the highest ACC (66.33%) and Macro-F1 (48.58%), suggesting robust alignment between textual cues and image content. GPT-5 followed closely, ranking second in Macro-F1 (44.95%) while also maintaining competitive ACC (60.67%). Notably, Gemini 2.5 Pro achieved the best Text-based F1 (37.31%), indicating a relative advantage in grounding inconsistencies to specific regions, although its overall ACC (44.67%) lags behind leading models.

- On the open-source side, Qwen2.5-VL-72B performed competitively in terms of ACC (61.33%), nearly matching proprietary leaders, but its Macro-F1 (36.51%) and Text-based F1 (12.32%) were substantially weaker. This discrepancy highlights a gap between classification accuracy and balanced, fine-grained reasoning across categories. Other open-source MLLMs such as LLaVA-NeXT-34B exhibited consistently low performance, with Macro-F1 and Text-based F1 dropping below 25% and 5%, respectively, reflecting their limited capacity for multimodal semantic consistency reasoning.

Overall, the results reveal that while certain proprietary MLLMs (e.g., OpenAI o4-mini-high and GPT-5) demonstrate promising capability in detecting Text–Image Inconsistency, the subtask remains far from solved. The relatively low Text-based F1 scores across all models indicate that fine-grained grounding of inconsistencies is especially challenging, underscoring the need for future models to better integrate visual fraud reasoning with spatial localization.

# E  CASE STUDY

We provide 22 representative cases selected from the outputs of GPT-5, our best-performing model, to comprehensively cover the full scope of THEMIS. These cases span all seven academic scenarios (Chart, Diagram, Micrograph, Stained Micrograph, Physical Object, Medical Imaging, and Others) and are systematically organized across the five fraud types: Splicing, Copy-Move, AI-Generated (Image Inference Forgery/Targeted Region Editing), Duplication (Global/Local), and Text–Image Inconsistency (Numerical/Trend).

## E.1  SPLICING: CHART (SUCCESS CASE)

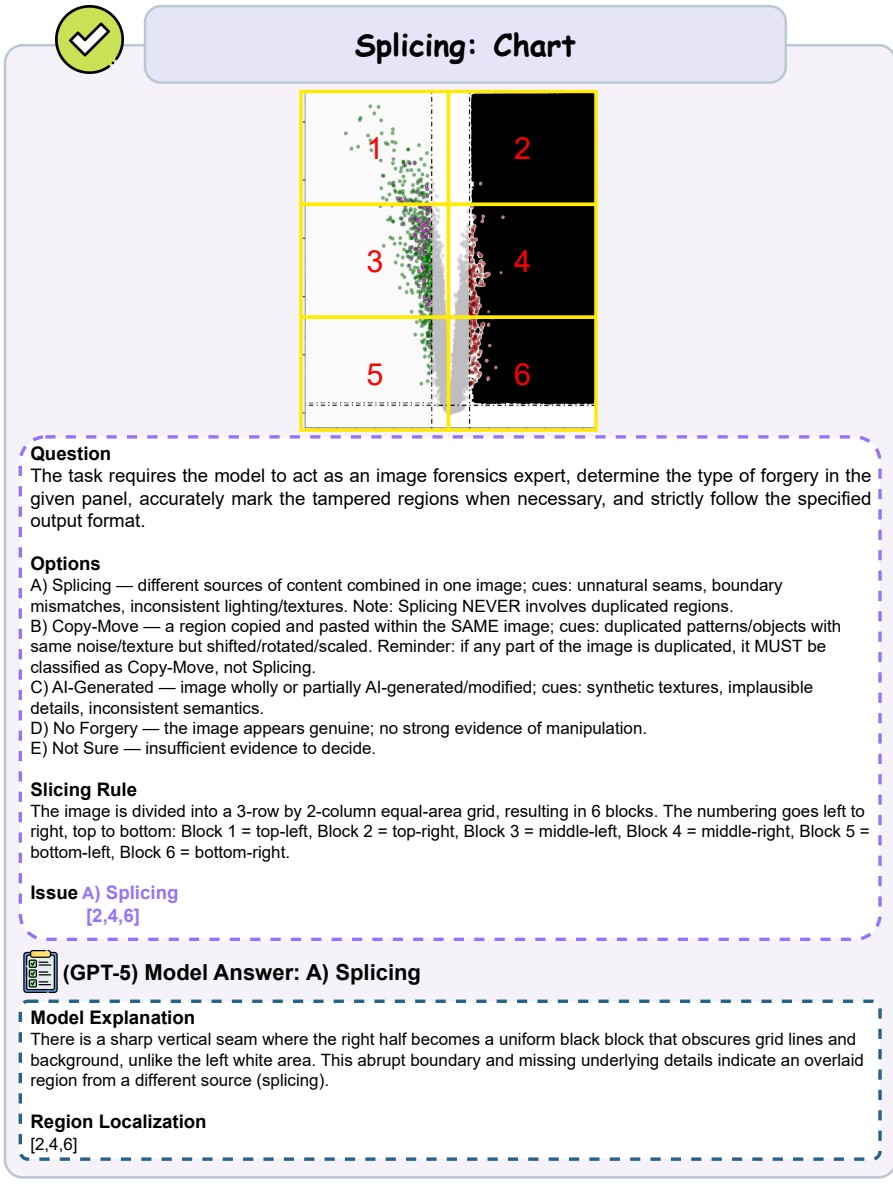

Figure 8: **Success case by GPT-5 on Splicing.**

E.2   SPLICING: MEDICAL IMAGING (FAILURE CASE)

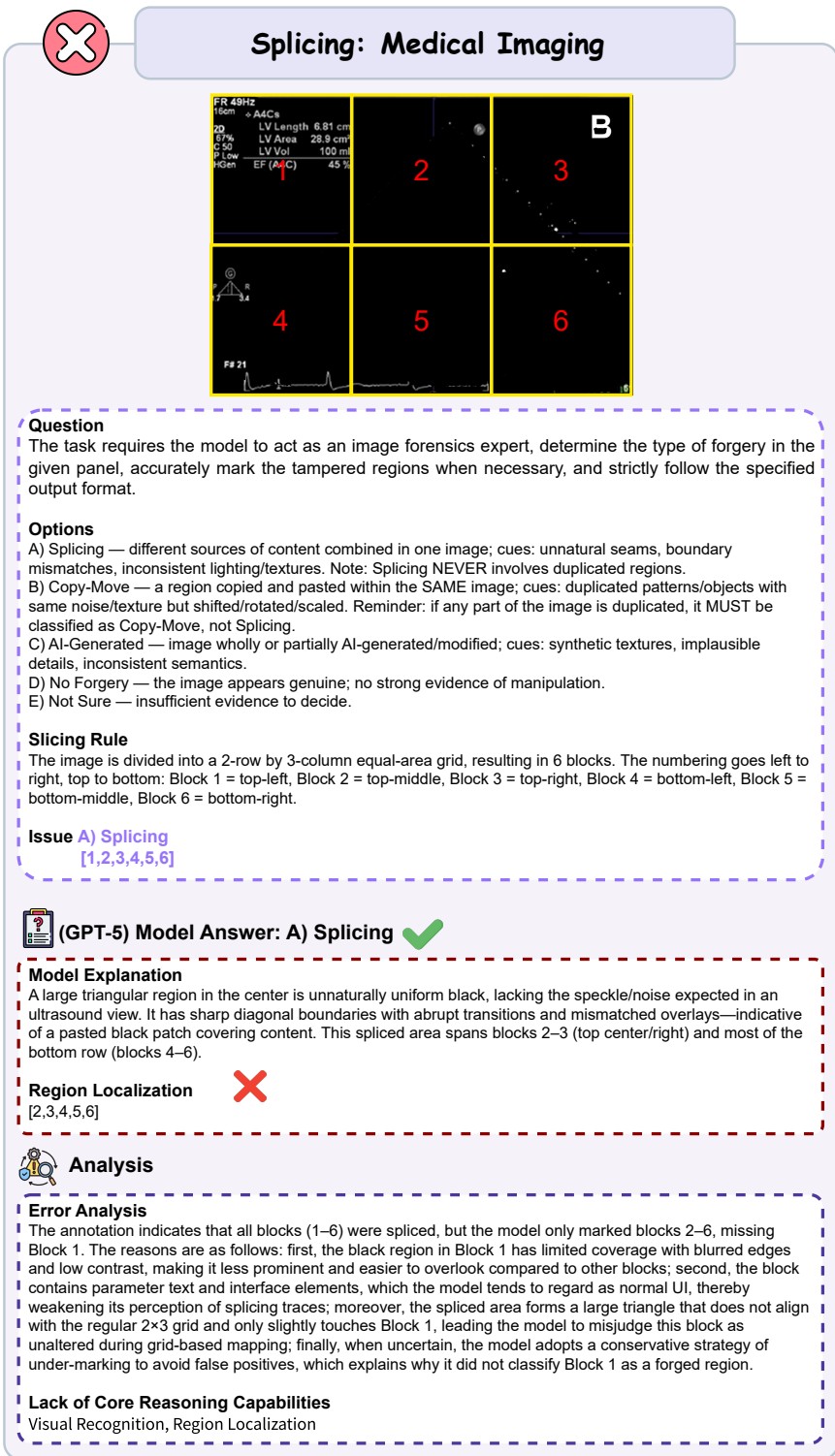

Figure 9: **Failure case by GPT-5 on Splicing.** The model correctly identified the forgery type but mislocalized it.

E.3  SPLICING: CHART (FAILURE CASE)

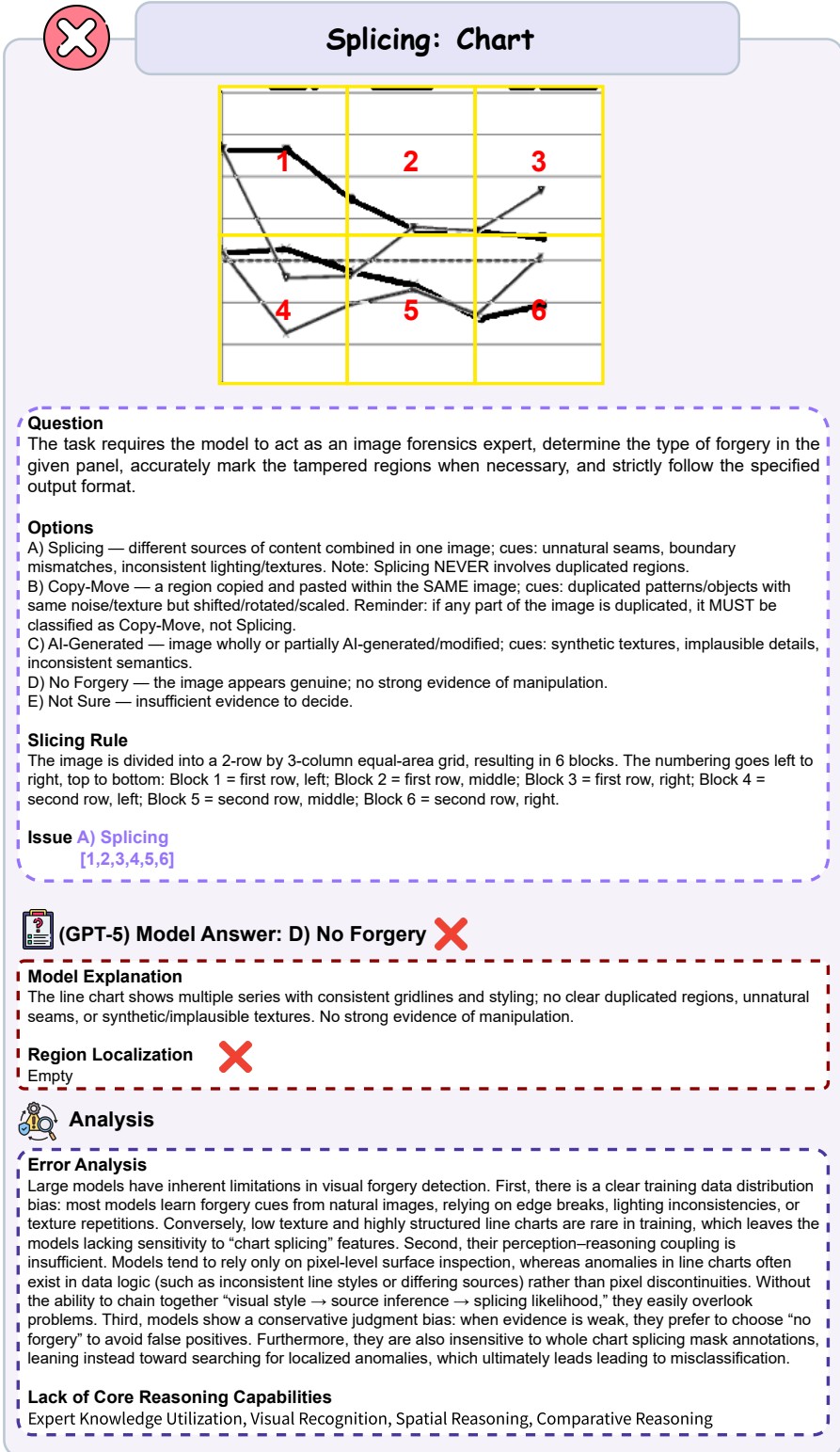

Figure 10: **Failure case by GPT-5 on Splicing.** The model misidentified and mislocalized.

### E.4   COPY-MOVE: STAINED MICROGRAPH (SUCCESS CASE)

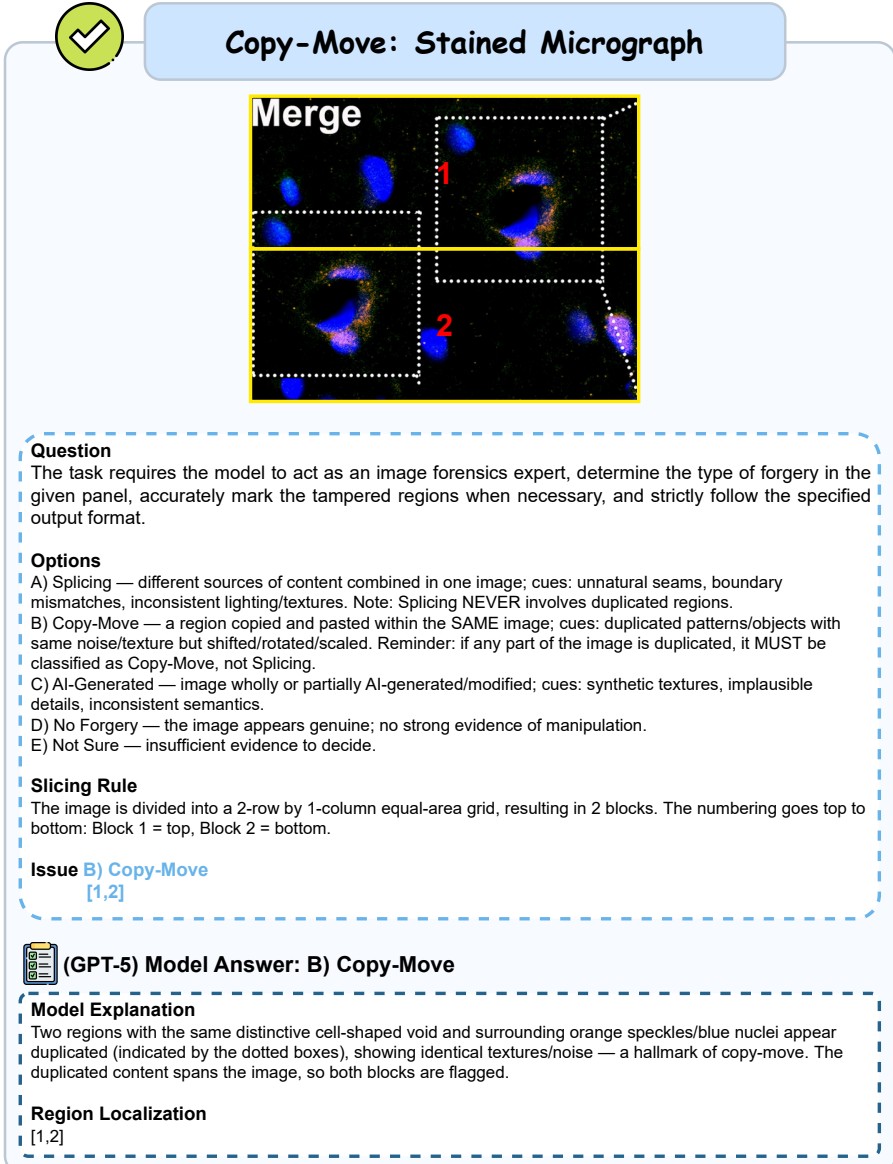

**Copy-Move: Stained Micrograph**

**Question**
The task requires the model to act as an image forensics expert, determine the type of forgery in the given panel, accurately mark the tampered regions when necessary, and strictly follow the specified output format.

**Options**
A) Splicing — different sources of content combined in one image; cues: unnatural seams, boundary mismatches, inconsistent lighting/textures. Note: Splicing NEVER involves duplicated regions.
B) Copy-Move — a region copied and pasted within the SAME image; cues: duplicated patterns/objects with same noise/texture but shifted/rotated/scaled. Reminder: if any part of the image is duplicated, it MUST be classified as Copy-Move, not Splicing.
C) AI-Generated — image wholly or partially AI-generated/modified; cues: synthetic textures, implausible details, inconsistent semantics.
D) No Forgery — the image appears genuine; no strong evidence of manipulation.
E) Not Sure — insufficient evidence to decide.

**Slicing Rule**
The image is divided into a 2-row by 1-column equal-area grid, resulting in 2 blocks. The numbering goes top to bottom: Block 1 = top, Block 2 = bottom.

**Issue** B) Copy-Move
[1,2]

**(GPT-5) Model Answer: B) Copy-Move**

**Model Explanation**
Two regions with the same distinctive cell-shaped void and surrounding orange speckles/blue nuclei appear duplicated (indicated by the dotted boxes), showing identical textures/noise — a hallmark of copy-move. The duplicated content spans the image, so both blocks are flagged.

**Region Localization**
[1,2]

Figure 11: **Success case by GPT-5 on Copy-Move.**

E.5   COPY-MOVE: STAINED MICROGRAPH (FAILURE CASE 1)

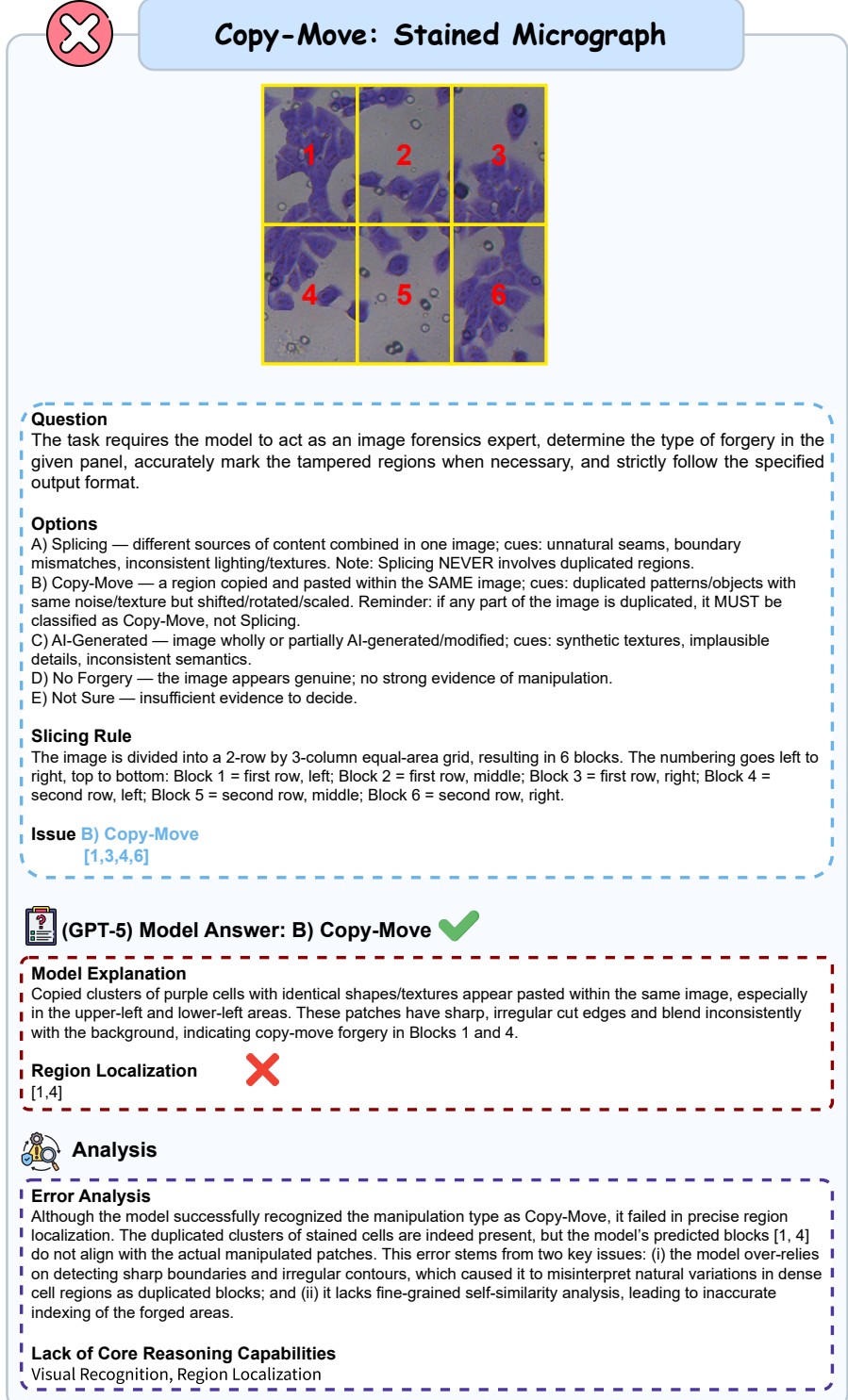

Figure 12: **Failure case by GPT-5 on Copy-Move.** The model correctly identified the forgery type but mislocalized it.

### E.6 COPY-MOVE: STAINED MICROGRAPH (FAILURE CASE 2)

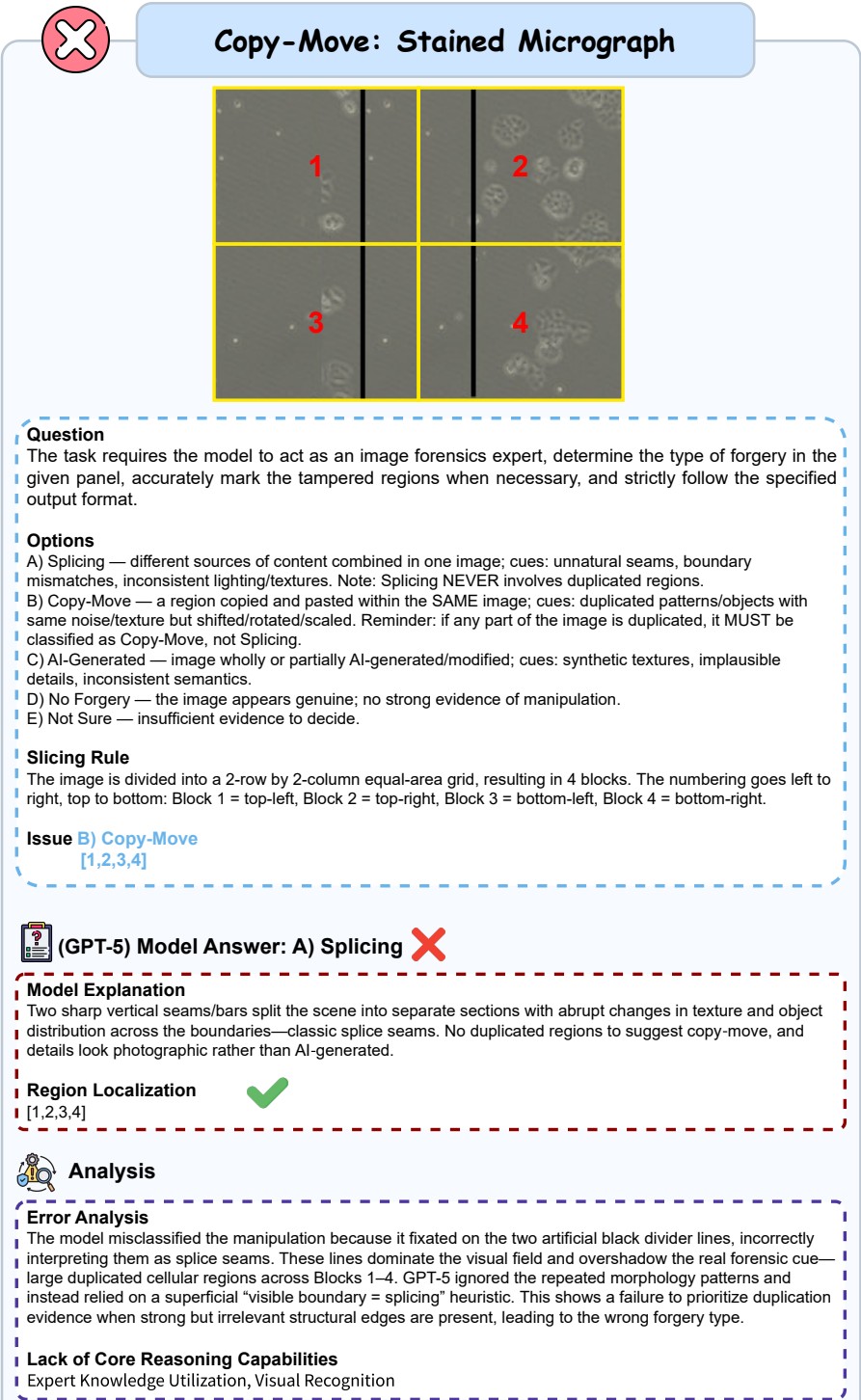

Figure 13: **Failure case by GPT-5 on Copy-Move.** The model correctly localized the forgery region but misidentified it.

## E.7 COPY-MOVE: STAINED MICROGRAPH (FAILURE CASE 3)

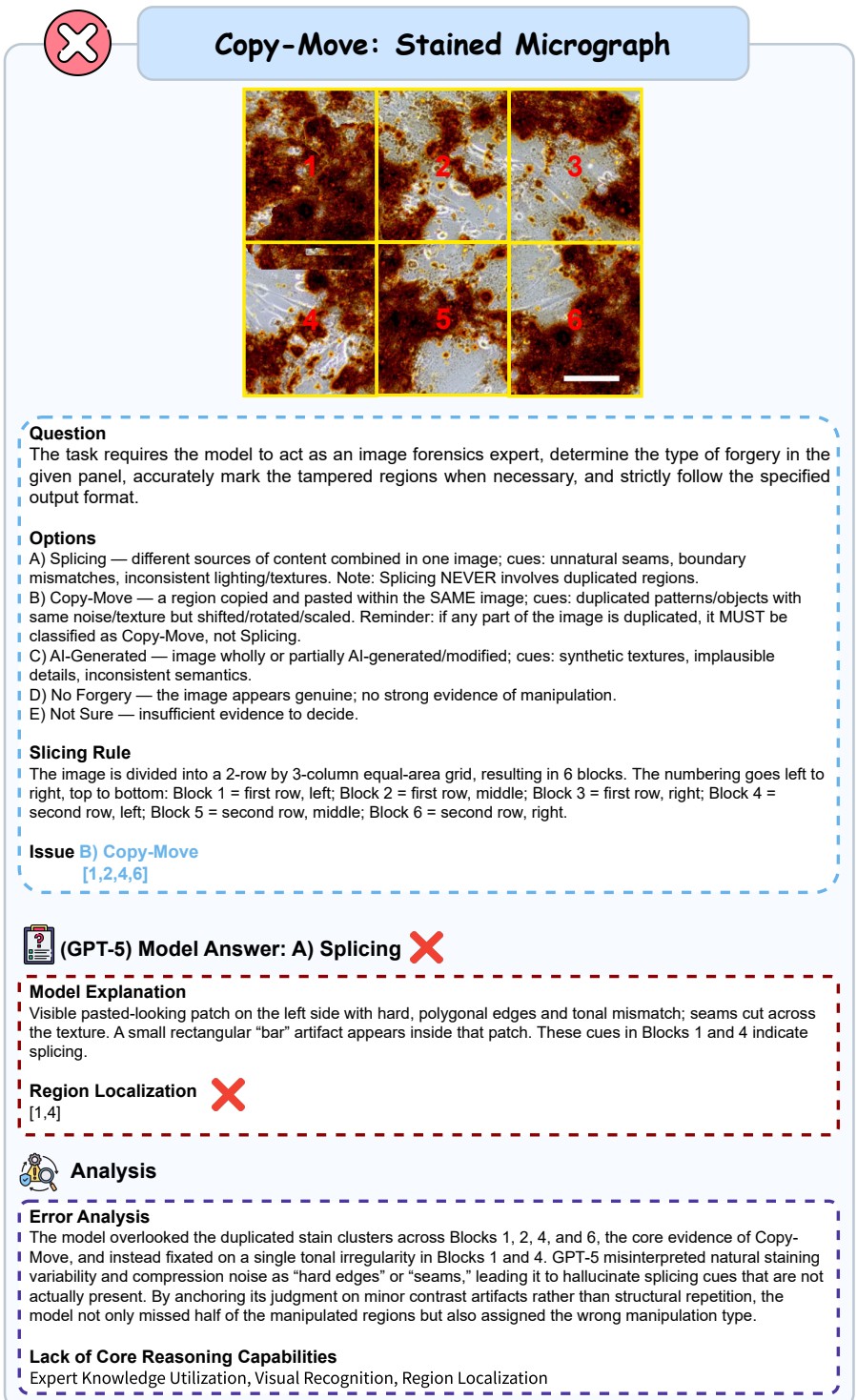

Figure 14: **Failure case by GPT-5 on Copy-Move.** The model misidentified and mislocalized.

## E.8  AI-GENERATED: MEDICAL IMAGING (SUCCESS CASE)

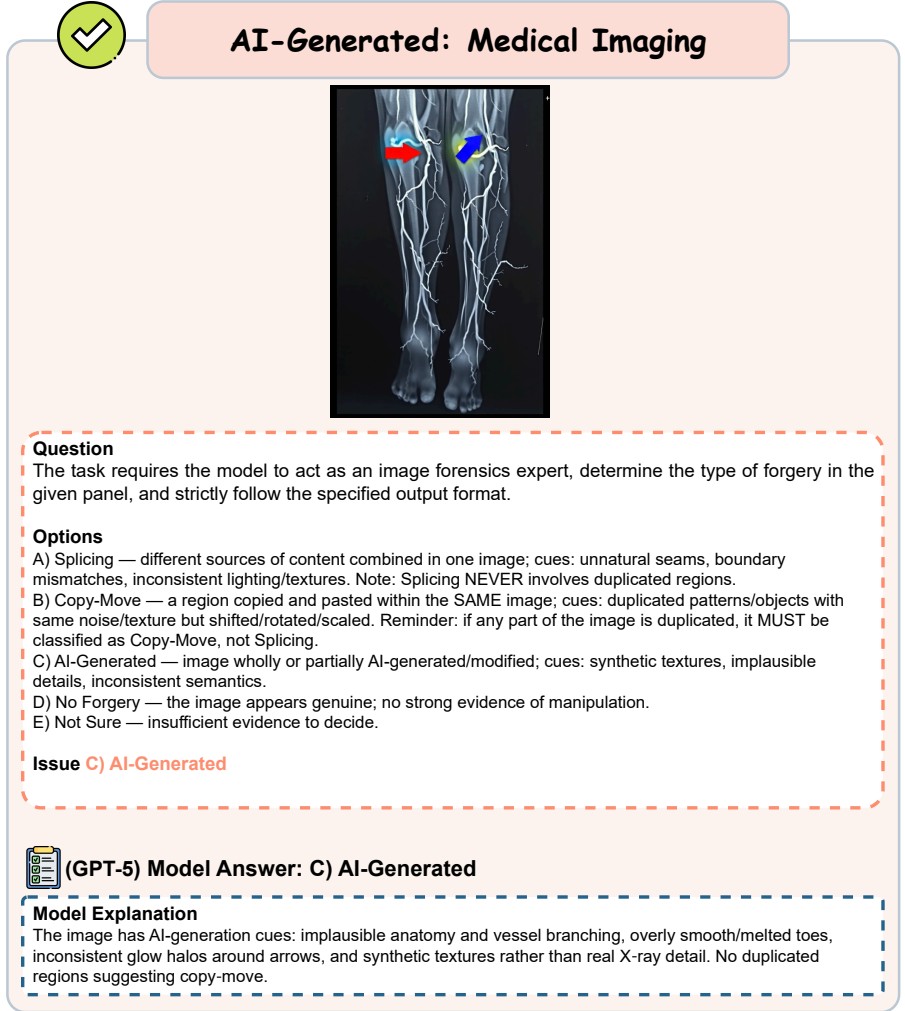

**AI-Generated: Medical Imaging**

**Question**
The task requires the model to act as an image forensics expert, determine the type of forgery in the given panel, and strictly follow the specified output format.

**Options**
A) Splicing — different sources of content combined in one image; cues: unnatural seams, boundary mismatches, inconsistent lighting/textures. Note: Splicing NEVER involves duplicated regions.
B) Copy-Move — a region copied and pasted within the SAME image; cues: duplicated patterns/objects with same noise/texture but shifted/rotated/scaled. Reminder: if any part of the image is duplicated, it MUST be classified as Copy-Move, not Splicing.
C) AI-Generated — image wholly or partially AI-generated/modified; cues: synthetic textures, implausible details, inconsistent semantics.
D) No Forgery — the image appears genuine; no strong evidence of manipulation.
E) Not Sure — insufficient evidence to decide.

**Issue C) AI-Generated**

**(GPT-5) Model Answer: C) AI-Generated**

**Model Explanation**
The image has AI-generation cues: implausible anatomy and vessel branching, overly smooth/melted toes, inconsistent glow halos around arrows, and synthetic textures rather than real X-ray detail. No duplicated regions suggesting copy-move.

Figure 15: **Success case by GPT-5 on AI-Generated.**  The panel is entirely forged via Image Inference Forgery.

### E.9   AI-Generated: Micrograph (Failure Case)

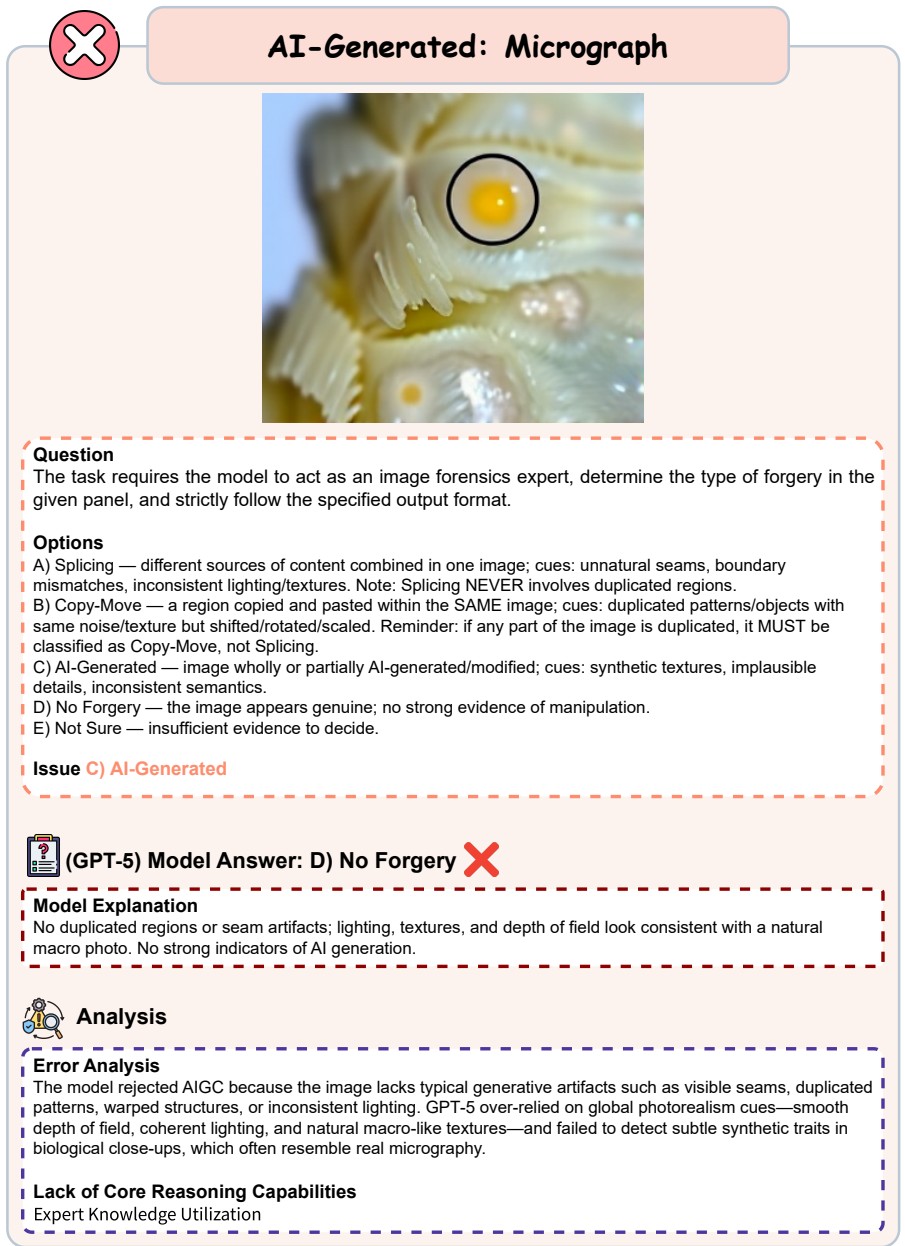

Figure 16: **Failure case by GPT-5 on AI-Generated.** The model misidentified the forgery type. The panel is entirely forged via Image Inference Forgery.

## E.10 AI-GENERATED: DIAGRAM (SUCCESS CASE)

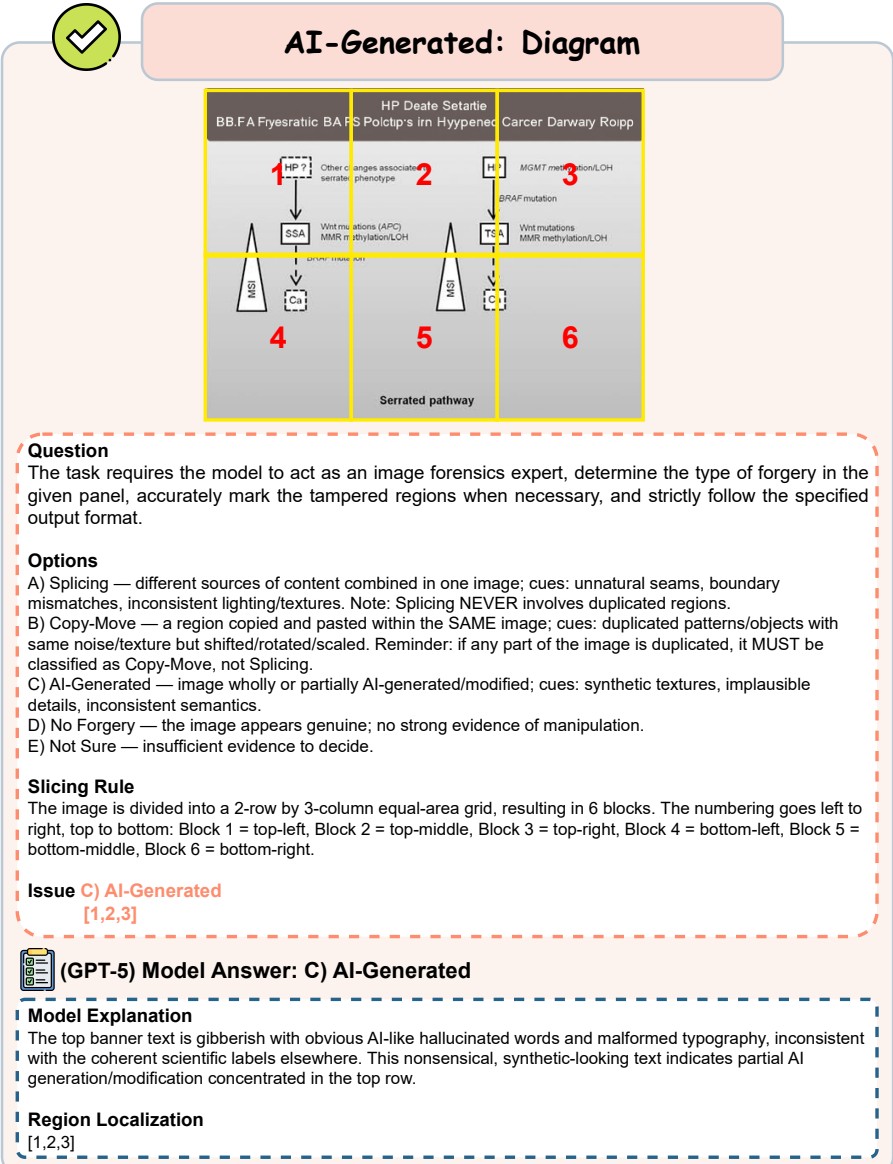

**Question**
The task requires the model to act as an image forensics expert, determine the type of forgery in the given panel, accurately mark the tampered regions when necessary, and strictly follow the specified output format.

**Options**
A) Splicing — different sources of content combined in one image; cues: unnatural seams, boundary mismatches, inconsistent lighting/textures. Note: Splicing NEVER involves duplicated regions.
B) Copy-Move — a region copied and pasted within the SAME image; cues: duplicated patterns/objects with same noise/texture but shifted/rotated/scaled. Reminder: if any part of the image is duplicated, it MUST be classified as Copy-Move, not Splicing.
C) AI-Generated — image wholly or partially AI-generated/modified; cues: synthetic textures, implausible details, inconsistent semantics.
D) No Forgery — the image appears genuine; no strong evidence of manipulation.
E) Not Sure — insufficient evidence to decide.

**Slicing Rule**
The image is divided into a 2-row by 3-column equal-area grid, resulting in 6 blocks. The numbering goes left to right, top to bottom: Block 1 = top-left, Block 2 = top-middle, Block 3 = top-right, Block 4 = bottom-left, Block 5 = bottom-middle, Block 6 = bottom-right.

**Issue** C) AI-Generated
[1,2,3]

**(GPT-5) Model Answer: C) AI-Generated**

**Model Explanation**
The top banner text is gibberish with obvious AI-like hallucinated words and malformed typography, inconsistent with the coherent scientific labels elsewhere. This nonsensical, synthetic-looking text indicates partial AI generation/modification concentrated in the top row.

**Region Localization**
[1,2,3]

Figure 17: **Success case by GPT-5 on AI-Generated.** The panel is partially forged via Targeted Region Editing.

## E.11 AI-GENERATED: CHART (FAILURE CASE 1)

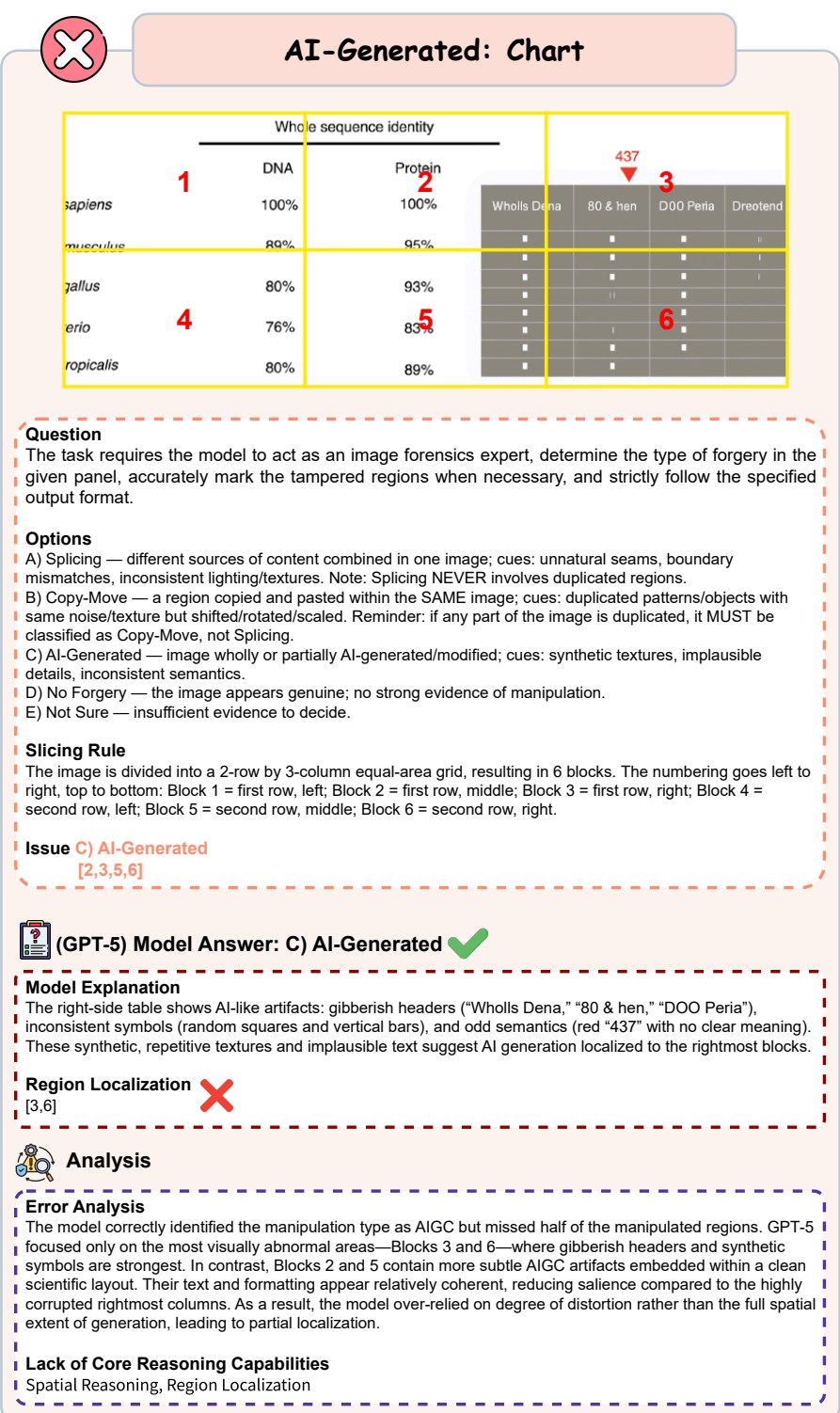

Figure 18: **Failure case by GPT-5 on AI-Generated.** The model correctly identified the forgery type but mislocalized it. The panel is partially forged via Targeted Region Editing.

## E.12 AI-GENERATED: PHYSICAL OBJECT (FAILURE CASE)

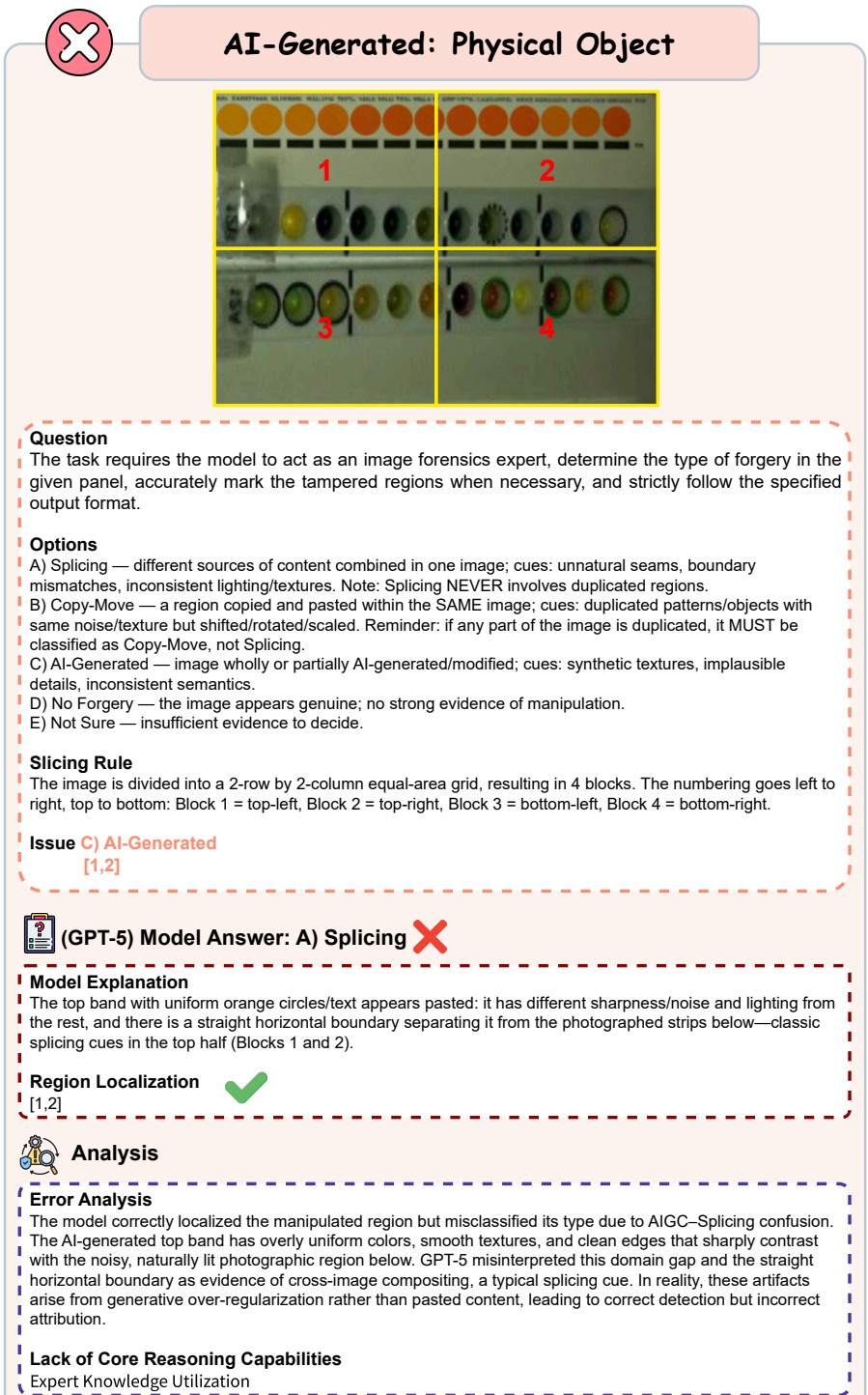

Figure 19: **Failure case by GPT-5 on AI-Generated.** The model correctly localized the forgery region but misidentified it. The panel is partially forged via Targeted Region Editing.

## E.13 AI-Generated: Chart (Failure Case 2)

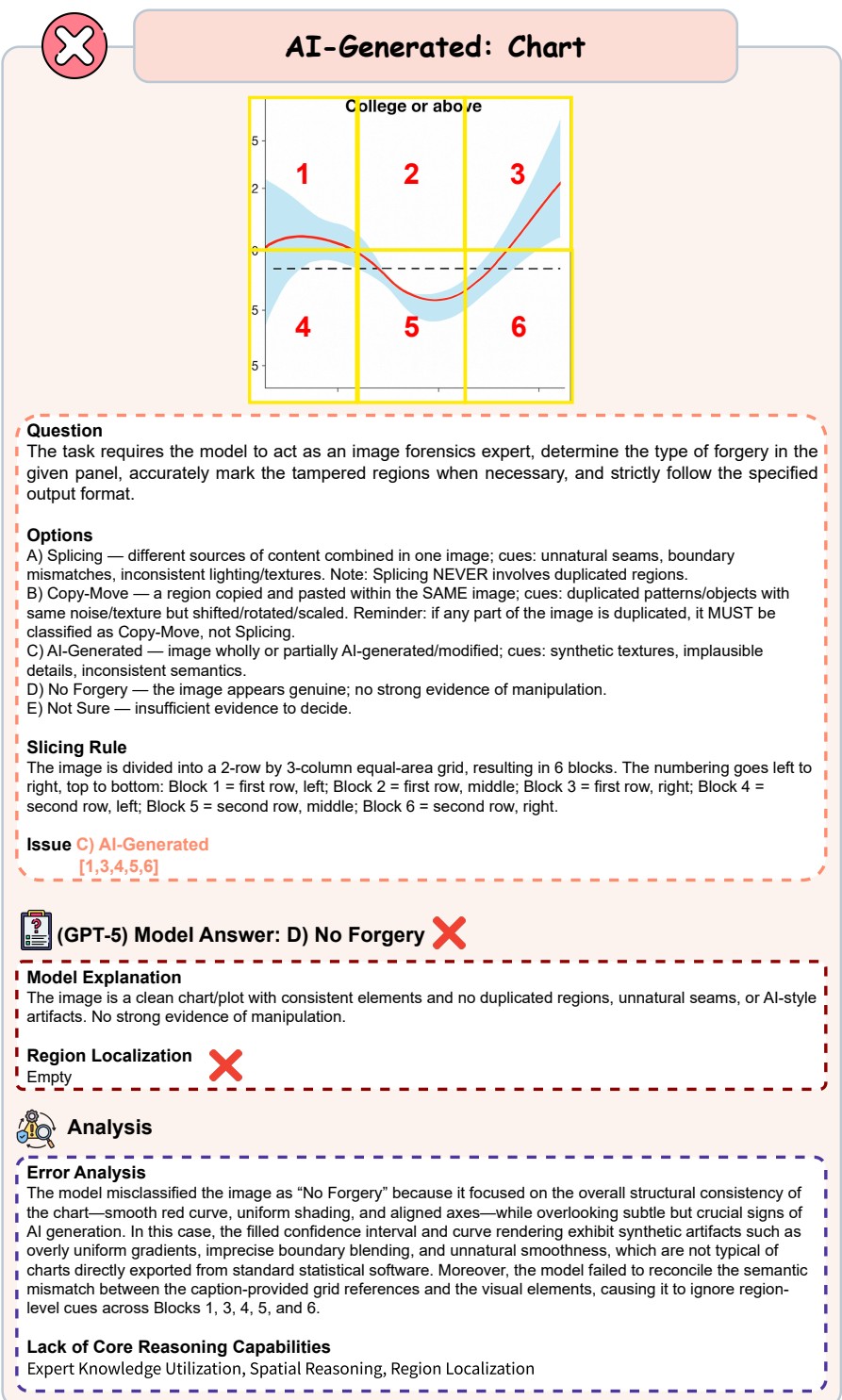

Figure 20: **Failure case by GPT-5 on AI-Generated.** The model misidentified and mislocalized. The panel is partially forged via Targeted Region Editing.

## E.14 DUPLICATION: MICROGRAPH (GLOBAL – SUCCESS CASE)

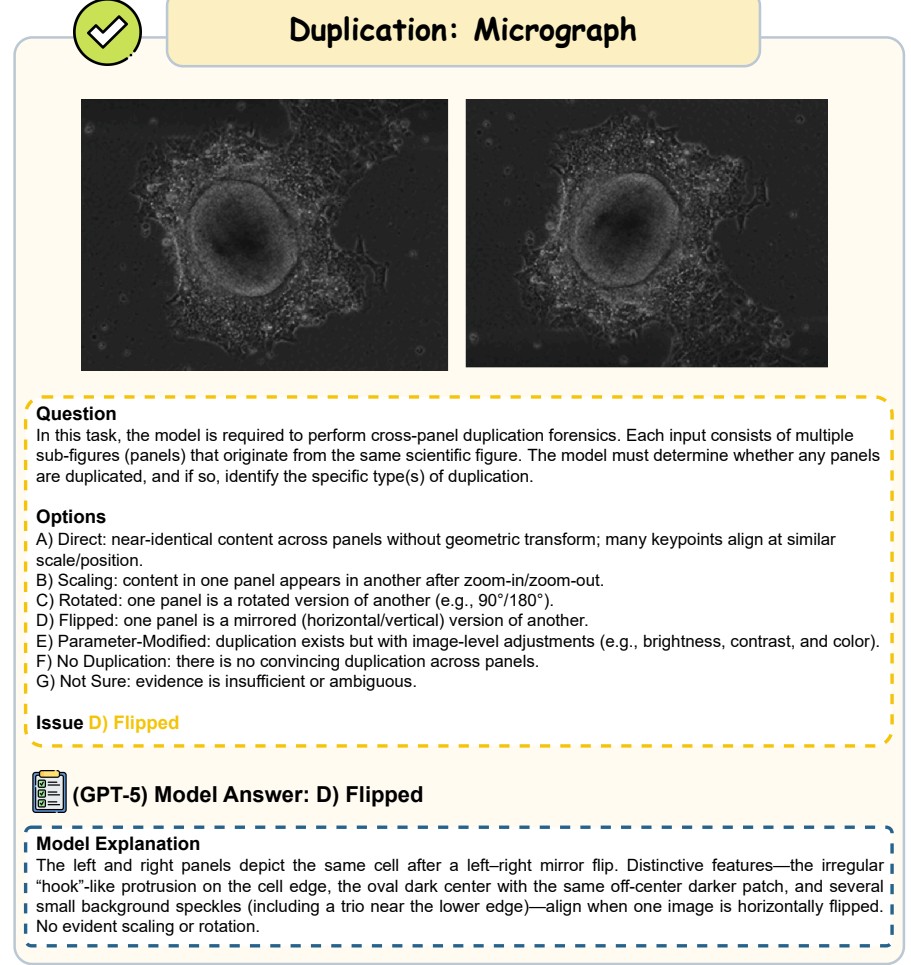

Figure 21: **Success case by GPT-5 on Duplication.** These panels resulted from Global Duplication with flip.

## E.15 DUPLICATION: STAINED MICROGRAPH (GLOBAL – FAILURE CASE)

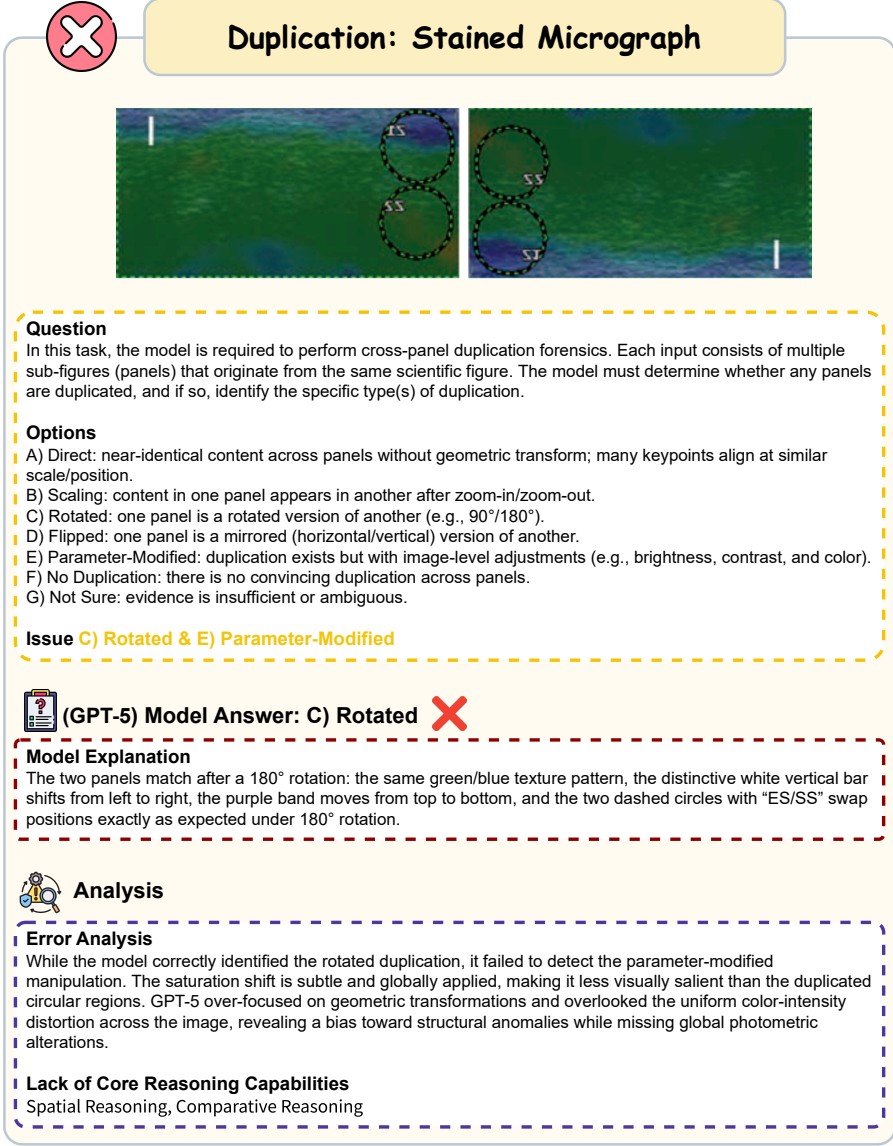

Figure 22: **Failure case by GPT-5 on Duplication.** These panels resulted from Global Duplication with rotation and parameter modification.

### E.16 DUPLICATION: STAINED MICROGRAPH (LOCAL – FAILURE CASE 1)

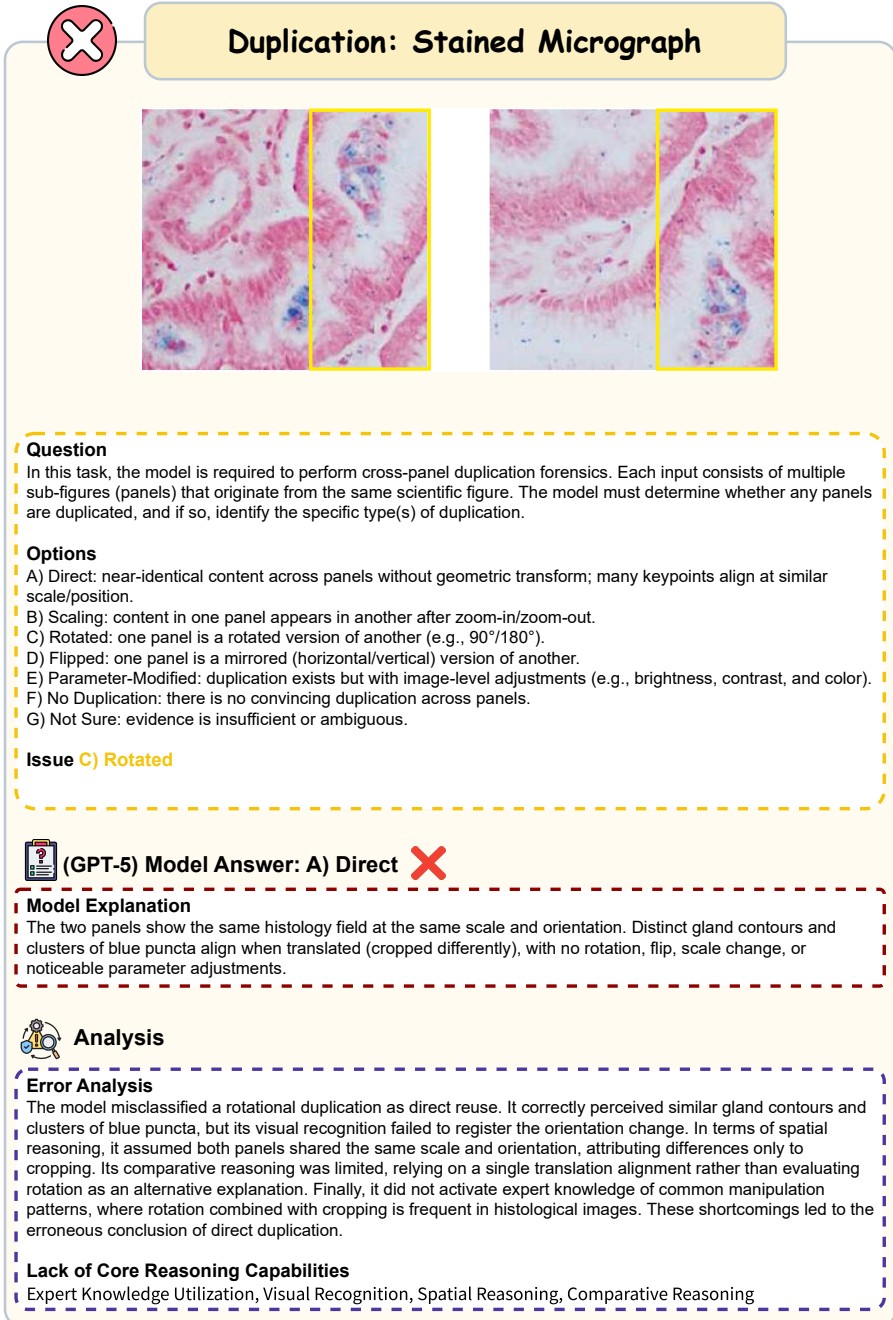

Figure 23: **Failure case by GPT-5 on Duplication.** These panels resulted from Local Duplication with rotation. The duplicated region is indicated by the yellow boxes.

E.17 DUPLICATION: STAINED MICROGRAPH (LOCAL – FAILURE CASE 2)

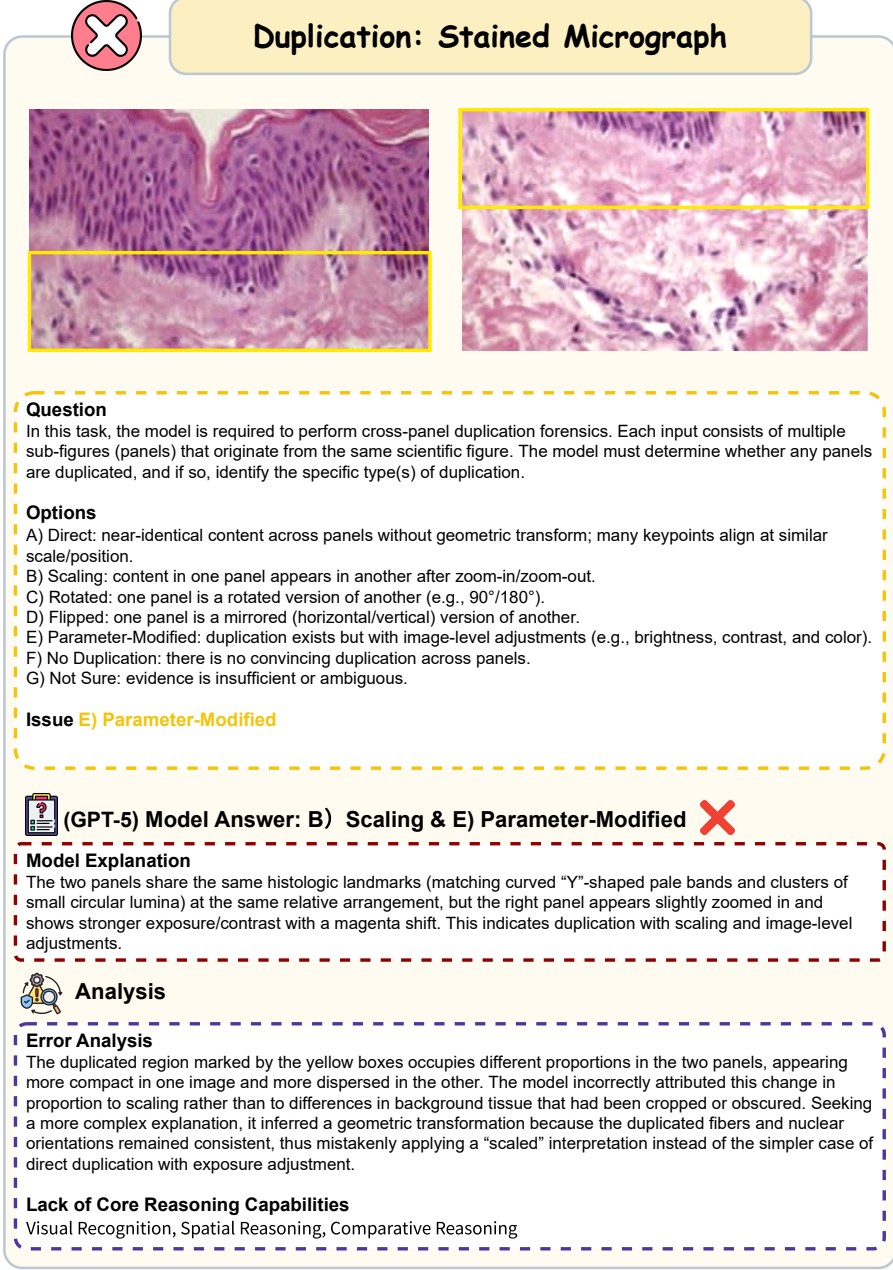

Figure 24: **Failure case by GPT-5 on Duplication.** These panels resulted from Local Duplication with parameter modification. The duplicated region is indicated by the yellow boxes.

## E.18 TEXT–IMAGE INCONSISTENCY: OTHERS (NUMERICAL – SUCCESS CASE)

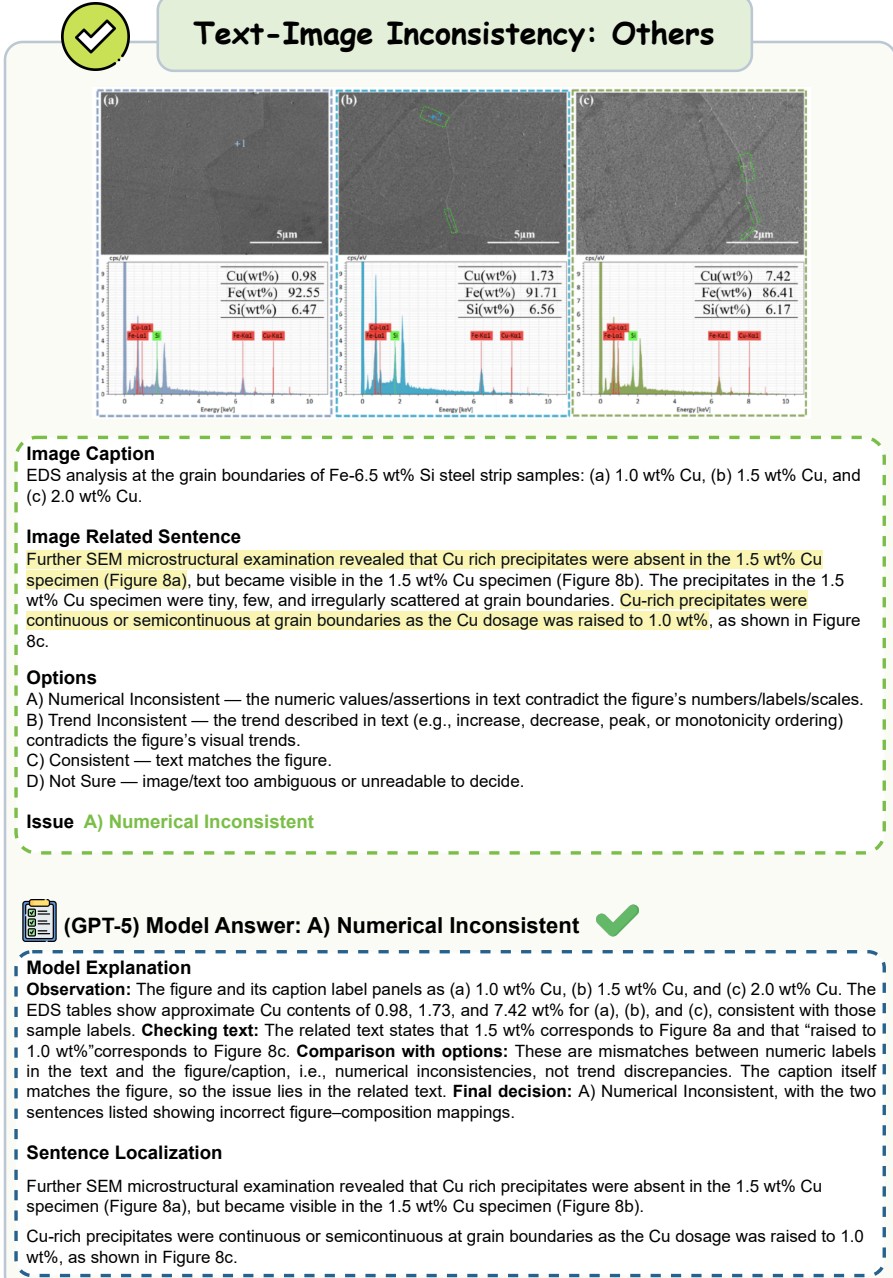

**Text-Image Inconsistency: Others**

**Image Caption**
EDS analysis at the grain boundaries of Fe-6.5 wt% Si steel strip samples: (a) 1.0 wt% Cu, (b) 1.5 wt% Cu, and (c) 2.0 wt% Cu.

**Image Related Sentence**
Further SEM microstructural examination revealed that Cu rich precipitates were absent in the 1.5 wt% Cu specimen (Figure 8a), but became visible in the 1.5 wt% Cu specimen (Figure 8b). The precipitates in the 1.5 wt% Cu specimen were tiny, few, and irregularly scattered at grain boundaries. Cu-rich precipitates were continuous or semicontinuous at grain boundaries as the Cu dosage was raised to 1.0 wt%, as shown in Figure 8c.

**Options**
A) Numerical Inconsistent — the numeric values/assertions in text contradict the figure's numbers/labels/scales.
B) Trend Inconsistent — the trend described in text (e.g., increase, decrease, peak, or monotonicity ordering) contradicts the figure's visual trends.
C) Consistent — text matches the figure.
D) Not Sure — image/text too ambiguous or unreadable to decide.

**Issue  A) Numerical Inconsistent**

---

**(GPT-5) Model Answer: A) Numerical Inconsistent** ✅

**Model Explanation**
**Observation:** The figure and its caption label panels as (a) 1.0 wt% Cu, (b) 1.5 wt% Cu, and (c) 2.0 wt% Cu. The EDS tables show approximate Cu contents of 0.98, 1.73, and 7.42 wt% for (a), (b), and (c), consistent with those sample labels. **Checking text:** The related text states that 1.5 wt% corresponds to Figure 8a and that "raised to 1.0 wt%"corresponds to Figure 8c. **Comparison with options:** These are mismatches between numeric labels in the text and the figure/caption, i.e., numerical inconsistencies, not trend discrepancies. The caption itself matches the figure, so the issue lies in the related text. **Final decision:** A) Numerical Inconsistent, with the two sentences listed showing incorrect figure–composition mappings.

**Sentence Localization**

Further SEM microstructural examination revealed that Cu rich precipitates were absent in the 1.5 wt% Cu specimen (Figure 8a), but became visible in the 1.5 wt% Cu specimen (Figure 8b).

Cu-rich precipitates were continuous or semicontinuous at grain boundaries as the Cu dosage was raised to 1.0 wt%, as shown in Figure 8c.

Figure 25: **Success case by GPT-5 on Text–Image Numerical Inconsistency.**

## E.19 Text–Image Inconsistency: Others (Numerical – Failure Case 1)

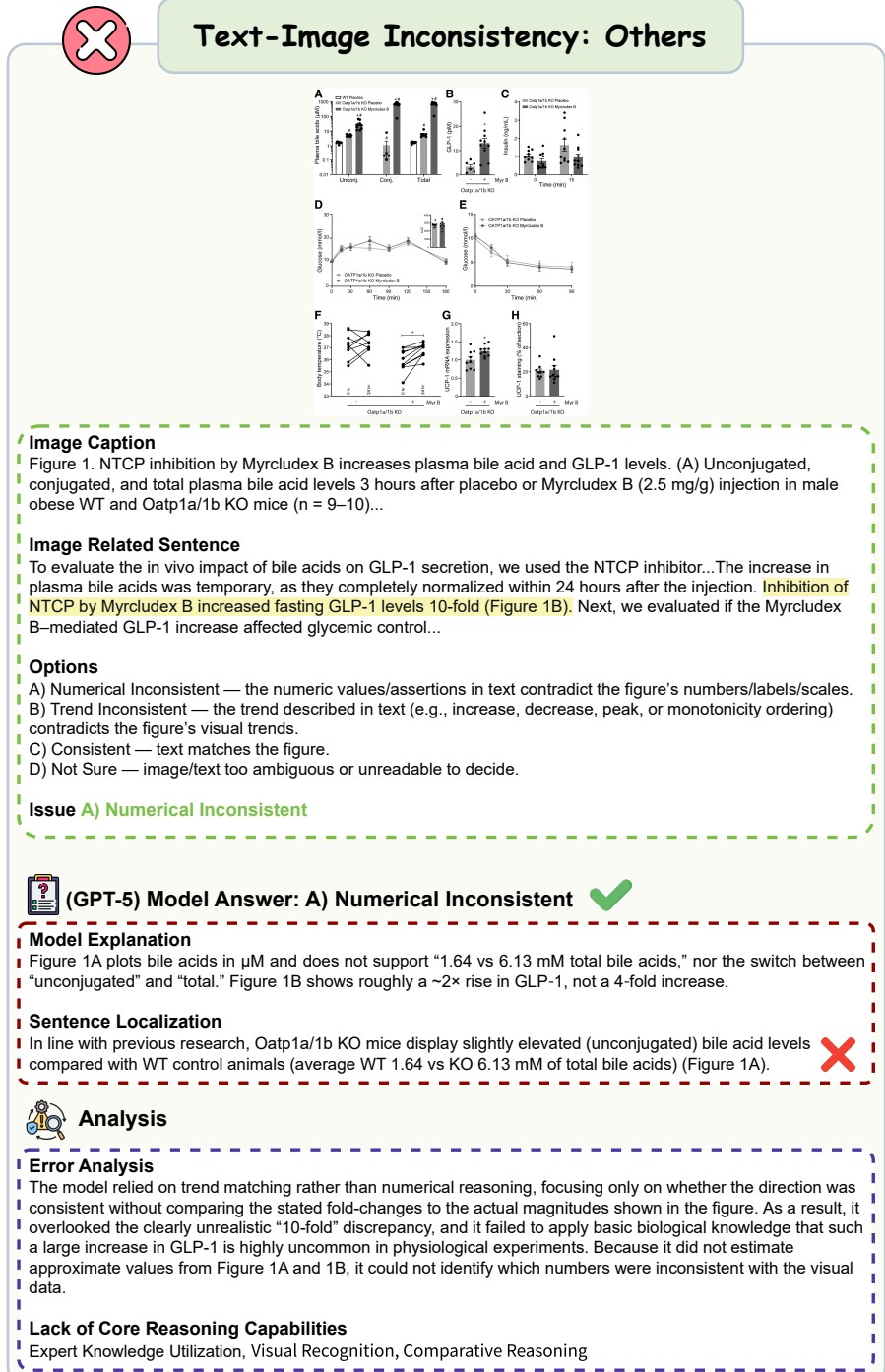

**Text-Image Inconsistency: Others**

**Image Caption**
Figure 1. NTCP inhibition by Myrcludex B increases plasma bile acid and GLP-1 levels. (A) Unconjugated, conjugated, and total plasma bile acid levels 3 hours after placebo or Myrcludex B (2.5 mg/g) injection in male obese WT and Oatp1a/1b KO mice (n = 9–10)...

**Image Related Sentence**
To evaluate the in vivo impact of bile acids on GLP-1 secretion, we used the NTCP inhibitor...The increase in plasma bile acids was temporary, as they completely normalized within 24 hours after the injection. Inhibition of NTCP by Myrcludex B increased fasting GLP-1 levels 10-fold (Figure 1B). Next, we evaluated if the Myrcludex B–mediated GLP-1 increase affected glycemic control...

**Options**
A) Numerical Inconsistent — the numeric values/assertions in text contradict the figure's numbers/labels/scales.
B) Trend Inconsistent — the trend described in text (e.g., increase, decrease, peak, or monotonicity ordering) contradicts the figure's visual trends.
C) Consistent — text matches the figure.
D) Not Sure — image/text too ambiguous or unreadable to decide.

**Issue A) Numerical Inconsistent**

📋 **(GPT-5) Model Answer: A) Numerical Inconsistent** ✅

**Model Explanation**
Figure 1A plots bile acids in μM and does not support "1.64 vs 6.13 mM total bile acids," nor the switch between "unconjugated" and "total." Figure 1B shows roughly a ~2× rise in GLP-1, not a 4-fold increase.

**Sentence Localization**
In line with previous research, Oatp1a/1b KO mice display slightly elevated (unconjugated) bile acid levels compared with WT control animals (average WT 1.64 vs KO 6.13 mM of total bile acids) (Figure 1A). ❌

🔍 **Analysis**

**Error Analysis**
The model relied on trend matching rather than numerical reasoning, focusing only on whether the direction was consistent without comparing the stated fold-changes to the actual magnitudes shown in the figure. As a result, it overlooked the clearly unrealistic "10-fold" discrepancy, and it failed to apply basic biological knowledge that such a large increase in GLP-1 is highly uncommon in physiological experiments. Because it did not estimate approximate values from Figure 1A and 1B, it could not identify which numbers were inconsistent with the visual data.

**Lack of Core Reasoning Capabilities**
Expert Knowledge Utilization, Visual Recognition, Comparative Reasoning

Figure 26: **Failure case by GPT-5 on Text–Image Numerical Inconsistency.** The model identified but mislocalized.

E.20   TEXT–IMAGE INCONSISTENCY: OTHERS (NUMERICAL – FAILURE CASE 2)

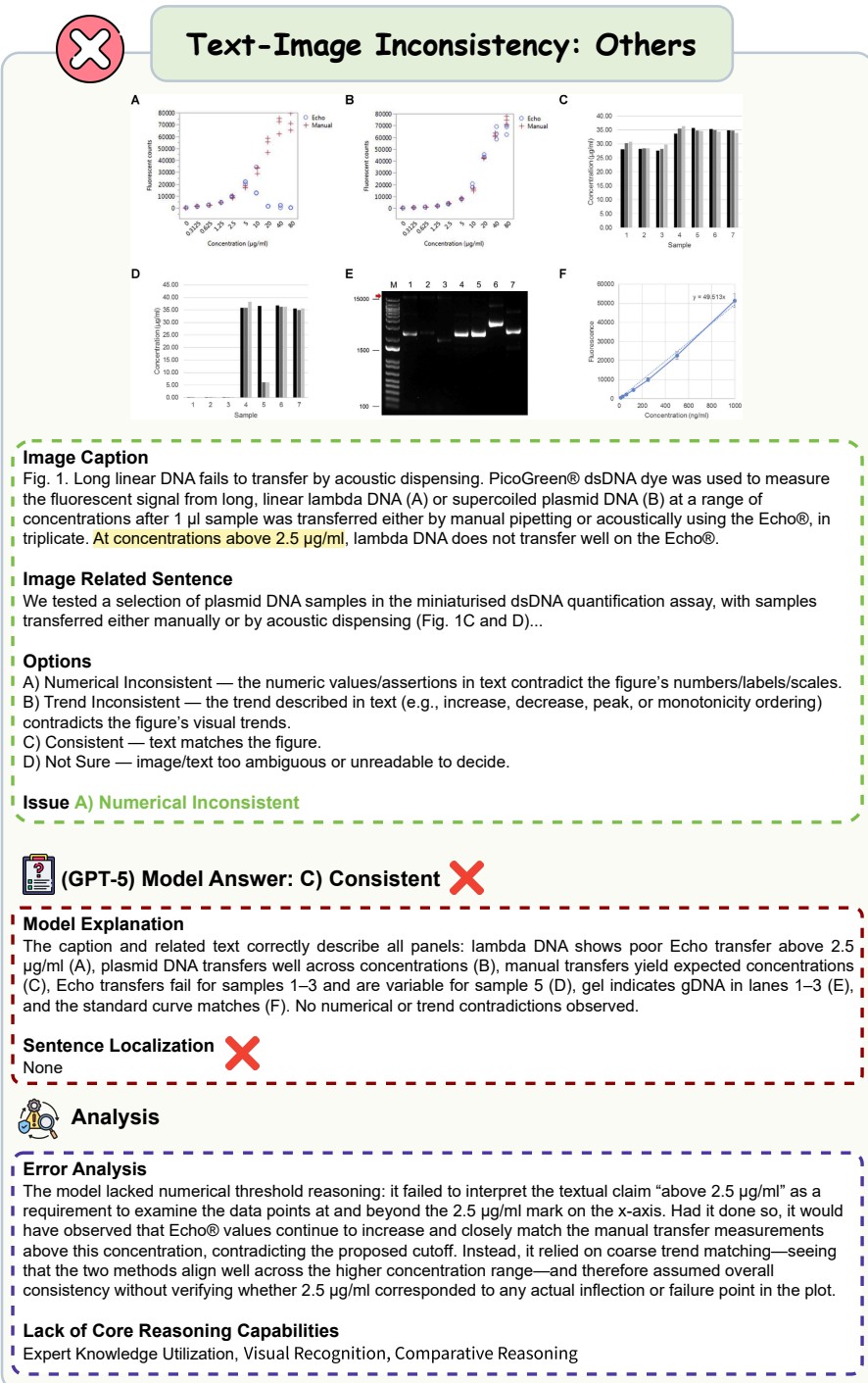

**Image Caption**
Fig. 1. Long linear DNA fails to transfer by acoustic dispensing. PicoGreen® dsDNA dye was used to measure the fluorescent signal from long, linear lambda DNA (A) or supercoiled plasmid DNA (B) at a range of concentrations after 1 μl sample was transferred either by manual pipetting or acoustically using the Echo®, in triplicate. At concentrations above 2.5 μg/ml, lambda DNA does not transfer well on the Echo®.

**Image Related Sentence**
We tested a selection of plasmid DNA samples in the miniaturised dsDNA quantification assay, with samples transferred either manually or by acoustic dispensing (Fig. 1C and D)...

**Options**
A) Numerical Inconsistent — the numeric values/assertions in text contradict the figure's numbers/labels/scales.
B) Trend Inconsistent — the trend described in text (e.g., increase, decrease, peak, or monotonicity ordering) contradicts the figure's visual trends.
C) Consistent — text matches the figure.
D) Not Sure — image/text too ambiguous or unreadable to decide.

**Issue A) Numerical Inconsistent**

**(GPT-5) Model Answer: C) Consistent** ❌

**Model Explanation**
The caption and related text correctly describe all panels: lambda DNA shows poor Echo transfer above 2.5 μg/ml (A), plasmid DNA transfers well across concentrations (B), manual transfers yield expected concentrations (C), Echo transfers fail for samples 1–3 and are variable for sample 5 (D), gel indicates gDNA in lanes 1–3 (E), and the standard curve matches (F). No numerical or trend contradictions observed.

**Sentence Localization** ❌
None

**Analysis**

**Error Analysis**
The model lacked numerical threshold reasoning: it failed to interpret the textual claim "above 2.5 μg/ml" as a requirement to examine the data points at and beyond the 2.5 μg/ml mark on the x-axis. Had it done so, it would have observed that Echo® values continue to increase and closely match the manual transfer measurements above this concentration, contradicting the proposed cutoff. Instead, it relied on coarse trend matching—seeing that the two methods align well across the higher concentration range—and therefore assumed overall consistency without verifying whether 2.5 μg/ml corresponded to any actual inflection or failure point in the plot.

**Lack of Core Reasoning Capabilities**
Expert Knowledge Utilization, Visual Recognition, Comparative Reasoning

Figure 27: **Failure case by GPT-5 on Text–Image Numerical Inconsistency.** The model misidentified and mislocalized.

## E.21   TEXT–IMAGE INCONSISTENCY: OTHERS (TREND – FAILURE CASE 1)

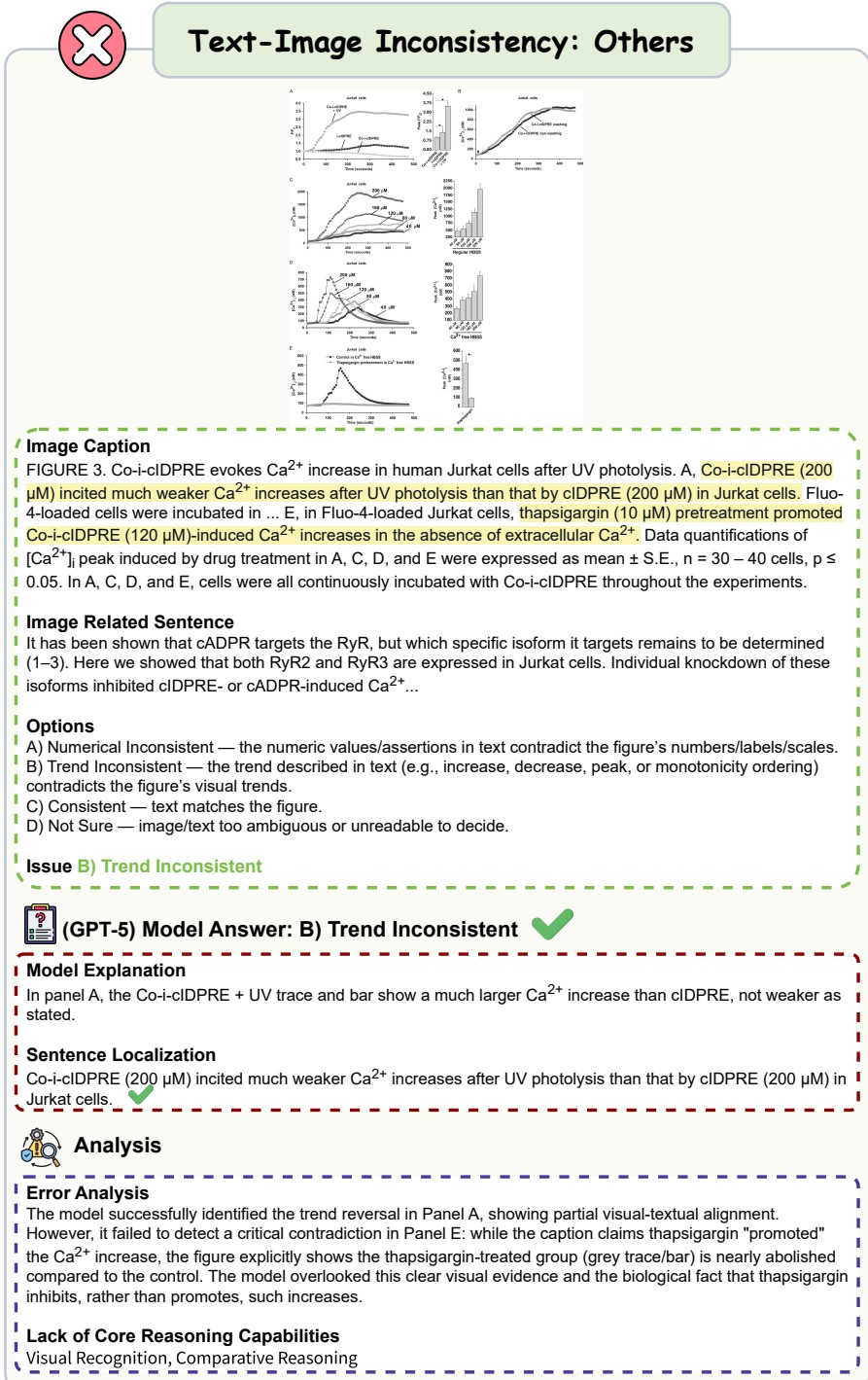

**Text-Image Inconsistency: Others**

**Image Caption**
FIGURE 3. Co-i-cIDPRE evokes $Ca^{2+}$ increase in human Jurkat cells after UV photolysis. A, Co-i-cIDPRE (200 µM) incited much weaker $Ca^{2+}$ increases after UV photolysis than that by cIDPRE (200 µM) in Jurkat cells. Fluo-4-loaded cells were incubated in ... E, in Fluo-4-loaded Jurkat cells, thapsigargin (10 µM) pretreatment promoted Co-i-cIDPRE (120 µM)-induced $Ca^{2+}$ increases in the absence of extracellular $Ca^{2+}$. Data quantifications of $[Ca^{2+}]_i$ peak induced by drug treatment in A, C, D, and E were expressed as mean ± S.E., n = 30 − 40 cells, p ≤ 0.05. In A, C, D, and E, cells were all continuously incubated with Co-i-cIDPRE throughout the experiments.

**Image Related Sentence**
It has been shown that cADPR targets the RyR, but which specific isoform it targets remains to be determined (1–3). Here we showed that both RyR2 and RyR3 are expressed in Jurkat cells. Individual knockdown of these isoforms inhibited cIDPRE- or cADPR-induced $Ca^{2+}$...

**Options**
A) Numerical Inconsistent — the numeric values/assertions in text contradict the figure's numbers/labels/scales.
B) Trend Inconsistent — the trend described in text (e.g., increase, decrease, peak, or monotonicity ordering) contradicts the figure's visual trends.
C) Consistent — text matches the figure.
D) Not Sure — image/text too ambiguous or unreadable to decide.

**Issue** B) Trend Inconsistent

**(GPT-5) Model Answer: B) Trend Inconsistent** ✅

**Model Explanation**
In panel A, the Co-i-cIDPRE + UV trace and bar show a much larger $Ca^{2+}$ increase than cIDPRE, not weaker as stated.

**Sentence Localization**
Co-i-cIDPRE (200 µM) incited much weaker $Ca^{2+}$ increases after UV photolysis than that by cIDPRE (200 µM) in Jurkat cells. ✅

**🔍 Analysis**

**Error Analysis**
The model successfully identified the trend reversal in Panel A, showing partial visual-textual alignment. However, it failed to detect a critical contradiction in Panel E: while the caption claims thapsigargin "promoted" the $Ca^{2+}$ increase, the figure explicitly shows the thapsigargin-treated group (grey trace/bar) is nearly abolished compared to the control. The model overlooked this clear visual evidence and the biological fact that thapsigargin inhibits, rather than promotes, such increases.

**Lack of Core Reasoning Capabilities**
Visual Recognition, Comparative Reasoning

Figure 28: **Failure case by GPT-5 on Text–Image Trend Inconsistency.** The model identified but mislocalized.

## E.22 Text–Image Inconsistency: Others (Trend – Failure Case 2)

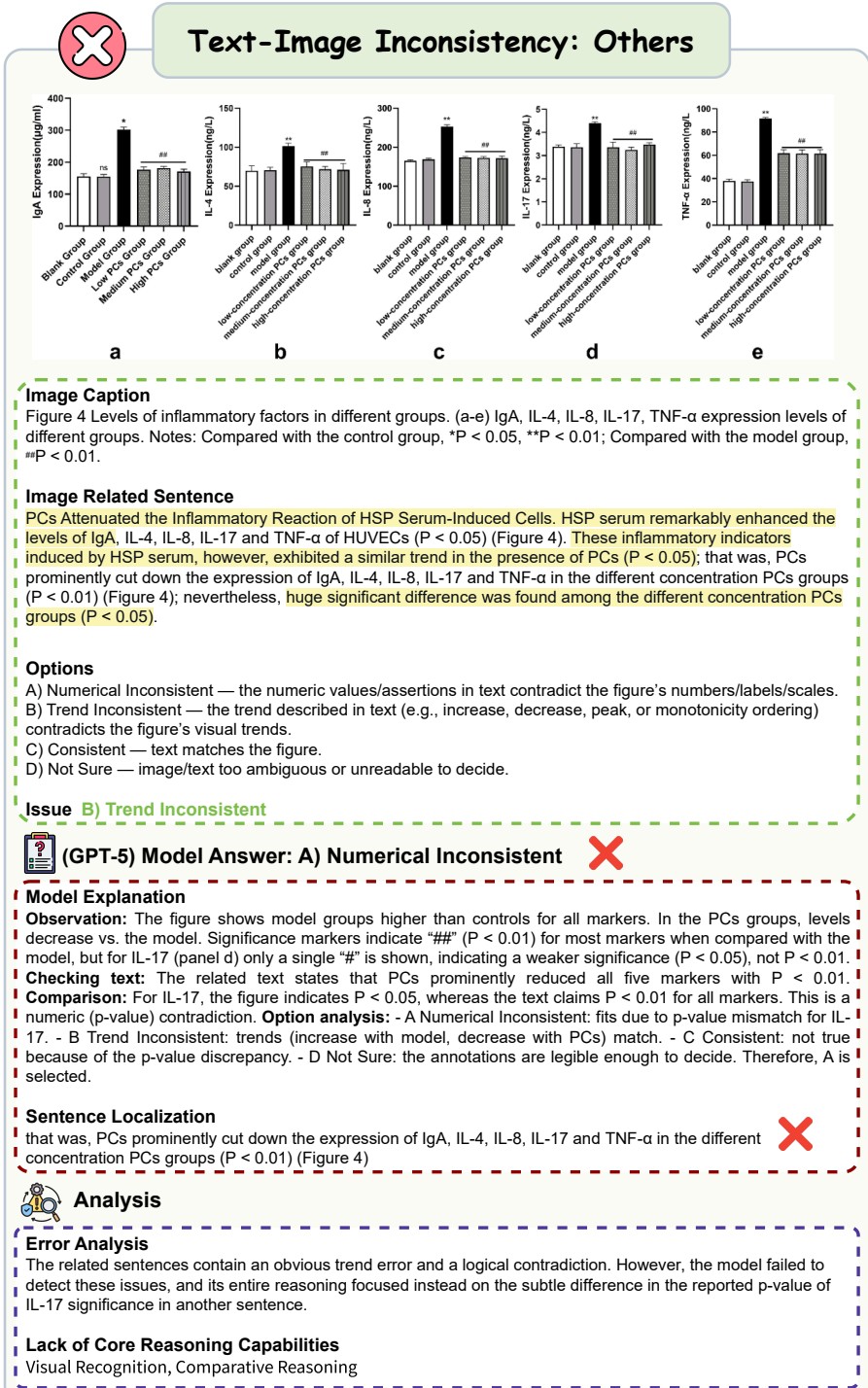

**Figure 29: Failure case by GPT-5 on Text–Image Trend Inconsistency.** The model misidentified and mislocalized.

# F DOCUMENT EXTRACTION AND PARSING DETAILS

## F.1 JUSTIFICATION OF PIPELINE SELECTION

Although recent document parsing tools (such as MinerU (Wang et al., 2024a)) offer more convenient interfaces for PDF structural analysis, these tools are not designed for constructing a scientific paper fraud forensics benchmark. In particular, they do not satisfy the following three key requirements of THEMIS: (1) **figure-level extraction**, (2) **accurate caption association**, and (3) **panel-level segmentation**.

In contrast, our Fitz + YOLOv7 (Wang et al., 2023) pipeline is widely used in our production environment. Our YOLOv7 model was trained on a 200k dataset of manually annotated academic panels and has passed commercial deployment-level reliability validation. In our evaluation, the Fitz + YOLOv7 pipeline achieved 94.10% precision, 93.94% recall, 94.02% F1 score, and 92.12% average IoU. The examples in Figure 30 show typical failure cases we observed when using MinerU, as well as the correct extraction results produced by our pipeline on the same samples.

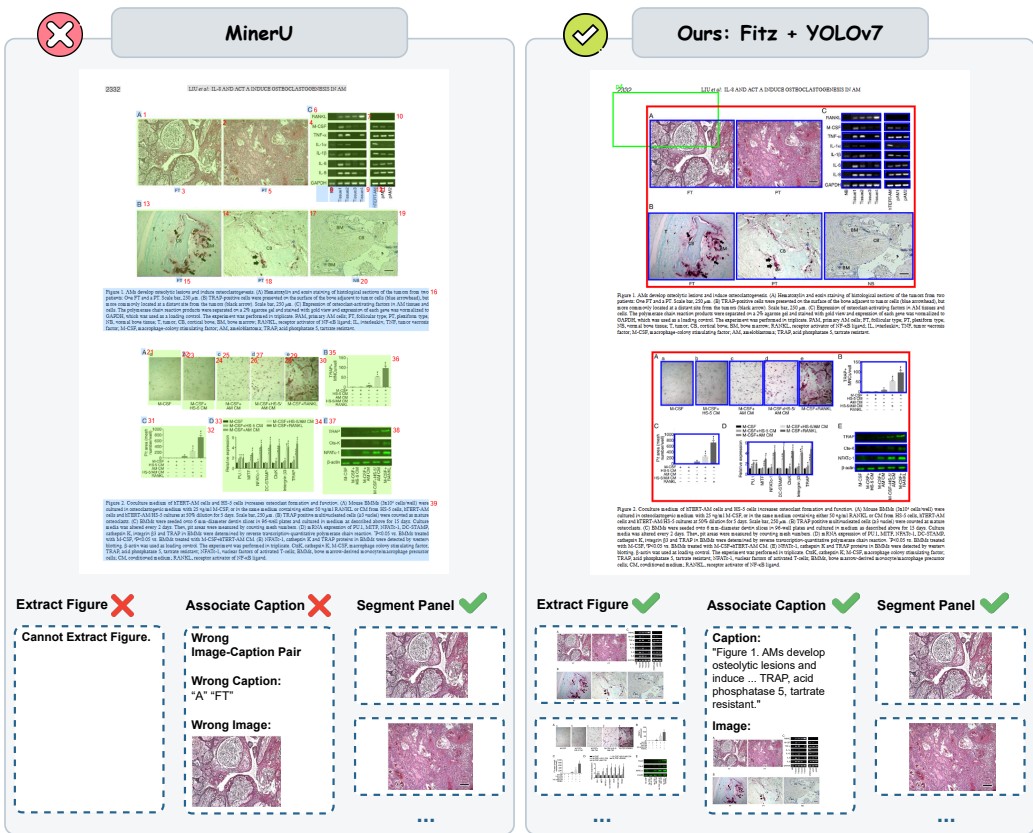

Figure 30: **Left: Extraction and Parsing by MinerU.** The green highlighted regions indicate MinerU's panel-level parsing outputs, while the blue regions represent its caption parsing results. MinerU **fails to** extract complete figures and frequently mismatches image–caption pairs. **Right: Extraction and Parsing by our pipeline.** The regions highlighted with red boxes correspond to **figures**, while those highlighted with blue boxes correspond to **panels**. Green boxes are used solely to verify the effectiveness of the visualization. Our extraction pipeline correctly identifies both full figures and their corresponding panels without structural or visual degradation.

Furthermore, across multiple document parsing methods evaluated in our production environment, we found that the Fitz + YOLOv7 pipeline consistently preserves the visual fidelity of all extracted figures and panels, without introducing noticeable degradation in resolution or clarity. In contrast, other document parsing tools (such MinerU) often perform panel slicing directly on rasterized PDF renderings, leading to a visible loss of clarity.

## F.2 ERROR ANALYSIS AND MANUAL CORRECTION

Our document extraction and parsing pipeline first calls Fitz to obtain complete figure regions from each PDF page, and then applies our YOLOv7-based detector to parse sub-panels within each figure.

Within the THEMIS construction pipeline, all automatically extracted figures and panels undergo rigorous human verification. Any panel-level extraction errors, such as boundary offsets, under-segmentation, or missed sub-panels, are manually corrected before inclusion in the final benchmark. This ensures that every figure and panel used in THEMIS is accurate and complete.

We further analyze representative parsing failure cases in Figures 31–33. These examples illustrate typical error modes, including overlapping panels, weakened boundary cues, and dense textual interference, and demonstrate how these cases arise.

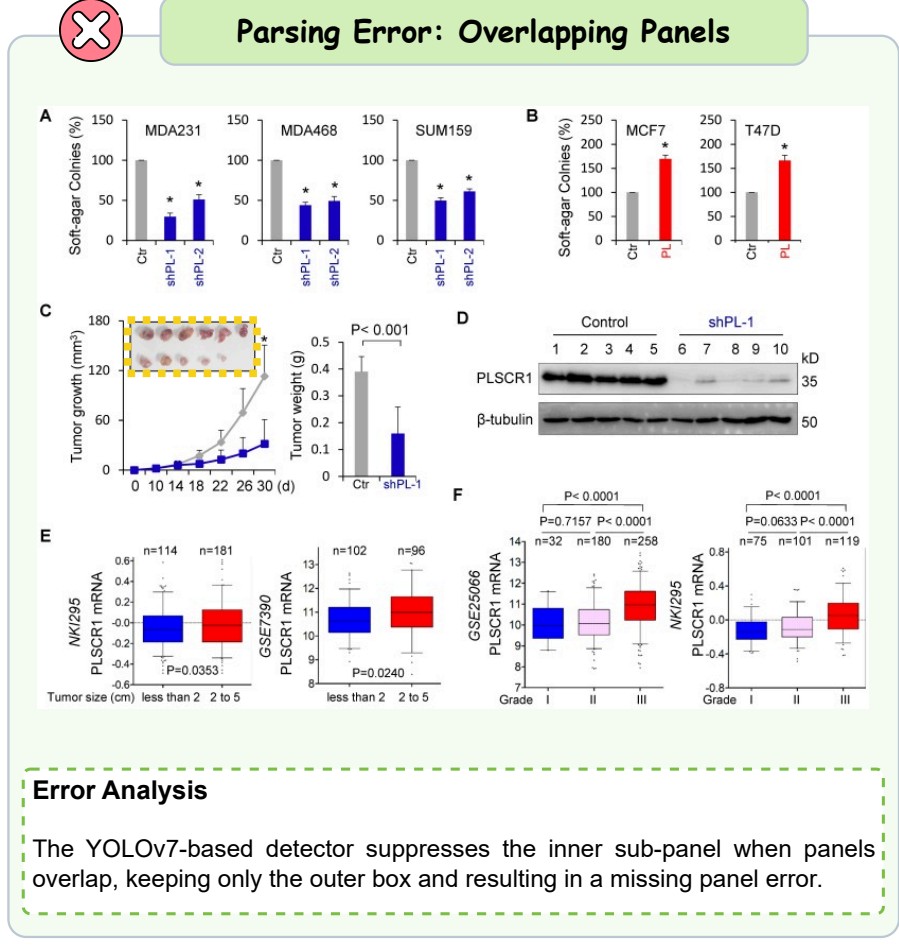

Figure 31: One missing panel highlighted with a yellow box.

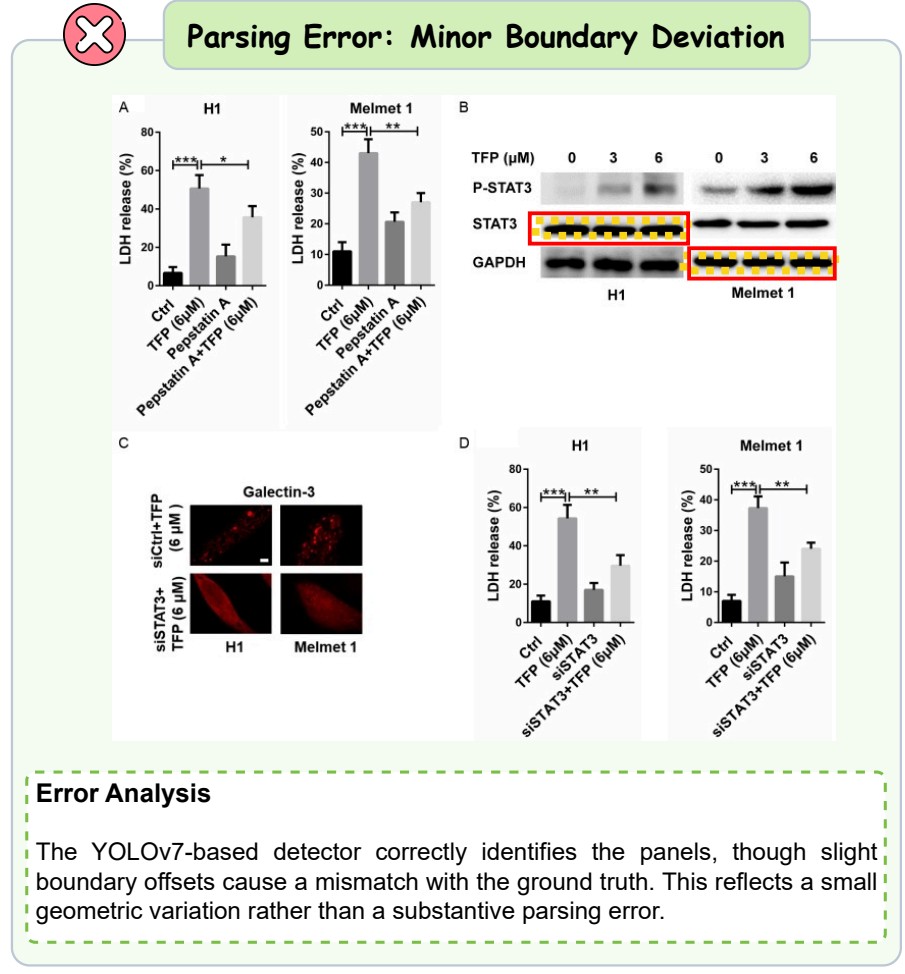

**Parsing Error: Minor Boundary Deviation**

**Error Analysis**

The YOLOv7-based detector correctly identifies the panels, though slight boundary offsets cause a mismatch with the ground truth. This reflects a small geometric variation rather than a substantive parsing error.

Figure 32: Two missing panels highlighted with yellow boxes. Two extra panels highlighted with red boxes.

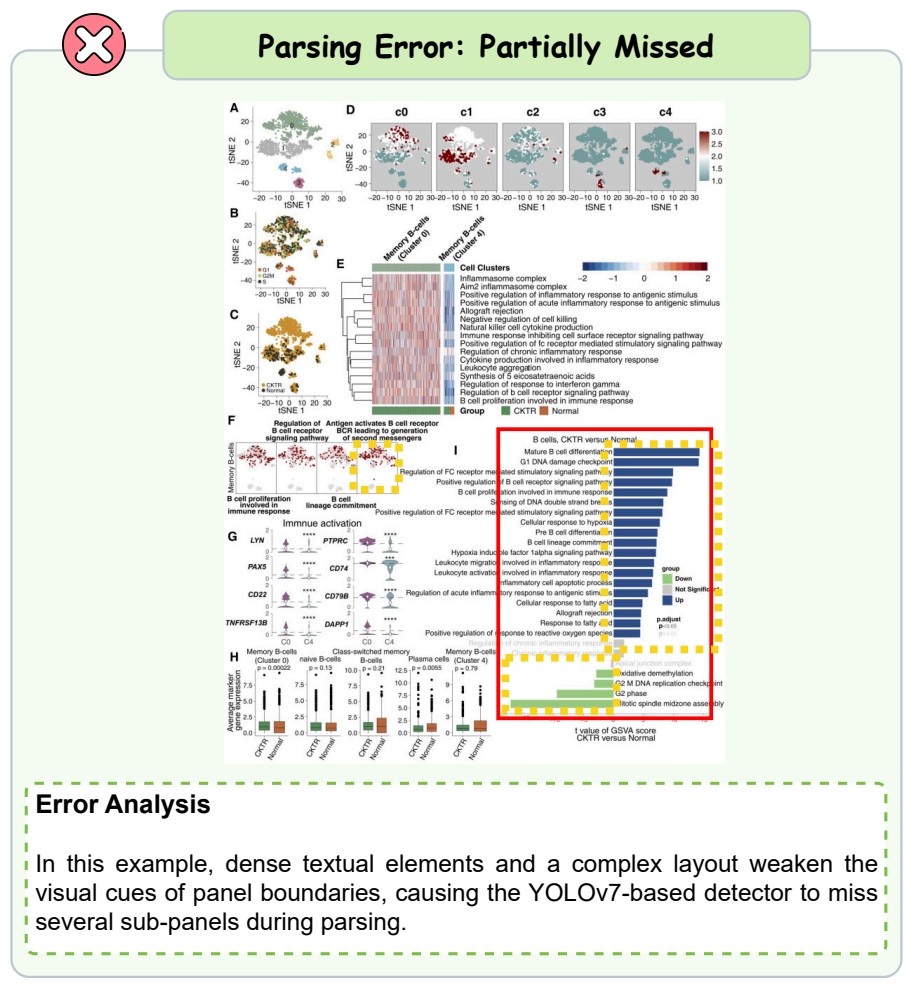

Figure 33: Three missing panels highlighted with yellow boxes. One extra panel highlighted with a red box.

# G    USE OF LLMS

In the preparation of this manuscript, LLMs were used to assist with language editing to improve readability and fluency. Additionally, generative models played a role in the construction of our benchmark, specifically for synthesizing controlled fraud data. Detailed methodologies regarding their usage in data generation are provided in Appendix A. We emphasize that all research concepts, experimental protocols, and analyses were human-led and validated. The authors reviewed all AI-assisted content and bear full responsibility for the scientific integrity of the work.

