# OpenReview forum: "THEMIS: Towards Holistic Evaluation of MLLMs for Scientific Paper Fraud Forensics"
_ICLR.cc/2026/Conference — ICLR 2026 Poster_

### Official Review · Reviewer_W3Ck · 2025-10-27

**Soundness:** 3
**Presentation:** 4
**Contribution:** 2
**Rating:** 8
**Confidence:** 3

**Summary:**

This paper presents THEMIS, a novel multi-task benchmark for evaluating Multimodal Large Language Models (MLLMs) in scientific paper fraud forensics. It features over 4K questions across 7 real-world academic scenarios (from retracted papers and synthetic data), covers 5 fraud tasks with 16 fine-grained manipulations, and maps tasks to 5 core reasoning capabilities. Experiments on 11 leading MLLMs show even top models (e.g., GPT-5 with 56.29% BRI) fall below passing thresholds, revealing limitations in handling compound manipulations and imbalanced capabilities.

**Strengths:**

* The starting point is novel and has practical application value.
* The data construction is complete, clear and reproducible.

**Weaknesses:**

* Typo: There is an issue with the citation in 4.4 Appendix
* From the perspective of the benchmark, it is quite well done. However, in the long run, this topic should be more suitable for optimizing, training, and fine-tuning models. If there are fine-tuning results, the value of this benchmark will be even higher.

**Questions:**

* In your opinion, how should the community use your benchmark? Is it to evaluate a model's ability to detect cheating and then select an excellent model to serve as a judge? Or are there more ambitious goals?

---

> ### Author Response · Authors · 2025-11-22
> **Rebuttal to Reviewer W3Ck's Comments (1)**
>
> > ### **Weaknesses 1: The citation of Case Study in 4.4 Appendix**
>
> We sincerely appreciate the reviewer's careful reading and valuable feedback. Regarding the typo you mentioned, we have already corrected it in the main text, and the link directing to the Case Study section is now functioning properly.
>
> **In the initial submission, due to the 50MB file-size limit, we had to place the full Case Study document inside the Supplementary Material as a ZIP file, which unfortunately led to the broken hyperlink in Section 4.4.** We assure you that in the camera-ready version, we will further optimize our compression strategy to avoid such issues.

---

> ### Author Response · Authors · 2025-11-22
> **Rebuttal to Reviewer W3Ck's Comments (2)**
>
> > ### **Weaknesses 2: More exploratory experiments on the higher value of this benchmark**
>
> Thank you for your insightful feedback, which has greatly helped us improve the quality of our work. We fully agree with your suggestion regarding more exploratory experiments to further demonstrate the higher value of this benchmark. **We are currently exploring fine-tuning–based experiments**, and we will update this section with new results once we obtain meaningful findings. Your suggestion is highly inspiring to us, and it aligns closely with the direction we plan to continuously pursue in our future work.

---

> ### Author Response · Authors · 2025-11-22
> **Rebuttal to Reviewer W3Ck's Comments (3)**
>
> > ### **Question 1: How the Community Can Leverage Our Benchmark**
>
> > ### Q: How the Community Can Benefit from This Benchmark?
>
> Thank you for this thoughtful question. THEMIS is designed to serve the community in two complementary ways. **First, it provides a rigorous testbed for evaluating and comparing multimodal models on purely visual, fine-grained fraud reasoning tasks.** This allows researchers to identify models with strong potential for academic image forensics. **Second, by examining the fine-grained error patterns revealed by our multi-dimensional evaluation framework, the community can gain a clearer understanding of current model limitations in fraud-related visual reasoning and derive more targeted research directions.**
>
> Our results show that even state-of-the-art models struggle with the challenges posed by THEMIS, particularly under compound manipulations involving multiple tampering operations. The benchmark also exposes highly imbalanced “capability profiles” across key visual reasoning dimensions such as Visual Recognition and Region Localization. These findings highlight THEMIS as both a broad and deeply challenging benchmark—one that not only assesses existing capabilities but also inspires future diagnostic evaluations and advances in model robustness.
>
> > ### Q: Are there more ambitious goals?
>
> Regarding the more ambitious goals, we envision THEMIS serving as a foundation for broader community progress. First, our construction pipeline can **support the open-source community** by generating richer and more comprehensive fraud data, including Splicing, Copy-Move, AI-Generated forgeries, Duplication, and Text–Image Consistency, which can be used to train end-to-end models specifically tailored for scientific-image fraud reasoning. Second, lightweight domain-specific models can be leveraged to produce additional supervision signals or synthetic annotations, enabling **further optimization of larger multimodal models**. Ultimately, our long-term goal is to **enhance the visual fraud reasoning capabilities and generalization of multimodal large models** in real scientific-review scenarios.

---

### Official Review · Reviewer_f9Wq · 2025-10-28

**Soundness:** 3
**Presentation:** 4
**Contribution:** 4
**Rating:** 8
**Confidence:** 4

**Summary:**

This paper proposes a new benchmark, termed THEMIS, to comprehensively evaluate the capability of current MLLMs in scientific paper fraud forensics. Compared with existing visual fraud reasoning benchmarks, the proposed THEMIS exhibits several advantages, including real-world scenarios & complexity, tasks diversity & granularity, and multi-dimensional capability evaluation. Based on THEMIS, the authors extensively evaluate recent open-source and proprietary MLLMs and conduct in-depth analysis.

**Strengths:**

1. THEMIS defines a comprehensive taxonomy for the field of scientific paper fraud forensics.  Specifically, THEMIS covers 7 academic scenarios, 5 tasks, 16 manipulation operations, and 5 core reasoning capabilities, which is more diverse than existing visual fraud reasoning benchmarks.
2. The data quality of THEMIS is high. The synthetic data is rigorously reviewed by human experts. Moreover, THEMIS contains real samples in addition to synthetic samples, which makes it closer to real-world applications.
3. The authors provide the details of data curation and question generation, which lays a soild groundwork for future research.
4. The authors conduct comprehensive evaluation and in-depth analysis based on THEMIS. The MLLMs involved includes 6 proprietary ones and 5 open-source ones. The evaluation results demonstrate the significant challenges posed by this benchmark. Furthermore, the analysis conclusions provides deep insights into the related field.

**Weaknesses:**

1. The MLLMs evaluated in this paper are not comprehensive enough. It is recommended to supplement the results of InternVL3.5, GLM4.5V, Gemini-2.5-Pro, Claude, etc.
2. There is a lack of comparsion on difference parameter sizes of the same series of MLLMs (e.g., Qwen2.5-VL-3B/7B/32B/72B).
3. The conclusion in lines 373-375 is not well explained.

**Questions:**

In lines 1443-1444, "However, performance drops markedly on synthetic data, with some tasks almost completely failing (e.g., LLaMA approaches zero on both splicing and copy-move)". This statement may be inconsistent with Table 10.

---

> ### Author Response · Authors · 2025-11-22
> **Rebuttal to Reviewer f9Wq's Comments (1)**
>
> > ### **Weaknesses 1: Recommended to supplement the results of InternVL3.5, GLM4.5V, Gemini-2.5-Pro, Claude, etc.**
>
> Following reviewer's suggestion, **we have updated 4 additional models on THEMIS experiments** : InternVL3.5, GLM4.5V, Gemini 2.5 Pro, Claude 4.5 Sonnet. Please refer to Table 2 for the updated version.
>
> **The four newly added models exhibit findings consistent with our overall results.**
>
> The results are reported in the following table. We report the Identification (Id) Score and Localization (Loc) Score of five types of fraud tasks including: Splicing (SPL), Copy-Move (CM), AI-generated(AIG), DUP (Duplication), Text-Image Inconsistency (TII). Together with Balanced Robustness Index(BRI). More details on the evaluation metrics are provided in Appendix B.
>
> | Model            | Single-Mode Forgery Identification (Id) |      |      |      |      | Single-Mode Forgery Localization (Loc) |      |      |   |   Composite Manipulation Operations (Id)    | Cross-Modal Inconsistency (Id & Loc)  |  | BRI |
> |------------------|--------------------------------------------|------|------|------|------|--------------------------------------------|------|------|------|------|----------------------------------------|--------------------------------------|-----|
> |                  | SPL        | CM         | AIG         | Avg   |      | SPL        | CM         | AIG         | Avg   | DUP  | TII Id Score  | TII Loc Score |     |
> | **InternVL3.5**  | 30.97      | 58.42      | 67.00       | 52.13 |      | 43.37      | 40.24      | 33.19       | 38.93 | 10.34 | 55.00         | 4.77          | 26.35 |
> | **GLM4.5V**      | 29.23      | 58.43      | 54.51       | 47.39 |      | 8.50       | 22.22      | 10.63       | 13.78 | 15.72 | 53.66         | 22.69         | 28.25 |
> | **Gemini 2.5 Pro** | 30.60    | 47.46      | 14.90       | 30.99 |      | 46.65      | 46.60      | 47.20       | 46.82 | 15.63 | 44.66         | 37.30         | 28.80 |
> | **Claude 4.5 Sonnet** | 29.71 | 75.66      | 30.43       | 45.27 |      | 14.94      | 44.11      | 20.98       | 26.68 | 15.29 | 35.66         | 25.77         | 26.66 |

---

> ### Author Response · Authors · 2025-11-22
> **Rebuttal to Reviewer f9Wq's Comments (2)**
>
> > ### **Weaknesses 2: Comparison on difference parameter sizes of the same series of MLLMs (e.g., Qwen2.5-VL-3B/7B/32B/72B)**
>
> Following reviewer' suggestion, we have updated comparisons across different parameter scales within the same MLLM family (e.g., Qwen2.5-VL-32B/72B). Please refer to Table 2 for the updated version.
>
> | Model            | Single-Mode Forgery Identification (Id) |      |      |      |   Single-Mode Forgery Localization (Loc)   |    |      |  | Composite Manipulation Operations (Id)  | Cross-Modal Inconsistency (Id & Loc) |  | BRI | |
> |------------------|-------------------------------------------|------|------|------|------|-------------------------------------------|------|------|------|------|---------------------------------|--------------------------------------|-----|
> |                  | SPL     | CM      | AIG     | Avg   | SPL     | CM      | AIG     | Avg   | DUP   | TII Id Score | TII Loc Score |     |
> | **Qwen2.5 VL 32B** | 30.31   | 57.48   | 39.24   | 31.23 | 5.90    | 18.93   | 7.44    | 10.76  | 14.28 | 58.33        | 17.94         | 9.14 |
> | **Qwen2.5 VL 72B** | 36.40   | 77.27   | 51.06   | 54.91 | 51.66   | 55.16   | 35.57   | 47.46  | 16.75 | 61.33        | 12.32         | 47.16 |
>
> A notable finding is that **Qwen2.5-VL-32B consistently underperforms Qwen2.5-VL-72B across all tasks**, with particularly **large degradation in Single-Mode Forgery Localization**.
>
> For the small-parameter models Qwen2.5-VL-3B and Qwen2.5-VL-7B, we observe in small-sample evaluations that observe that they **fail to perform reliably on visual fraud reasoning tasks**. Their outputs often **ignore the required response format** and **lack coherent reasoning chains**, suggesting limited instruction-following capability and insufficient logical inference ability for this domain.
>
> The **response case from Qwen2.5-VL-3B** exhibits multiple critical issues, including **multiple option selections**, **abnormal or incomplete explanations**, **non-compliant output formatting**, and other inconsistent behaviors.
>
> - \<CHOICE\>A/B/C/D/\>E\>E\>E\>E\>E\>E\>E\>E\>E\>E\>...
> - \<CHOICE\>A\</CHOICE\>\n\<MASK\>1,2,3,4,5,6\</MASK\>\n\<EXPLANATION\>The image shows a clear splicing effect, as indicated indicated...
> - \<CHOICE\>A/B\>C\>CHOICE, Block 1, 1
>
> The **response case from Qwen2.5-VL-7B** presents similar critical issues to Qwen2.5-VL-3B.
> - \<CHOICE\>A\</CHOICE\>\n\<MASK\>1,2,3,4\</MASK\>\n\<EXPLANATION\>The image shows a single panel with no apparent signs of splicing, copy-move, or AI generation. The consistent texture and lighting across
> - \<CHOICE\>A\</CHOICE\>\n\<MASK\>1,2\</MASK\>\n\<EXPLANATION\>The image shows a single panel with no apparent signs of splicing, not the entire
> - \<CHOICE\>A\</CHOICE\>\n\<MASK\>4\</MASK\>\n\<EXPLANATION\>The image shows a clear splicing effect, as indicated by the unnatural seam between the two distinct sections of the image. This is evident in Block 4, which has a different color gradient compared to determine if there's any evidence of splicing.\n\<CHOICE\>C\</CHOICE\>\n\<MASK\>

---

> ### Author Response · Authors · 2025-11-22
> **Rebuttal to Reviewer f9Wq's Comments (3)**
>
> > ### **Weaknesses 3 & Question :  Findings from Real-Data Evaluation and Explanations**
>
> We sincerely appreciate the reviewer’s valuable suggestions. Following your comments, we **have strengthened the analysis of experimental results in the main text** and **revised the wording of Appendix C.1** to ensure greater accuracy.
>
> The updates are as follows:
>
> Synthetic data **surpass real data in difficulty for the Duplication task** because they incorporate multiple compounded manipulations. These include **geometric transformations** (horizontal flipping, vertical flipping, and 180° rotation) and **parameter-modified operations** (brightness modification, contrast adjustment, saturation control, hue shifting, and color temperature variation), which are generally more complex than the direct or scaling duplication commonly observed in real cases.
>
> **For the Forgery task, our synthetic data exert comparable detection pressure to real data**, as they fully reflect diverse real-world fraud behaviors and cover the known fraudulent operations reported in retracted publications.
>
> The primary exception lies in the Splicing task, where real samples remain more challenging due to their inherently more sophisticated merging patterns.
>
> As shown in Appendix Table C.1, model performance on synthetic and real data is closely aligned for Copy-Move task. For instance, GPT-5 achieves a 36.41% localization score on synthetic data, compared to 33.39% on real data. In Duplication task, synthetic data impose even greater difficulty; for example, Gemini 2.5 Flash reaches a 49.58% identification score on real cases, yet drops to only 24.96% on synthetic data.
>
> The convergence of performance between synthetic and real data demonstrates that our fraud data generation pipeline can continuously produce high-quality and scalable fraud samples despite the scarcity and prohibitively high annotation cost of real-world cases. This enables robust evaluation of five key capabilities **(Expert Knowledge Utilization, Visual Recognition, Spatial Reasoning, Region Localization, and Comparative Reasoning)** within an **extensible testbed** for multimodal scientific forgery assessment.
>
> Moreover, model performance on synthetic data serves as a reliable indicator of real-world capability. For example, GPT-5, the strongest model on synthetic detection tasks, also exhibits relatively stable performance on real samples. Thus, synthetic data not only provide comprehensive testing of reasoning capabilities but also function as a predictive signal for real scientific review scenarios.
>
> We further observe that real splicing cases are more challenging than synthetic ones because real-world fraud operations often employ more complex and subtle splicing strategies, such as localized contrast/brightness adjustment or texture smoothing at the merged boundaries. An improved splicing synthesis method will be incorporated in future work to better mimic these nuanced operations and more effectively expose current models'  limitations in boundary perception.

---

> > ### Comment · Reviewer_f9Wq · 2025-11-25
> >
> > Thank you for your efforts and detailed responses. I would like to maintain the original score.

---

> > > ### Author Response · Authors · 2025-11-26
> > >
> > > Thanks for your reply! If there are any remaining questions or points requiring further clarification, we would be pleased to engage in further discussion.

---

### Official Review · Reviewer_bNnH · 2025-10-28

**Soundness:** 3
**Presentation:** 3
**Contribution:** 4
**Rating:** 6
**Confidence:** 3

**Summary:**

This paper constructs a benchmark for forgery forensics in academic papers, covering various forms of academic misconduct across multiple research domains. The benchmark includes both simple manipulations such as copy-move and fully AI-generated forgeries, and provides a comprehensive evaluation of existing MLLMs. The results show that current MLLMs perform poorly on this benchmark, with localization tasks performing significantly worse than attribution tasks, highlighting the substantial room for improvement in this area.
**As I am not deeply familiar with this specific domain, I would prefer to refer to the opinions of other reviewers when determining the final score.**

**Strengths:**

1. The constructed benchmark covers a wide range of academic disciplines and forgery types, and the evaluation includes a relatively comprehensive set of model categories.
2. Although there are a few typos, the paper is overall well-written, and the figures and visual presentations are clear and well-designed.

**Weaknesses:**

1. The authors mention using the **Fitz** library and **YOLOv7** for information extraction and segmentation. However, to my knowledge, there are now more accurate tools in the document extraction domain, such as **dots.ocr** and **MinerU**. Given the diversity of samples, the effectiveness of simply applying Fitz (PyMuPDF) and YOLOv7 is questionable. Since subsequent steps rely heavily on accurate information extraction, this stage could be improved to ensure higher benchmark quality and reliability.
2. The appendix should include more **case examples** from different categories within the benchmark to give readers an intuitive understanding of its scope and difficulty level.

**Questions:**

Please refer to the *Weakness* section for detailed explanations.
If the authors can adequately address my concerns, I would be very willing to raise my score.

---

> ### Author Response · Authors · 2025-11-21
> **Rebuttal to Reviewer bNnH's Comments (1)**
>
> > ### **Weaknesses 1: Use of Document Parsing Tools**
>
> **【Why the Fitz library and YOLOv7 are adopted in our pipeline】**
>
> Our task requires constructing datasets that include **Figure, Caption, Related Sentence, and Panel** annotations.
>
> The Fitz library and our YOLOv7 have been adopted in our production environments. In particular, the YOLOv7 model was fully trained on 200,000 manually annotated academic panels, and **has been validated to meet production-level reliability and commercial deployment standards**.
>
> To ensure the completeness and accuracy required by our benchmark, we adopt the following data construction pipeline:
>
> Step 1. We first extract full **figures** using the Fitz library. Here, “figures” refer to full-figure units paired one-to-one with their captions.
>
> Step 2. We construct accurate **<figure, caption, related-sentence> triples** through manual verification, assisted by GPT-4o mini. Related sentences are particularly important for building Text–Image Inconsistency samples.
>
> Step 3. We then segment each figure into **panels** using our trained YOLOv7 model. Here, “panels” refer to the smallest indivisible visual units. This step is essential for generating AI-Generated, Copy-Move, Splicing, and Duplication samples.
>
>
>  **【Why our dataset construction cannot rely on dots.ocr or MinerU】**
>
> However, existing tools such as **dots.ocr and MinerU** cannot fulfill the requirements of our tasks.
>
> • **dots.ocr** cannot extract related sentences and does not support panel segmentation.
>
> • **MinerU** cannot extract figures, fails to correctly extract captions, and cannot obtain related sentences.
>
> As shown in the table below.
>
> | Construction Step \ Tool Used | Extract Figures | Parse Figure Captions | Parse Related Sentences | Split Panels |
> |-------------------------------|-----------------|------------------------|--------------------------|--------------|
> | MinerU                        | ❌              | ❌ (over 90% captions incorrectly assigned to panels) | ❌        | ✅           |
> | dots.ocr                      | ✅              | ✅                      | ❌                        | ❌           |
> | Ours                          | ✅ Fitz         | ✅ Manual (GPT-4o mini assisted) | ✅ Manual (GPT-4o mini assisted) | ✅ YOLOv7  |
>
> Furthermore, the limitations of these two tools within our dataset construction pipeline are summarized as follows:
>
> Step 1. PDF → Figure.
>
> This step is a necessary prerequisite for both Step 2 and Step 3. However, **MinerU is unable to extract full figures**.
>
> Step 2. PDF → <Figure, Caption, Related Sentence>.
>
> This step is essential for constructing Text–Image Inconsistency data, as inconsistencies may arise in either the caption or the related sentences. However, **dots.ocr cannot obtain related sentences**, and **MinerU not only fails to extract related sentences but also produces incorrect caption associations**—often mismatching a caption to a random panel rather than to the corresponding figure.
>
> Step 3. Figure → Panel.
>
> This step is required for constructing AI-Generated, Copy-Move, Splicing, and Duplication samples, as our investigation of real-world retraction cases shows that scientific misconduct is often localized at the panel level. However, **dots.ocr does not support panel segmentation**.
>
> Therefore, compared with these tools, our data construction pipeline is both necessary and scientifically grounded, ensuring that all required components are accurately produced for the benchmark.

---

> ### Author Response · Authors · 2025-11-21
> **Rebuttal to Reviewer bNnH's Comments (2)**
>
> > ### **Weaknesses 2:  Case Expansion and Updating**
>
> Thank you for your suggestion. We sincerely appreciate your request to include additional case studies, as it helps us present the findings more transparently and in a more practical context. Based on your feedback, we have expanded and organized detailed case examples to enhance clarity and better support our contribution.
>
> We have updated Appendix E in the Supplementary Material with **22 representative cases** , which comprehensively cover the full scope of THEMIS.
>
> These cases span all **7 academic scenarios**  (Chart, Diagram, Physical Object, Medical Imaging, Micrograph, Stained Micrograph, Others) .
>
> They are systematically organized across **5 tasks** : Splicing, Copy-Move, AI-Generated forgeries (Image Inference Forgery / Targeted Region Editing), Duplication (Global / Local), and Text–Image Inconsistency.
>
> The summary of all 22 cases is provided below.
>
> | # | Task | Academic Scenario | Lack of Core Reasoning Abilities | Correct/Error | Error Pattern |
> |---|------|------------------|---------------------------------|----------------|---------------|
> | 1 | AI-Generated (Image Inference Forgery) | Medical Imaging | – | **correct** | – |
> | 2 | AI-Generated (Image Inference Forgery) | Micrograph | Expert Knowledge Utilization | **error** | misidentified |
> | 3 | AI-Generated (Targeted Region Editing) | Diagram | – | **correct** | – |
> | 4 | AI-Generated (Targeted Region Editing) | Chart | Region Localization, Spatial Reasoning | **error** | identified & mislocalized |
> | 5 | AI-Generated (Targeted Region Editing) | Physical Object | Expert Knowledge Utilization | **error** | misidentified & localized |
> | 6 | AI-Generated (Targeted Region Editing) | Chart | Region Localization, Expert Knowledge Utilization, Spatial Reasoning | **error** | misidentified & mislocalized |
> | 7 | Splicing | Chart | – | **correct** | – |
> | 8 | Splicing | Medical Imaging | Region Localization, Visual Recognition | **error** | identified & mislocalized |
> | 9 | Splicing | Chart | Expert Knowledge Utilization, Visual Recognition, Spatial Reasoning, Comparative Reasoning | **error** | misidentified & mislocalized |
> |10 | Copy-Move | Stained Micrograph | – | **correct** | – |
> |11 | Copy-Move | Stained Micrograph | Region Localization, Visual Recognition | **error** | identified & mislocalized |
> |12 | Copy-Move | Stained Micrograph | Expert Knowledge Utilization, Visual Recognition | **error** | misidentified & localized |
> |13 | Copy-Move | Stained Micrograph | Expert Knowledge Utilization, Visual Recognition, Region Localization | **error** | misidentified & mislocalized |
> |14 | Duplication (Global, Flipped) | Micrograph | – | **correct** | – |
> |15 | Duplication (Rotated & Parameter-modified) | Stained Micrograph | Spatial Reasoning, Comparative Reasoning | **error** | misidentified |
> |16 | Duplication (Rotated) | Stained Micrograph | Expert Knowledge Utilization, Visual Recognition, Spatial Reasoning, Comparative Reasoning | **error** | misidentified |
> |17 | Duplication (Parameter-modified) | Stained Micrograph | Visual Recognition, Spatial Reasoning, Comparative Reasoning | **error** | misidentified |
> |18 | Text-Image Inconsistency (Numerical) | Others | – | **correct** | – |
> |19 | Text-Image Inconsistency (Numerical) | Others | Visual Recognition, Comparative Reasoning, Expert Knowledge Utilization | **error** | identified & mislocalized |
> |20 | Text-Image Inconsistency (Consistent) | Others | Visual Recognition, Comparative Reasoning, Expert Knowledge Utilization | **error** | misidentified & mislocalized |
> |21 | Text-Image Inconsistency (Trend) | Others | Visual Recognition, Comparative Reasoning | **error** | identified & mislocalized |
> |22 | Text-Image Inconsistency (Trend) | Others | Visual Recognition, Comparative Reasoning | **error** | misidentified & mislocalized |
>
> We hope the revisions adequately address your concerns and appreciate your consideration.

---

> ### Author Response · Authors · 2025-11-26
> **Follow-Up on Our Rebuttal Responses**
>
> Dear Reviewer bNnH,
>
> Thank you again for your careful review and insightful comments on our paper. We have now provided detailed responses and additional analyses in the rebuttal to address your concerns. If you have a moment to look over our replies and let us know whether there are any remaining points that would benefit from further clarification, we would be very grateful.
>
> Wishing you a joyful Thanksgiving! :-)
>
> Best regards,
> Authors

---

> ### Comment · Reviewer_bNnH · 2025-11-27
>
> First of all, I apologize for only having time now to provide a detailed response to the authors’ rebuttal. The rebuttal has addressed most of my concerns, but I believe there are still two issues that are particularly crucial for the benchmark.
>
> 1. **Regarding the construction of the benchmark**
>    The overall quality of the benchmark is tightly coupled with the reliability of the document extraction process. The rebuttal suggests that the extraction pipeline is sufficient, but this appears incomplete. In practice, more recent document parsing tools—such as *MinerU*—are capable of extracting figures and other structural elements, whereas the *fitz*-based extraction used in the paper is comparatively dated and relies heavily on heuristic rules. I therefore encourage the authors to provide concrete examples of potential failure cases in their extraction process and to explain how these cases are detected and handled.
>
> 2. **Regarding the presentation of benchmark examples**
>    As a benchmark, the number of examples shown in Figure 4 of the main paper is insufficient. I strongly recommend including a more comprehensive set of question–answer examples in the appendix. Specifically, the appendix should contain additional examples for all five types of fraud detection tasks, including the question text, the outputs produced by different models, and annotations indicating whether each response is correct. I noticed that the authors mentioned in the rebuttal that Appendix E would include 22 additional examples, but I could not find them anywhere in the PDF. I understand that the authors may not have had enough time to provide a large number of examples, but at the very least, there should be one illustrative example for each task type.

---

> > ### Author Response · Authors · 2025-11-27
> > **Clarification on Appendix E: Complete Benchmark Examples in Supplementary Zip File**
> >
> > Dear Reviewer bNnH,
> >
> > Thank you for your thoughtful feedback. We sincerely apologize for any confusion regarding the presentation of benchmark examples in Appendix E.
> >
> > To clarify, due to the size limitations of the PDF (<50MB), **we placed the full set of 22 examples (covering all five fraud detection task types) in the ''Supplementary Material'' Zip file on OpenReview**, as noted in Appendix E’s header. Each example includes: The question text, Outputs from evaluated models, Correctness annotations, and Detailed explanations.
> >
> > The PDF appendix currently contains a placeholder (Section E) directing reviewers to the Zip file for these details. We regret that this might have been overlooked and are happy to highlight the Zip file’s location (i.e., under "Supplementary Material" on OpenReview).
> >
> > Regarding your first point about extraction failure cases, we are preparing concrete examples and mitigation strategies and will share them soon. Thank you for your patience!
> >
> > Best regards,
> >
> > Authors

---

> > > ### Comment · Reviewer_bNnH · 2025-11-28
> > > **Thx for Detailed Statement**
> > >
> > > I appreciate the reviewers for pointing out the exact location of the supplementary examples. These examples clearly demonstrate the input–output cases of the evaluation tasks, which greatly help in intuitively understanding how current strong models (e.g., GPT-5) perform on this benchmark and what capabilities they still lack. The high-quality images and questions shown in the examples also alleviate my earlier concerns about the dataset quality. Therefore, I believe the authors have addressed essentially all of my concerns. I would like to raise my score to an 8 to reflect my recognition of this work. However, it seems that I currently do not have access to the "edit" button to modify my score directly, but I believe it is necessary to state my position explicitly in this review.

---

> > ### Author Response · Authors · 2025-11-28
> > **Clarification on Appendix F: Concrete Examples of Extraction Failure Cases and How They Are Addressed**
> >
> > Dear Reviewer bNnH,
> >
> > In line with your earlier suggestion, we have prepared a concise supplementary note **in Appendix F** that explains our choice of document parsing tools and presents several representative failure-case analyses. **This material has been included in the “Supplementary Material’’ zip file on OpenReview**. In our evaluation, the Fitz + YOLOv7 pipeline **achieves a precision of 0.9410, a recall of 0.9394, an F1 score of 0.9402, and an average IoU of 0.9212**. We additionally provide analyses of representative extraction failure cases. These examples illustrate typical error modes, including overlapping panels, weakened boundary cues, and dense textual interference. All such parsing errors are manually corrected before inclusion in the final benchmark.
> >
> > Thank you again for your insightful and constructive feedback! We’re excited to hear that our revisions have effectively addressed your concerns!

---

### Author Response · Authors · 2025-12-02
**Summary of Rebuttal**

Dear Program Chairs, Senior Area Chairs, Area Chairs, and Reviewers,

We sincerely appreciate the reviewers' constructive suggestions and the Area Chairs' guidance during the rebuttal phase. **We are encouraged that reviewers recognized our core contributions, including the construction of the new dataset, comprehensive experimental evaluation, and thorough analytical results.**

During discussions, we clarified:

1. The pipeline’s adequacy for figure and panel extraction, providing clear support that our extraction workflow fully satisfies the functional requirements of dataset construction.

2. Expanded Case Study section, improving completeness and navigation. Minor corrections addressing the broken hyperlink noted by reviewers.

3. Additional model and parameter-scale comparative analysis, reaffirming the original findings and model-ranking stability.

4. Refined real-data analysis showing that synthetic data largely mirror real-world difficulty and reveal key model limitations.

We are pleased that:

- **All Reviewers expressed strong endorsement of our work, resulting in an average rating of 8 (8, 8, 8).**

  - By providing additional model comparisons and clearer explanations of the experimental findings, we **fully addressed Reviewer f9Wq’s concerns**, leading the reviewer to maintain the original favorable score.

  - By addressing concerns on extraction tools and case examples, we **reassured Reviewer bNnH**, leading the reviewer to **raise the rating  from 6 to 8**.

Key Strengths Acknowledged by Reviewers:

- **Comprehensive Benchmark Design** (f9Wq)

- **High-Quality and Realistic Data** (f9Wq & W3Ck)

- **Clear Presentation and Strong Methodology** (bNnH & f9Wq)

- **Extensive and Insightful Evaluation** (bNnH & f9Wq)

- **Novelty and Practical Value** (W3Ck)

Addressed Concerns with Detailed Analyses:

- Clarifications on Document Parsing Tools (bNnH)

- Case Study Expansion and Corrections (bNnH & W3Ck)

- Comparative Analysis with Additional MLLMs and Qwen Scaling (f9Wq)

- Findings from Real-Data Evaluation (f9Wq)

- Exploratory Experiments on Benchmark Value (W3Ck)

- Community Use and Future Directions of the Benchmark (W3Ck)

Demonstrated Contributions:

- **High-Quality and Reproducible Data Construction Pipeline**: A rigorously validated extraction–construction pipeline (figures, captions, related sentences, panels) ensures high-quality samples.

- **Comprehensive and Diagnostic Evaluation**: Extensive experiments on 16 leading MLLMs (including newly added 4 commercial models and 2 differently sized Qwen variants) reveal universal limitations, capability imbalances, and vulnerabilities to compound manipulations.

- **Methodological Insights for the Community**: Fine-grained error analyses and capability profiling provide actionable diagnostic signals. THEMIS serves as both a rigorous evaluation testbed and a scalable foundation for training stronger visual fraud reasoning models, advancing future multimodal scientific-review research.

We have incorporated all suggested improvements (e.g., expanded analyses, additional comparative experiments) into the updated version.

Thank all the reviewers for your invaluable feedback in strengthening our work.

Best regards,

Authors

---

### Meta-Review · Area_Chair_ZKwv · 2026-01-06

**Summary:**

This paper presents a large-scale multi-task benchmark,THEMIS, for evaluating multi-modal large language models (MLLMs), for “fraud reasoning” within scientific papers. The benchmark comprises of over 40k questions, spanning 7 academic scenarios, covering 5 fraud detection tasks, 16 fine-grained manipulations, and 5 visual fraud reasoning capabilities, with both synthetic and real-world samples. The authors provide a rigorous data construction pipeline (Fitz + YOLOv7 with manual verification and failure-case analysis), clear documentation, and extensive supplementary examples. Their evaluation on 11 MLLMs show that even state-of-the-art models struggle with localization and compound manipulations, highlighting clear research gaps and practical impact. The paper is well written and clearly presented.

**Reviewer Concerns:**

Reviewers main concerns included extraction reliability, presentation of benchmark examples, coverage of additional models, and real vs synthetic data analysis, all of which have been addressed satisfactorily.

**Reviewer Scores:**

Extensive discussion took place between reviewers and authors resulting in clarity and an overall stronger paper. The paper received a Rating of 8 from two reviewers and 6 from a third reviewer.

---

### Decision · Program_Chairs · 2026-01-26

Accept (Poster)